# A Sharp Analysis of Model-based Reinforcement Learning with Self-Play

## Abstract

Model-based algorithms—algorithms that explore the environment through building and utilizing an estimated model—are widely used in reinforcement learning practice and theoretically shown to achieve optimal sample efficiency for single-agent reinforcement learning in Markov Decision Processes (MDPs). However, for multi-agent reinforcement learning in Markov games, the current best known sample complexity for model-based algorithms is rather suboptimal and compares unfavorably against recent model-free approaches. In this paper, we present a sharp analysis of model-based self-play algorithms for multi-agent Markov games. We design an algorithm *Optimistic Nash Value Iteration* (Nash-VI) for two-player zero-sum Markov games that is able to output an $\epsilon$-approximate Nash policy in $\tilde{\mathcal{O}}(H^3 SAB/\epsilon^2)$ episodes of game playing, where $S$ is the number of states, $A, B$ are the number of actions for the two players respectively, and $H$ is the horizon length. This significantly improves over the best known model-based guarantee of $\tilde{\mathcal{O}}(H^4 S^2 AB/\epsilon^2)$, and is the first that matches the information-theoretic lower bound $\Omega(H^3 S(A + B)/\epsilon^2)$ except for a $\min\{A, B\}$ factor. In addition, our guarantee compares favorably against the best known model-free algorithm if $\min\{A, B\} = o(H^3)$, and outputs a single Markov policy while existing sample-efficient model-free algorithms output a nested mixture of Markov policies that is in general non-Markov and rather inconvenient to store and execute. We further adapt our analysis to designing a provably efficient task-agnostic algorithm for zero-sum Markov games, and designing the first line of provably sample-efficient algorithms for multi-player general-sum Markov games.

## 1 Introduction

This paper is concerned with the problem of multi-agent reinforcement learning (multi-agent RL), in which multiple agents learn to make decisions in an unknown environment in order to maximize their (own) cumulative rewards. Multi-agent RL has achieved significant recent success in traditionally hard AI challenges including large-scale strategy games (such as GO) (Silver et al., 2016; 2017), real-time video games involving team play such as Starcraft and Dota2 (OpenAI, 2018; Vinyals et al., 2019), as well as behavior learning in complex social scenarios (Baker et al., 2020). Achieving human-like (or super-human) performance in these games using multi-agent RL typically requires a large number of samples (steps of game playing) due to the necessity of exploration, and how to improve the sample complexity of multi-agent RL has been an important research question.

One prevalent approach towards solving multi-agent RL is *model-based* methods, that is, to use the existing visitation data to build an estimate of the model (i.e. transition dynamics and rewards), run an offline planning algorithm on the estimated model to obtain the policy, and play the policy in the environment. Such a principle underlies some of the earliest single-agent online RL algorithms such as E3 (Kearns & Singh, 2002) and RMax (Brafman & Tennenholtz, 2002), and is conceptually appealing for multi-agent RL too since the multi-agent structure does not add complexity onto the model estimation part and only requires an appropriate multi-agent planning algorithm (such as value iteration for games (Shapley, 1953)) in a black-box fashion. On the other hand, *model-free* methods do not directly build estimates of the model, but instead directly estimate the value functions or action-value (Q) functions of the problem at the optimal/equilibrium policies, and play the greedy policies with respect to the estimated value functions. Model-free algorithms have also

Table 1: Sample complexity (the required number of episodes) for algorithms to find $\epsilon$-approximate Nash equlibrium policies in zero-sum Markov games: VI-explore and VI-UCLB by Bai & Jin (2020), OMVI-SM by Xie et al. (2020), and Nash Q/V-learning by Bai et al. (2020). The lower bound was proved by Jin et al. (2018); Domingues et al. (2020).

| | Algorithm | Task-Agnostic | $\sqrt{T}$-Regret | Sample Complexity | Output Policy |
|---|---|---|---|---|---|
| Model-based | VI-explore | Yes | | $\tilde{\mathcal{O}}(H^5 S^2 AB/\epsilon^2)$ | a single Markov policy |
| | VI-ULCB | | Yes | $\tilde{\mathcal{O}}(H^4 S^2 AB/\epsilon^2)$ | |
| | OMVI-SM | | Yes | $\tilde{\mathcal{O}}(H^4 S^3 A^3 B^3/\epsilon^2)$ | |
| | Algorithm 2 | Yes | | $\tilde{\mathcal{O}}(H^4 SAB/\epsilon^2)$ | |
| | Algorithm 1 | | Yes | $\tilde{\mathcal{O}}(H^3 SAB/\epsilon^2)$ | |
| Model-free | Nash Q-learning | | | $\tilde{\mathcal{O}}(H^5 SAB/\epsilon^2)$ | nested mixture of Markov policies |
| | Nash V-learning | | | $\tilde{\mathcal{O}}(H^6 S(A+B)/\epsilon^2)$ | |
| | Lower Bound | - | - | $\Omega(H^3 S(A+B)/\epsilon^2)$ | - |

been well developed for multi-agent RL such as friend-or-foe Q-Learning (Littman, 2001) and Nash Q-Learning (Hu & Wellman, 2003).

While both model-based and model-free algorithms have been shown to be provably efficient in multi-agent RL in a recent line of work (Bai & Jin, 2020; Xie et al., 2020; Bai et al., 2020), a more precise understanding of the optimal sample complexities within these two types of algorithms (respectively) is still lacking. In the specific setting of two-player zero-sum Markov games, the current best sample complexity for model-based algorithms is achieved by the VI-ULCB (Value Iteration with Upper/Lower Confidence Bounds) algorithm (Bai & Jin, 2020; Xie et al., 2020): In a tabular Markov game with $S$ states, $\{A, B\}$ actions for the two players, and horizon length $H$, VI-ULCB is able to find an $\epsilon$-approximate Nash equilibrium policy in $\tilde{\mathcal{O}}(H^4 S^2 AB/\epsilon^2)$ episodes of game playing. However, compared with the information-theoretic lower bound $\Omega(H^3 S(A+B)/\epsilon^2)$, this rate has suboptimal dependencies on all of $H$, $S$, and $A$, $B$. In contrast, the current best sample complexity for *model-free* algorithms is achieved by Nash V-Learning (Bai et al., 2020), which finds an $\epsilon$-approximate Nash policy in $\tilde{\mathcal{O}}(H^6 S(A + B)/\epsilon^2)$ episodes. Compared with the lower bound, this is tight except for a $\text{poly}(H)$ factor, which may seemingly suggest that model-free algorithms could be superior to model-based ones in multi-agent RL. However, such a conclusion would be in stark contrast to the single-agent MDP setting, where it is known that model-based algorithms are able to achieve minimax optimal sample complexities (Jaksch et al., 2010; Azar et al., 2017). It naturally arises whether model-free algorithms are indeed superior in multi-agent settings, or whether the existing analyses of model-based algorithms are not tight. This motivates us to ask the following research question:

**Question**: How sample-efficient are model-based algorithms in multi-agent RL?

In this paper, we advance the theoretical understandings of multi-agent RL by presenting a sharp analysis of model-based algorithms on Markov games. Our core contribution is the design of a new model-based algorithm *Optimistic Nash Value Iteration* (Nash-VI) that achieves an almost optimal sample complexity for zero-sum Markov games and improves significantly over existing model-based approaches. We summarize our main contributions as follows. A comparison between our and prior results can be found in Table 1.

• We design a new model-based algorithm *Optimistic Nash Value Iteration* (Nash-VI) that provably finds $\epsilon$-approximate Nash equilibria for Markov games in $\tilde{\mathcal{O}}(H^3 SAB/\epsilon^2)$ episodes of game playing (Section 3). This improves over the best existing model-based algorithm by $O(HS)$ and is the first algorithm that matches the sample complexity lower bound except for a $\tilde{\mathcal{O}}(\min\{A, B\})$ factor, showing that model-based algorithms can indeed achieve an almost optimal sample complexity. Further, unlike state-of-the-art model-free algorithms such as Nash V-Learning (Bai et al., 2020), this algorithm achieves in addition a $\tilde{\mathcal{O}}(\sqrt{T})$ regret bound, and outputs a simple Markov policy (instead of a nested mixture of Markov policies as returned by Nash V-Learning).

- We design an alternative algorithm *Optimistic Value Iteration with Zero Reward* (VI-Zero) that is able to perform task-agnostic (reward-free) learning for multiple Markov games sharing the same transition (Section 4). For $N > 1$ games with the same transition and different (known) rewards, VI-Zero can find $\epsilon$-approximate Nash policy for all games simultaneously in $\tilde{\mathcal{O}}(H^4 SAB \log N/\epsilon^2)$ episodes of game playing, which scales logarithmically in the number of games.

- We design the first line of sample-efficient algorithms for *multi-player* general-sum Markov games. In a multi-player game with $M$ players and $A_i$ actions per player, we show that an $\epsilon$ near-optimal policy can be found in $\tilde{\mathcal{O}}(H^4 S^2 \prod_{i\in[M]} A_i/\epsilon^2)$ episodes, where the desired optimality can be either one of Nash equilibrium, correlated equilibrium (CE), or coarse correlated equilibrium (CCE). We achieve this guarantee by either a multi-player version of Nash-VI or a multi-player version of reward-free value iteration (Section 5 & Appendix C).

Due to space limit, we defer a detailed survey of related works to Appendix A.

## 2 PRELIMINARIES

In this paper, we consider Markov Games (MGs, Shapley, 1953; Littman, 1994), which are also known as stochastic games in the literature. Markov games are the generalization of standard Markov Decision Processes (MDPs) into the multi-player setting, where each player seeks to maximize her own utility. For simplicity, in this section we describe the important special case of *two-player zero-sum games*, and return to the general formulation in Appendix C.

Formally, we consider the tabular episodic version of two-player zero-sum Markov game, which we denote as $\mathrm{MG}(H, \mathcal{S}, \mathcal{A}, \mathcal{B}, \mathbb{P}, r)$. Here $H$ is the number of steps in each episode, $\mathcal{S}$ is the set of states with $|\mathcal{S}| \leq S$, $(\mathcal{A}, \mathcal{B})$ are the sets of actions of the max-player and the min-player respectively with $|\mathcal{A}| \leq A$ and $|\mathcal{B}| \leq B$, $\mathbb{P} = \{\mathbb{P}_h\}_{h\in[H]}$ is a collection of transition matrices, so that $\mathbb{P}_h(\cdot|s, a, b)$ gives the distribution of the next state if action pair $(a, b)$ is taken at state $s$ at step $h$, and $r = \{r_h\}_{h\in[H]}$ is a collection of reward functions, where $r_h \colon \mathcal{S} \times \mathcal{A} \times \mathcal{B} \to [0, 1]$ is the deterministic reward function at step $h$.[1] This reward represents both the gain of the max-player and the loss of the min-player, making the problem a zero-sum Markov game.

In each episode of this MG, we start with a *fixed initial state* $s_1$. At each step $h \in [H]$, both players observe state $s_h \in \mathcal{S}$, and pick their own actions $a_h \in \mathcal{A}$ and $b_h \in \mathcal{B}$ simultaneously. Then, both players observe the actions of their opponent, receive reward $r_h(s_h, a_h, b_h)$, and then the environment transitions to the next state $s_{h+1} \sim \mathbb{P}_h(\cdot|s_h, a_h, b_h)$. The episode ends when $s_{H+1}$ is reached.

**Policy, value function.** A (Markov) policy $\mu$ of the max-player is a collection of $H$ functions $\{\mu_h \colon \mathcal{S} \to \Delta_{\mathcal{A}}\}_{h\in[H]}$, each mapping from a state to a distribution over actions. (Here $\Delta_{\mathcal{A}}$ is the probability simplex over action set $\mathcal{A}$.) Similarly, a policy $\nu$ of the min-player is a collection of $H$ functions $\{\nu_h \colon \mathcal{S} \to \Delta_{\mathcal{B}}\}_{h\in[H]}$. We use the notation $\mu_h(a|s)$ and $\nu_h(b|s)$ to represent the probability of taking action $a$ or $b$ for state $s$ at step $h$ under Markov policy $\mu$ or $\nu$ respectively.

We use $V_h^{\mu,\nu} \colon \mathcal{S} \to \mathbb{R}$ to denote the value function at step $h$ under policy $\mu$ and $\nu$, so that $V_h^{\mu,\nu}(s)$ gives the expected cumulative rewards received under policy $\mu$ and $\nu$, starting from $s$ at step $h$:

$$V_h^{\mu,\nu}(s) := \mathbb{E}_{\mu,\nu}\left[\sum_{h'=h}^H r_{h'}(s_{h'}, a_{h'}, b_{h'}) \Big| s_h = s\right]. \tag{1}$$

We also define $Q_h^{\mu,\nu} \colon \mathcal{S} \times \mathcal{A} \times \mathcal{B} \to \mathbb{R}$ to be the Q-value function at step $h$ so that $Q_h^{\mu,\nu}(s, a, b)$ gives the cumulative rewards received under policy $\mu$ and $\nu$, starting from $(s, a, b)$ at step $h$:

$$Q_h^{\mu,\nu}(s, a, b) := \mathbb{E}_{\mu,\nu}\left[\sum_{h'=h}^H r_{h'}(s_{h'}, a_{h'}, b_{h'}) \Big| s_h = s, a_h = a, b_h = b\right]. \tag{2}$$

For simplicity, we define operator $\mathbb{P}_h$ as $[\mathbb{P}_h V](s, a, b) := \mathbb{E}_{s'\sim\mathbb{P}_h(\cdot|s,a,b)} V(s')$ for any value function $V$. We also use notation $[\mathbb{D}_\pi Q](s) := \mathbb{E}_{(a,b)\sim\pi(\cdot,\cdot|s)} Q(s, a, b)$ for any action-value function $Q$. By definition of value functions, we have the Bellman equation

$$Q_h^{\mu,\nu}(s, a, b) = (r_h + \mathbb{P}_h V_{h+1}^{\mu,\nu})(s, a, b), \qquad V_h^{\mu,\nu}(s) = (\mathbb{D}_{\mu_h \times \nu_h} Q_h^{\mu,\nu})(s)$$

---

[1] We assume the rewards in $[0, 1]$ for normalization. Our results directly generalize to randomized reward functions, since learning the transition is more difficult than learning the reward.

for all $(s, a, b, h) \in \mathcal{S} \times \mathcal{A} \times \mathcal{B} \times [H]$, and at the $(H+1)^{\text{th}}$ step we have $V_{H+1}^{\mu,\nu}(s) = 0$ for all $s \in \mathcal{S}$.

**Best response and Nash equilibrium.** For any policy of the max-player $\mu$, there exists a *best response* of the min-player, which is a policy $\nu^\dagger(\mu)$ satisfying $V_h^{\mu,\nu^\dagger(\mu)}(s) = \inf_\nu V_h^{\mu,\nu}(s)$ for any $(s,h) \in \mathcal{S} \times [H]$. We denote $V_h^{\mu,\dagger} := V_h^{\mu,\nu^\dagger(\mu)}$. By symmetry, we can also define $\mu^\dagger(\nu)$ and $V_h^{\dagger,\nu}$. It is further known (cf. (Filar & Vrieze, 2012)) that there exist policies $\mu^\star, \nu^\star$ that are optimal against the best responses of the opponents, in the sense that

$$V_h^{\mu^\star,\dagger}(s) = \sup_\mu V_h^{\mu,\dagger}(s), \qquad V_h^{\dagger,\nu^\star}(s) = \inf_\nu V_h^{\dagger,\nu}(s), \qquad \text{for all } (s,h).$$

We call these optimal strategies $(\mu^\star, \nu^\star)$ the Nash equilibrium of the Markov game, which satisfies the following minimax equation [2]:

$$\sup_\mu \inf_\nu V_h^{\mu,\nu}(s) = V_h^{\mu^\star,\nu^\star}(s) = \inf_\nu \sup_\mu V_h^{\mu,\nu}(s).$$

Intuitively, a Nash equilibrium gives a solution in which no player has anything to gain by changing only her own policy. We further abbreviate the values of Nash equilibrium $V_h^{\mu^\star,\nu^\star}$ and $Q_h^{\mu^\star,\nu^\star}$ as $V_h^\star$ and $Q_h^\star$. We refer readers to Appendix D for Bellman optimality equations for (the value functions) of) the best responses and the Nash equilibrium.

**Learning Objective.** We measure the suboptimality of any pair of general policies $(\hat\mu, \hat\nu)$ using the gap between their performance and the performance of the optimal strategy (i.e., Nash equilibrium) when playing against the best responses respectively:

$$V_1^{\dagger,\hat\nu}(s_1) - V_1^{\hat\mu,\dagger}(s_1) = \left[ V_1^{\dagger,\hat\nu}(s_1) - V_1^\star(s_1) \right] + \left[ V_1^\star(s_1) - V_1^{\hat\mu,\dagger}(s_1) \right]$$

**Definition 1** ($\epsilon$-approximate Nash equilibrium). A pair of general policies $(\hat\mu, \hat\nu)$ is an $\epsilon$-**approximate Nash equilibrium**, if $V_1^{\dagger,\hat\nu}(s_1) - V_1^{\hat\mu,\dagger}(s_1) \le \epsilon$.

**Definition 2** (Regret). Let $(\mu^k, \nu^k)$ denote the policies deployed by the algorithm in the $k^{\text{th}}$ episode. After a total of $K$ episodes, the regret is defined as

$$\text{Regret}(K) = \sum_{k=1}^K (V_1^{\dagger,\nu^k} - V_1^{\mu^k,\dagger})(s_1).$$

One goal of reinforcement learning is to design algorithms for Markov games that can find an $\epsilon$-approximate Nash equilibrium using a number of episodes that is small in its dependency on $S, A, B, H$ as well as $1/\epsilon$ (PAC sample complexity bound). An alternative goal is to design algorithms for Markov games that achieves regret that is sublinear in $K$, and polynomial in $S, A, B, H$ (regret bound). We remark that any sublinear regret algorithm can be directly converted to a polynomial-sample PAC algorithm via the standard online-to-batch conversion (see e.g., Jin et al. (2018)).

## 3 OPTIMISTIC NASH VALUE ITERATION

In this section, we present our main algorithm—Optimistic Nash Value Iteration (Nash-VI), and provide its theoretical guarantee.

### 3.1 ALGORITHM DESCRIPTION

We describe our Nash-VI Algorithm 1. In each episode, the algorithm can be decomposed into two parts.

- Line 3-13 (Optimistic planning from the estimated model): Performs value iteration with bonus using the empirical estimate of the transition $\hat{\mathbb{P}}$, and computes a new (joint) policy $\pi$ which is "greedy" with respect to the estimated value functions;

---

[2]The minimax theorem here is different from the one for matrix games, i.e. $\max_\phi \min_\psi \phi^\top A \psi = \min_\psi \max_\phi \phi^\top A \psi$ for any matrix $A$, since here $V_h^{\mu,\nu}(s)$ is in general not bilinear in $\mu, \nu$.

- Line 16-19 (Play the policy and update the model estimate): Executes the policy $\pi$, collects samples, and updates the estimate of the transition $\hat{\mathbb{P}}$.

At a high-level, this two-phase strategy is standard in the majority of model-based RL algorithms, and also underlies provably efficient model-based algorithms such as UCBVI for single-agent (MDP) setting (Azar et al., 2017) and VI-ULCB for the two-player Markov game setting (Bai & Jin, 2020). However, VI-ULCB has two undesirable drawbacks: the sample complexity is not tight in any of $H$, $S$, and $A$, $B$ dependency, and its computational complexity is PPAD-complete (a complexity class conjectured to be computationally hard (Daskalakis, 2013)).

As we elaborate in the following, our Nash-VI algorithm differs from VI-ULCB in a few important technical aspects, which allows it to significantly improve the sample complexity over VI-ULCB, and ensures that our algorithm terminates in polynomial time.

Before digging into explanations of techniques, we remark that line 14-15 is only used for computing the output policies. It chooses policy $\pi^{\text{out}}$ to be the policy in the episode with minimum gap $(\overline{V}_1 - \underline{V}_1)(s_1)$. Our final output policies $(\mu^{\text{out}}, \nu^{\text{out}})$ are simply the *marginal policies* of $\pi^{\text{out}}$. That is, for all $(s, h) \in \mathcal{S} \times [H]$, $\mu_h^{\text{out}}(\cdot|s) := \sum_{b \in \mathcal{B}} \pi_h^{\text{out}}(\cdot, b|s)$, and $\nu_h^{\text{out}}(\cdot|s) := \sum_{a \in \mathcal{A}} \pi_h^{\text{out}}(a, \cdot|s)$.

### 3.1.1 OVERVIEW OF TECHNIQUES

**Auxiliary bonus $\gamma$.** The major improvement over VI-ULCB (Bai & Jin, 2020) comes from the use of a different style of bonus term $\gamma$ (line 8), in addition to the standard bonus $\beta$ (line 7), in value iteration steps (line 9-10). This is also the main technical contribution of our Nash-VI algorithm. This auxiliary bonus $\gamma$ is computed by applying the empirical transition matrix $\hat{\mathbb{P}}_h$ to the gap at the next step $\overline{V}_{h+1} - \underline{V}_{h+1}$, This is very different from standard bonus $\beta$, which is typically designed according to the concentration inequalities.

The main purpose of these value iteration steps (line 9-10) is to ensure that the estimated values $\overline{Q}_h$ and $\underline{Q}_h$ are with high probability the upper bound and the lower bound of the $Q$-value of the current policy when facing best responses (see Lemma 20 and 22 for more details) [3]. To do so, prior work (Bai & Jin, 2020) only adds bonus $\beta$, which needs to be as large as $\tilde{\Theta}(\sqrt{S/t})$. In contrast, the inclusion of auxiliary bonus $\gamma$ in our algorithm allows a much smaller choice for bonus $\beta$—which scales only as $\tilde{\mathcal{O}}(\sqrt{1/t})$—while still maintaining valid confidence bounds. This technique alone brings down the sample complexity to $\tilde{\mathcal{O}}(H^4 SAB/\epsilon^2)$, removing an entire $S$ factor compared to VI-ULCB. Furthermore, the coefficient in $\gamma$ is only $c/H$ for some absolute constant $c$, which ensures that the introduction of error term $\gamma$ would hurt the overall sample complexity only up to a constant factor.

**Bernstein concentration.** Our Nash-VI allows two choices of the bonus function $\beta = \text{BONUS}(t, \hat{\sigma}^2)$:

$$\text{Hoeffding type: } c(\sqrt{H^2\iota/t} + H^2 S\iota/t), \qquad \text{Bernstein type: } c(\sqrt{\hat{\sigma}^2\iota/t} + H^2 S\iota/t). \quad (3)$$

where $\hat{\sigma}^2$ is the estimated variance, $\iota$ is the logarithmic factors and $c$ is absolute constant. The $\hat{\mathbb{V}}$ in line 7 is the empirical variance operator defined as $\hat{\mathbb{V}}_h V = \hat{\mathbb{P}}_h V^2 - (\hat{\mathbb{P}}_h V)^2$ for any $V \in [0, H]^S$. The design of both bonuses stem from the Hoeffding and Bernstein concentration inequalities. Further, the Bernstein bonus uses a sharper concentration, which saves an $H$ factor in sample complexity compared to the Hoeffding bonus (similar to the single-agent setting (Azar et al., 2017)). This further reduces the sample complexity to $\tilde{\mathcal{O}}(H^3 SAB/\epsilon^2)$ which matches the lower bound in all $H, S, \epsilon$ factors.

**Coarse Correlated Equalibirum (CCE).** The prior algorithm VI-ULCB (Bai & Jin, 2020) computes the "greedy" policy with respect to the estimated value functions by directly computing the Nash equilibrium for the $Q$-value at each step $h$. However, since the algorithm maintains both the upper confidence bound and lower confidence bound of the $Q$-value, this leads to the requirement

---

[3]We remark that the current policy is stochastic. This is different from the single-agent setting, where the algorithm only seeks to provide an upper bound of the value of the optimal policy where the optimal policy is not random. Due to this difference, the techniques of Azar et al. (2017) cannot be directly applied here.

---

**Algorithm 1** Optimistic Nash Value Iteration (Nash-VI)

---

1: **Initialize:** for any $(s, a, b, h), \overline{Q}_h(s, a, b) \leftarrow H, \underline{Q}_h(s, a, b) \leftarrow 0, \Delta \leftarrow H, N_h(s, a, b) \leftarrow 0.$
2: **for** episode $k = 1, \ldots, K$ **do**
3:      **for** step $h = H, H - 1, \ldots, 1$ **do**
4:          **for** $(s, a, b) \in \mathcal{S} \times \mathcal{A} \times \mathcal{B}$ **do**
5:             $t \leftarrow N_h(s, a, b).$
6:             **if** $t > 0$ **then**
7:                 $\beta \leftarrow \text{BONUS}(t, \widehat{\mathbb{V}}_h[(\overline{V}_{h+1} + \underline{V}_{h+1})/2](s, a, b)).$
8:                 $\gamma \leftarrow (c/H)\widehat{\mathbb{P}}_h(\overline{V}_{h+1} - \underline{V}_{h+1})(s, a, b).$
9:                 $\overline{Q}_h(s, a, b) \leftarrow \min\{(r_h + \widehat{\mathbb{P}}_h \overline{V}_{h+1})(s, a, b) + \gamma + \beta, H\}.$
10:                $\underline{Q}_h(s, a, b) \leftarrow \max\{(r_h + \widehat{\mathbb{P}}_h \underline{V}_{h+1})(s, a, b) - \gamma - \beta, 0\}.$
11:          **for** $s \in \mathcal{S}$ **do**
12:             $\pi_h(\cdot, \cdot | s) \leftarrow \text{CCE}(\overline{Q}_h(s, \cdot, \cdot), \underline{Q}_h(s, \cdot, \cdot)).$
13:             $\overline{V}_h(s) \leftarrow (\mathbb{D}_{\pi_h} \overline{Q}_h)(s); \quad \underline{V}_h(s) \leftarrow (\mathbb{D}_{\pi_h} \underline{Q}_h)(s).$
14:      **if** $(\overline{V}_1 - \underline{V}_1)(s_1) < \Delta$ **then**
15:          $\Delta \leftarrow (\overline{V}_1 - \underline{V}_1)(s_1)$ and $\pi^{\text{out}} \leftarrow \pi.$
16:      **for** step $h = 1, \ldots, H$ **do**
17:          take action $(a_h, b_h) \sim \pi_h(\cdot, \cdot | s_h)$, observe reward $r_h$ and next state $s_{h+1}.$
18:          add 1 to $N_h(s_h, a_h, b_h)$ and $N_h(s_h, a_h, b_h, s_{h+1}).$
19:      $\widehat{\mathbb{P}}_h(\cdot | s_h, a_h, b_h) \leftarrow N_h(s_h, a_h, b_h, \cdot)/N_h(s_h, a_h, b_h).$
20: **Output** $(\mu^{\text{out}}, \nu^{\text{out}})$ that are the marginal policies of $\pi^{\text{out}}.$

---

to compute the Nash equilibrium for a two-player general-sum matrix game, which is in general PPAD-complete (Daskalakis, 2013).

To overcome this computational challenge, we compute a relaxation of the Nash equilibrium—*Coarse Correlated Equalibirum (CCE)*—instead, a technique first introduced by Xie et al. (2020) to address reinforcement learning problems in Markov Games. Formally, for any pair of matrices $\overline{Q}, \underline{Q} \in [0, H]^{A \times B}$, $\text{CCE}(\overline{Q}, \underline{Q})$ returns a distribution $\pi \in \Delta_{\mathcal{A} \times \mathcal{B}}$ such that

$$\mathbb{E}_{(a,b) \sim \pi} \overline{Q}(a, b) \geq \max_{a^\star} \mathbb{E}_{(a,b) \sim \pi} \overline{Q}(a^\star, b), \qquad \mathbb{E}_{(a,b) \sim \pi} \underline{Q}(a, b) \leq \min_{b^\star} \mathbb{E}_{(a,b) \sim \pi} \underline{Q}(a, b^\star). \quad (4)$$

Intuitively, in a CCE the players choose their actions in a potentially correlated way such that no one can benefit from unilateral unconditional deviation. A CCE always exists, since Nash equilibrium is also a CCE and a Nash equilibrium always exists. Furthermore, a CCE can be computed by linear programming in polynomial time. We remark that different from Nash equilibrium where the policies of each player are independent, the policies given by CCE are in general correlated for each player. Therefore, executing such a policy (line 17) requires the cooperation of two players.

## 3.2 THEORETICAL GUARANTEES

Now we are ready to present the theoretical guarantees for Algorithm 1. We let $\pi^k$ denote the policy computed in line 12 in the $k^{\text{th}}$ episode, and $\mu^k, \nu^k$ denote the *marginal policy* of $\pi^k$ for each player.

**Theorem 3** (Nash-VI with Hoeffding bonus). *For any $p \in (0, 1]$, letting $\iota = \log(SABT/p)$, then with probability at least $1 - p$, Algorithm 1 with Hoeffding type bonus (3) (with some absolute $c > 0$) achieves:*

- $(V_1^{\dagger, \nu^{\text{out}}} - V_1^{\mu^{\text{out}}, \dagger})(s_1) \leq \epsilon$, *if the number of episodes $K \geq \Omega(H^4 SAB\iota/\epsilon^2 + H^3 S^2 AB\iota^2/\epsilon)$.*

- $\text{Regret}(K) = \sum_{k=1}^{K} (V_1^{\dagger, \nu^k} - V_1^{\mu^k, \dagger})(s_1) \leq \mathcal{O}(\sqrt{H^3 SABT\iota} + H^3 S^2 AB\iota^2).$

Theorem 3 provides both a sample complexity bound and a regret bound for Nash-VI to find an $\epsilon$-approximate Nash equilibrium. For small $\epsilon \leq H/(S\iota)$, the sample complexity scales as $\tilde{\mathcal{O}}(H^4 SAB/\epsilon^2)$. Similarly, for large $T \geq H^3 S^3 AB\iota^3$, the regret scales as $\tilde{\mathcal{O}}(\sqrt{H^3 SABT})$, where $T = KH$ is the total number of steps played within $K$ episodes. Theorem 3 is significant in that it

improves the sample complexity of the model-based algorithm in Markov games from $S^2$ to $S$ (and the regret from $S$ to $\sqrt{S}$). This is achieved by adding the new auxiliary bonus $\gamma$ in value iteration steps as explained in Section 3.1. The proof of Theorem 3 can be found in Appendix F.1.

Our next theorem states that when using Bernstein bonus instead of Hoeffding bonus as in (3), the sample complexity of Nash-VI algorithm can be further improved by a $H$ factor in the leading order term (and the regret improved by a $\sqrt{H}$ factor).

**Theorem 4** (Nash-VI with the Bernstein bonus). *For any $p \in (0,1]$, letting $\iota = \log(SABT/p)$, then with probability at least $1 - p$, Algorithm 1 with Bernstein type bonus (3) (with some absolute $c > 0$) achieves:*

- $(V_1^{\dagger,\nu^{out}} - V_1^{\mu^{out},\dagger})(s_1) \leq \epsilon$, *if the number of episodes* $K \geq \Omega(H^3SAB\iota/\epsilon^2 + H^3S^2AB\iota^2/\epsilon)$.

- $\mathrm{Regret}(K) = \sum_{k=1}^{K}(V_1^{\dagger,\nu^k} - V_1^{\mu^k,\dagger})(s_1) \leq \mathcal{O}(\sqrt{H^2SABT\iota} + H^3S^2AB\iota^2)$.

Compared with the information-theoretic sample complexity lower bound $\Omega(H^3S(A+B)\iota/\epsilon^2)$ and regret lower bound $\Omega(\sqrt{H^2S(A+B)T})$ (Bai & Jin, 2020), when $\epsilon$ is small, Nash-VI with Bernstein bonus achieves the optimal dependency on all of $H, S, \epsilon$ up to logarithmic factors in both the sample complexity and the regret, and the only gap that remains open is a $AB/(A+B) \leq \min\{A, B\}$ factor. The proof of Theorem 4 can be found in Appendix F.2.

**Comparison with model-free approaches.** Different from our model-based approach, a recently proposed model-free algorithm Nash V-Learning (Bai et al., 2020) achieves sample complexity $\tilde{\mathcal{O}}(H^6S(A+B)\iota/\epsilon^2)$, which has a tight $(A+B)$ dependency on $A, B$. However, our Nash-VI has the following important advantages over Nash V-Learning: 1. Our sample complexity has a better dependency on horizon $H$; 2. Our algorithm outputs a single pair of Markov policies $(\mu^{out}, \nu^{out})$ while their algorithm outputs a generic history-dependent policy that can be only written as a nested mixture of Markov policies; 3. The model-free algorithms in Bai et al. (2020) cannot be directly modified to obtain a $\sqrt{T}$-regret (so that the exploration policies can be arbitrarily poor), while our model-based algorithm has the $\sqrt{T}$-regret guarantee. We comment that although both Nash-VI and Nash V-Learning have polynomial running time, the later enjoys a better computational complexity because Nash-VI requires to solve LPs for computing CCEs in each episode.

## 4 REWARD-FREE LEARNING

In this section, we modify our model-based algorithm Nash-VI for the reward-free exploration setting (Jin et al., 2020b), which is also known as the task-agnostic (Zhang et al., 2020b) or reward-agnostic setting. Reward-free learning has two phases: In the exploration phase, the agent first collects a dataset of transitions $\mathcal{D} = \{(s_{k,h}, a_{k,h}, b_{k,h}, s_{k,h+1})\}_{(k,h) \in [K] \times [H]}$ from a Markov game $\mathcal{M}$ without the guidance of reward information. After the exploration, in the planning phase, for each task $i \in [N]$, $\mathcal{D}$ is augmented with stochastic reward information to become $\mathcal{D}^i = \{(s_{k,h}, a_{k,h}, b_{k,h}, s_{k,h+1}, r_{k,h})\}_{(k,h) \in [K] \times [H]}$, where $r_{k,h}$ is sampled from an unknown reward distribution with expectation equal to $r_h^i(s_{k,h}, a_{k,h}, b_{k,h})$. Here, we use $r^i$ to refer to the unknown reward function of the $i^{\mathrm{th}}$ task. The goal is to compute nearly-optimal policies for $N$ tasks under $\mathcal{M}$, given the augmented datasets.

There are strong practical motivations for considering the reward-free setting. First, in applications such as robotics, we face multiple tasks in sequential systems with shared transition dynamics (i.e. the world) but very different rewards. There, we prefer to learn the underlying transition independent of reward information. Second, from the algorithm design perspective, decoupling exploration and planning (i.e. performing exploration without reward information) can be valuable for designing new algorithms in more challenging settings (e.g. with function approximation).

Due to space limits, we defer the description of our algorithm Optimistic Value Iteration with Zero Reward (VI-Zero, Algorithm 2) to Appendix B and only state its theoretical guarantees here. The following theorem claims that the empirical transition $\widehat{\mathbb{P}}^{out}$ outputted by VI-Zero is close to the true transition $\mathbb{P}$, in the sense that any Nash equilibrium of the $\mathcal{M}(\widehat{\mathbb{P}}, \widehat{r}^i)$ ($i \in [N]$) is also an approximate

Nash equilibrium of the true underlying Markov game $\mathcal{M}(\mathbb{P}, r^i)$, where $\widehat{r}^i$ is the empirical estimate of $r^i$ computed using $\mathcal{D}^i$.

**Theorem 5** (Sample complexity of VI-Zero). *There exists an absolute constant $c$, for any $p \in (0, 1]$, $\epsilon \in (0, H]$, $N \in \mathbb{N}$, if we choose bonus $\beta_t = c(\sqrt{H^2\iota/t} + H^2 S\iota/t)$ with $\iota = \log(NSABT/p)$ and $K \geq c(H^4 SAB\iota/\epsilon^2 + H^3 S^2 AB\iota^2/\epsilon)$, then with probability at least $1 - p$, the output $\widehat{\mathbb{P}}^{\mathrm{out}}$ of Algorithm 2 has the following property: for any $N$ fixed reward functions $r^1, \ldots, r^N$, a Nash equilibrium of Markov game $\mathcal{M}(\widehat{\mathbb{P}}^{\mathrm{out}}, \widehat{r}^i)$ is also an $\epsilon$-approximate Nash equilibrium of the true Markov game $\mathcal{M}(\mathbb{P}, r^i)$ for all $i \in [N]$.*

Theorem 5 shows that, when $\epsilon$ is small, VI-Zero only needs $\tilde{\mathcal{O}}(H^4 SAB/\epsilon^2)$ samples to learn an estimate of the transition $\widehat{\mathbb{P}}^{\mathrm{out}}$, which is accurate enough to learn the approximate Nash equilibrium for any $N$ fixed rewards. The most important advantage of reward-free learning comes from the sample complexity only scaling polylogarithmically with respect to the number of tasks or reward functions $N$. This is in sharp contrast to the reward-aware algorithms (e.g. Nash-VI), where the algorithm has to be rerun for each different task, and the total sample complexity must scale linearly in $N$. In exchange for this benefit, compared to Nash-VI, VI-Zero loses a factor of $H$ in the leading term of sample complexity since we cannot use Bernstein bonus anymore due to the lack of reward information. VI-Zero also does not have a regret guarantee, since again without reward information, the exploration policies are naturally sub-optimal. The proof of Theorem 5 can be found in Appendix G.1.

**Connections with reward-free learning in MDPs.** Since MDPs are special cases of Markov games, our algorithm VI-Zero directly applies to the single-agent setting, and yields a sample complexity similar to existing results (Zhang et al., 2020b; Wang et al., 2020). However, distinct from existing results which require both the exploration algorithm and the planning algorithm to be specially designed to work together, our algorithm allows an arbitrary planning algorithm as long as it computes the Nash equilibrium of a Markov game with *known* transition and reward. Therefore, our results completely decouple the exploration and the planning.

**Lower bound for reward-free learning.** Finally, we comment that despite the sample complexity in Theorem 5 scaling as $AB$ instead of $A+B$, our next theorem states that unlike the general reward-aware setting, this $AB$ scaling is unavoidable in the reward-free setting. This reveals an intrinsic gap between the reward-free and reward-aware learning: An $A + B$ dependency is only achievable via sampling schemes that are reward-aware. A similar lower bound is also presented in recent work (Zhang et al., 2020a) for the discounted setting with a different hard instance construction.

**Theorem 6** (Lower bound for reward-free learning of Markov games). *There exists an absolute constant $c > 0$ such that for any $\epsilon \in (0, c]$, there exists a family of Markov games $\mathfrak{M}(\epsilon)$ satisfying that: for any reward-free algorithm $\mathfrak{A}$ using $K \leq cH^2 SAB/\epsilon^2$ episodes, there exists a Markov game $\mathcal{M} \in \mathfrak{M}(\epsilon)$ such that if we run $\mathfrak{A}$ on $\mathcal{M}$ and output policies $(\hat{\mu}, \hat{\nu})$, then with probability at least $1/4$, we have $(V_1^{\dagger, \hat{\nu}} - V_1^{\hat{\mu}, \dagger})(s_1) \geq \epsilon$.*

This lower bound shows that the sample complexity in Theorem 5 is optimal in $S$, $A$, $B$, and $\epsilon$. The proof of Theorem 6 can be found in Appendix G.3.

## 5 MULTI-PLAYER GENERAL-SUM GAMES

We adapt our analysis to multi-player general-sum games and present the first lines of provably efficient algorithms. Concretely, we design two model-based algorithms Multi-Nash-VI and Multi-VI-Zero (Algorithm 3 and Algorithm 4) that can find an ($\epsilon$-approximate) {NASH, CE, CCE} equilibrium for any multi-player general-sum Markov game in $\tilde{\mathcal{O}}(H^4 S^2 \prod_{i=1}^{m} A_i/\epsilon^2)$ episodes of game playing, where $A_i$ is the number of actions for player $i \in \{1, \ldots, m\}$ (Theorem 15 and Theorem 16). Due to space limit, we defer the detailed setups, algorithms and results to Appendix C.

## 6 CONCLUSION

In this paper, we provided a sharp analysis of model-based algorithms for Markov games. Our new algorithm Nash-VI can find an $\epsilon$-approximate Nash equilibrium of a zero-sum Markov game in $\tilde{\mathcal{O}}(H^3 SAB/\epsilon^2)$ episodes of game playing, which almost matches the sample complexity lower bound except for the $AB$ vs. $A + B$ dependency. We also applied our analysis to derive new efficient algorithms for task-agnostic game playing, as well as the first line of multi-player general-sum Markov games. There are a number of compelling future directions to this work. For example, can we achieve $A + B$ instead of $AB$ sample complexity for zero-sum games using model-based approaches (thus closing the gap between lower and upper bounds)? How can we design more efficient algorithms for general-sum games with better sample complexity (e.g., $\mathcal{O}(S)$ instead of $\mathcal{O}(S^2)$)? We leave these problems as future work.

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

## A    RELATED WORK

**Markov games.**    Markov games (or stochastic games) are proposed in the early 1950s (Shapley, 1953). They are widely used to model multi-agent RL. Learning the Nash equilibria of Markov games has been studied in Littman (1994; 2001); Hu & Wellman (2003); Hansen et al. (2013); Lee et al. (2020), where the transition matrix and reward are assumed to be known, or in the asymptotic setting where the number of data goes to infinity. These results do not directly apply to the non-asymptotic setting where the transition and reward are unknown and only a limited amount of data are available for estimating them.

Another line of work assumes certain strong reachability assumptions under which sophisticated exploration strategies are not required. A prevalent approach is to assume access to simulators (generative models) that enable the agent to directly sample transition and reward information for any state-action pair. In this setting, Jia et al. (2019); Sidford et al. (2019); Zhang et al. (2020a) provide non-asymptotic bounds on the number of calls to the simulator for finding an $\epsilon$ approximate Nash equilibrium. Wei et al. (2017) studies Markov games under an alternative assumption that no matter what strategy one agent sticks to, the other agent can always reach all states by playing a certain policy.

**Non-asymptotic guarantees without reachability assumptions.**    The recent work of Bai & Jin (2020); Xie et al. (2020) provide the first line of non-asymptotic sample complexity guarantees on learning Markov games without these reachability assumptions, in which exploration is essential. However, both results suffer from highly suboptimal sample complexity. The results of Xie et al. (2020) also apply to the linear function approximation setting. More recently, two model-free algorithms—Nash Q-Learning and Nash V-Learning—are shown to achieve better sample complexity guarantees (Bai et al., 2020). In particular, the Nash V-learning algorithm achieves the near-optimal dependence on $S$, $A$ and $B$. However, the dependence on $H$ is worse than our results and the output policy is a nested mixture, which is hard to implement. We compare our results with existing non-asymptotic guarantees in Table 1.

We remark that the classical R-max algorithm (Brafman & Tennenholtz, 2002) also provides provable guarantees for learning Markov games. However, Brafman & Tennenholtz (2002) uses a weaker definition of regret (similar to the online setting in Xie et al. (2020)), and consequently their result does not imply any sample complexity result for finding Nash equilibrium policies.

**Adversarial MDPs.**    Another way to model the multi-player bahavior is to use *adversarial MDPs*. Most work in this line considers the setting with adversarial rewards (Zimin & Neu, 2013; Rosenberg & Mansour, 2019; Jin et al., 2019), where the reward can be manipulated by an adversary arbitrarily and the goal is to compete with the optimal (stationary) policy in hindsight. Adversarial MDP with changing dynamics is computationally hard even under full-information feedback (Yadkori et al., 2013). Notice these results do not directly imply provable self-play algorithms in our setting, because the opponent in Markov games can affect both the reward and the transition.

**Single-agent RL.**    There is a rich literature on reinforcement learning in MDPs (see e.g., Jaksch et al., 2010; Osband et al., 2014; Azar et al., 2017; Dann et al., 2017; Strehl et al., 2006; Jin et al., 2018). MDP is a special case of Markov games, where only a single agent interacts with a stochastic environment. For the tabular episodic setting with nonstationary dynamics and no simulators, the best sample complexity is $\tilde{\mathcal{O}}(H^3SA/\epsilon^2)$, achieved by model-based algorithm in Azar et al. (2017) and model-free algorithms in Zhang et al. (2020c), respectively, where $S$ is the number of states, $A$ is the number of actions, $H$ is the length of each episode. Both of them match the lower bound $\Omega(H^3SA/\epsilon^2)$ (Jaksch et al., 2010; Osband & Van Roy, 2016; Jin et al., 2018).

**Reward-free and task-agnostic exploration.**    Jin et al. (2020a) proposes a new paradigm of learning an MDP, which they called reward-free exploration. In this setting, the agent goes through a two-stage process. In the exploration phase the agent can interacts with the environment without knowing the reward function and in the planning phase the reward function is given and the agent needs output a policy. The goal is to make the output policy near optimal for any given reward function. A closely related setting is task-agnostic learning, where the reward function is determined at the very beginning but not revealed until the planning phase. Notice algorithms for task-agnostic

---

**Algorithm 2** Optimistic Value Iteration with Zero Reward (VI-Zero)

---

**Require:** Bonus $\beta_t$.
1: **Initialize:** for any $(s, a, b, h)$, $\widetilde{V}_h(s, a, b) \leftarrow H$, $\Delta \leftarrow H$, $N_h(s, a, b) \leftarrow 0$.
2: **for** episode $k = 1, \dots, K$ **do**
3:     **for** step $h = H, H - 1, \dots, 1$ **do**
4:         **for** $(s, a, b) \in \mathcal{S} \times \mathcal{A} \times \mathcal{B}$ **do**
5:             $t \leftarrow N_h(s, a, b)$.
6:             **if** $t > 0$ **then**
7:                 $\widetilde{Q}_h(s, a, b) \leftarrow \min\{(\widehat{\mathbb{P}}_h \widetilde{V}_{h+1})(s, a, b) + \beta_t, H\}$.
8:         **for** $s \in \mathcal{S}$ **do**
9:             $\pi_h(s) \leftarrow \arg\max_{(a, b) \in \mathcal{A} \times \mathcal{B}} \widetilde{Q}_h(s, a, b)$.
10:             $\widetilde{V}_h(s) \leftarrow (\mathbb{D}_{\pi_h} \widetilde{Q}_h)(s)$.
11:     **if** $\widetilde{V}_1(s_1) < \Delta$ **then**
12:         $\Delta \leftarrow \widetilde{V}_1(s_1)$ and $\widehat{\mathbb{P}}^{\text{out}} \leftarrow \widehat{\mathbb{P}}$.
13:     **for** step $h = 1, \dots, H$ **do**
14:         take action $(a_h, b_h) \sim \pi_h(\cdot, \cdot|s_h)$, observe next state $s_{h+1}$.
15:         add 1 to $N_h(s_h, a_h, b_h)$ and $N_h(s_h, a_h, b_h, s_{h+1})$.
16:     $\widehat{\mathbb{P}}_h(\cdot|s_h, a_h, b_h) \leftarrow N_h(s_h, a_h, b_h, \cdot)/N_h(s_h, a_h, b_h)$.
17: **Output** $\widehat{\mathbb{P}}^{\text{out}}$.

---

learning can also be transferred to reward-free exploration by taking union bound w.r.t. different possible reward function. In Table 1, VI-explore (Bai & Jin, 2020) and Algorithm 2 can also be applied to this setting.

Jin et al. (2020a) also proposes an algorithm, which first finds a covering policy to maximize the probability to reach each state separately and then collects data following this policy. Zhang et al. (2020b) takes a different approach by first runs the optimistic Q-learning algorithm (Jin et al., 2018) with zero reward to explore the environment, and then they utilizes the trajectories collected to compute a policy in an incremental manner. Wang et al. (2020) follows a simialr scheme, but studies reward-free exploration in linear-parametrized MDPs.

# B    OPTIMISTIC VALUE ITERATION WITH ZERO REWARD – VI-ZERO

We now describe our algorithm for reward-free learning in zero-sum Markov games.

**Exploration phase.**    In the first phase of reward-free learning, we deploy algorithm Optimistic Value Iteration with Zero Reward (VI-Zero, Algorithm 2). This algorithm differs from the reward-aware Nash-VI (Algorithm 1) in two important aspects. First, we use zero reward in the exploration phase (Line 7), and only maintains an upper bound of the (reward-free) value function instead of both upper and lower bounds. Second, our exploration policy is the maximizing (instead of CCE) policy of the value function (Line 9). We remark that the $\widetilde{Q}_h(s, a, b)$ maintained in the algorithm 2 is no longer an upper bound for any actual value function (as it has no reward), but rather a measure of uncertainty or suboptimality that the agent may suffer—if she takes action $(a, b)$ at state $s$ and step $h$, and makes decisions by utilizing the empirical estimate $\widehat{\mathbb{P}}$ in the remaining steps (see a rigorous version of this statement in Lemma 27). Finally, the empirical transition $\widehat{\mathbb{P}}$ of the episode that minimizes $\widetilde{V}_1(s_1)$ is outputted and passed to the planning phase.

**Planning phase.**    After obtaining the estimate of tranisiton $\widehat{\mathbb{P}}$, our planning algorithm is rather simple. For the $i^{\text{th}}$ task, let $\widehat{r}^i$ be the empirical estimate of $r^i$ computed using the $i^{\text{th}}$ augmented dataset $\mathcal{D}^i$. Then we compute the Nash equilibrium of the Markov game $\mathcal{M}(\widehat{\mathbb{P}}, \widehat{r}^i)$ with estimated transition $\widehat{\mathbb{P}}$ and reward $\widehat{r}^i$. Since both $\widehat{\mathbb{P}}$ and $\widehat{r}^i$ are known exactly, this is a pure computation problem without any sampling error and can be efficiently solved by simple planning algorithms such as the vanilla Nash value iteration without optimism (see Appendix G.2 for more details).

## C  MULTIPLAYER GENERAL-SUM MARKOV GAMES

In this section, we extend both our model-based algorithms (Algorithm 1 and Algorithm 2) to the setting of multiplayer general-sum Markov games, and present corresponding theoretical guarantees.

### C.1  PROBLEM FORMULATION

A general-sum Markov game (general-sum MG) with $m$ players is a tuple $\text{MG}(H, \mathcal{S}, \{\mathcal{A}_i\}_{i=1}^m, \mathbb{P}, \{r_i\}_{i=1}^m)$, where $H$, $\mathcal{S}$ denote the length of each episode and the state space. Different from the two-player zero-sum setting, we now have $m$ different action spaces, where $\mathcal{A}_i$ is the action space for the $i^{\text{th}}$ player and $|\mathcal{A}_i| = A_i$. We let $\boldsymbol{a} := (a_1, \cdots, a_m)$ denote the (tuple of) joint actions by all $m$ players. $\mathbb{P} = \{\mathbb{P}_h\}_{h \in [H]}$ is a collection of transition matrices, so that $\mathbb{P}_h(\cdot|s, \boldsymbol{a})$ gives the distribution of the next state if actions $\boldsymbol{a}$ are taken at state $s$ at step $h$, and $r_i = \{r_{h,i}\}_{h \in [H]}$ is a collection of reward functions for $i^{\text{th}}$ player, so that $r_{h,i}(s, \boldsymbol{a})$ gives the reward received by the $i^{\text{th}}$ player if actions $\boldsymbol{a}$ are taken at state $s$ at step $h$.

In this section, we consider three versions of equlibrium for general-sum MGs: Nash equilibrium (NE), correlated equilibrium (CE), and coarse correlated equilibrium (CCE), all being standard solution notions in games (Nisan et al., 2007). These three notions coincide on two-player zero-sum games, but are not equivalent to each other on multi-player general-sum games; any one of them could be desired depending on the application at hand. Below we introduce their definitions.

**(Approximate) Nash equilibrium in general-sum MG.**   The policy of the $i^{\text{th}}$ player is denoted as $\pi_i := \{\pi_{h,i} : \mathcal{S} \to \Delta_{\mathcal{A}_i}\}_{h \in [H]}$. We denote the product policy of all the players as $\pi := \pi_1 \times \cdots \times \pi_M$, and denote the policy of the all the players except the $i^{\text{th}}$ player as $\pi_{-i}$. We define $V_{h,i}^\pi(s)$ as the expected cumulative reward that will be received by the $i^{\text{th}}$ player if starting at state $s$ at step $h$ and all players follow policy $\pi$. For any strategy $\pi_{-i}$, there also exists a *best response* of the $i^{\text{th}}$ player, which is a policy $\mu^\dagger(\pi_{-i})$ satisfying $V_{h,i}^{\mu^\dagger(\pi_{-i}),\pi_{-i}}(s) = \sup_{\pi_i} V_{h,i}^{\pi_i,\pi_{-i}}(s)$ for any $(s, h) \in \mathcal{S} \times [H]$. We denote $V_{h,i}^{\dagger,\pi_{-i}} := V_{h,i}^{\mu^\dagger(\pi_{-i}),\pi_{-i}}$. The Q-functions of the best response can be defined similarly.

Our first objective is to find an approximate Nash equilibrium of Markov games.

**Definition 7** ($\epsilon$-approximate Nash equilibrium in general-sum MG). A product policy $\pi$ is an $\epsilon$-**approximate Nash equilibrium** if $\max_{i \in [m]} (V_{1,i}^{\dagger,\pi_{-i}} - V_{1,i}^\pi)(s_1) \le \epsilon$.

The above definition requires the suboptimality gap $(V_{1,i}^{\dagger,\pi_{-i}} - V_{1,i}^\pi)(s_1)$ to be less than $\epsilon$ for all player $i$. This is consistent with the two-player case (Definition 1) up to a constant of 2, since in the two-player zero-sum setting, we have $V_{1,1}^\pi(s_1) = -V_{1,2}^\pi(s_1)$ for any product policy $\pi = (\mu, \nu)$, and therefore $(V_{1,1}^{\dagger,\nu} - V_{1,1}^{\mu,\dagger})(s_1) \le 2\max_{i \in [2]} (V_{1,i}^{\dagger,\pi_{-i}} - V_{1,i}^\pi)(s_1) \le 2(V_{1,1}^{\dagger,\nu} - V_{1,1}^{\mu,\dagger})(s_1)$. We can similarly define the regret.

**Definition 8** (Nash-regret in general-sum MG). Let $\pi^k$ denote the (product) policy deployed by the algorithm in the $k^{\text{th}}$ episode. After a total of $K$ episodes, the regret is defined as

$$\text{Regret}_{\text{Nash}}(K) = \sum_{k=1}^K \max_{i \in [m]} (V_{1,i}^{\dagger,\pi_{-i}^k} - V_{1,i}^{\pi^k})(s_1).$$

**(Approximate) CCE in general-sum MG.**   The coarse correlated equilibrium (CCE) is a relaxed version of Nash equilibrium in which we consider general correlated policies instead of product policies. Let $\mathcal{A} = \mathcal{A}_1 \times \cdots \times \mathcal{A}_m$ denote the joint action space.

**Definition 9** (CCE in in general-sum MG). A (correlated) policy $\pi := \{\pi_h(s) \in \Delta_{\mathcal{A}} : (h, s) \in [H] \times \mathcal{S}\}$ is a **CCE** if $\max_{i \in [m]} V_{h,i}^{\dagger,\pi_{-i}}(s) \le V_{h,i}^\pi(s)$ for all $(s, h) \in \mathcal{S} \times [H]$.

Compared with a Nash equilibrium, a CEE is not necessarily a product policy, that is, we may not have $\pi_h(s) \in \Delta_{\mathcal{A}_1} \times \cdots \times \Delta_{\mathcal{A}_m}$. Similarly, we also define $\epsilon$-approximate CCE and CCE-regret below.

**Definition 10** ($\epsilon$-approximate CCE in general-sum MG). A policy $\pi := \{\pi_h(s) \in \Delta_{\mathcal{A}} : (h, s) \in [H] \times \mathcal{S}\}$ is an $\epsilon$**-approximate CCE** if $\max_{i \in [m]} (V_{1,i}^{\dagger, \pi_{-i}} - V_{1,i}^{\pi})(s_1) \leq \epsilon$.

**Definition 11** (CCE-regret in general-sum MG). Let policy $\pi^k$ denote the (correlated) policy deployed by the algorithm in the $k^{\text{th}}$ episode. After a total of $K$ episodes, the regret is defined as

$$\text{Regret}_{\text{CCE}}(K) = \sum_{k=1}^{K} \max_{i \in [m]} (V_{1,i}^{\dagger, \pi_{-i}^k} - V_{1,i}^{\pi^k})(s_1).$$

**(Approximate) CE in general-sum MG.** The correlated equilibrium (CE) is another relaxation of the Nash equilibrium. To define CE, we first introduce the concept of strategy modification: A strategy modification $\phi := \{\phi_{h,s}(a) \in \mathcal{A}_i : (h, s, a) \in [H] \times \mathcal{S} \times \mathcal{A}_i\}$ for player $i$ is a set of $S \times H$ injective functions from $\mathcal{A}_i$ to itself. Let $\Phi_i$ denote the set of all possible strategy modifications for player $i$.

One can compose a strategy modification $\phi$ with any Markov policy $\pi$ and obtain a new policy $\phi \diamond \pi$ such that when policy $\pi$ chooses to play $\boldsymbol{a} := (a_1, \ldots, a_m)$ at state $s$ and step $h$, policy $\phi \diamond \pi$ will play $(a_1, \ldots, a_{i-1}, \phi_{h,s}(a_i), a_{i+1}, \ldots, a_m)$ instead.

**Definition 12** (CE in general-sum MG). A policy $\pi := \{\pi_h(s) \in \Delta_{\mathcal{A}} : (h, s) \in [H] \times \mathcal{S}\}$ is a **CE** if $\max_{i \in [m]} \max_{\phi \in \Phi_i} V_{h,i}^{\phi \diamond \pi}(s) \leq V_{h,i}^{\pi}(s)$ holds for all $(s, h) \in \mathcal{S} \times [H]$.

Similarly, we have an approximate version of CE and CE-regret.

**Definition 13** ($\epsilon$-approximate CE in Markov games). A policy $\pi := \{\pi_h(s) \in \Delta_{\mathcal{A}} : (h, s) \in [H] \times \mathcal{S}\}$ is an $\epsilon$**-approximate CE** if $\max_{i \in [m]} \max_{\phi \in \Phi_i} (V_{1,i}^{\phi \diamond \pi} - V_{1,i}^{\pi})(s_1) \leq \epsilon$.

**Definition 14** (CE-regret in multiplayer Markov games). Let product policy $\pi^k$ denote the policy deployed by the algorithm in the $k^{\text{th}}$ episode. After a total of $K$ episodes, the regret is defined as

$$\text{Regret}_{\text{CE}}(K) = \sum_{k=1}^{K} \max_{i \in [m]} \max_{\phi} (V_{1,i}^{\phi \diamond \pi^k} - V_{1,i}^{\pi^k})(s_1).$$

**Relationship between Nash, CE, and CCE** For general-sum MGs, we have $\{\text{Nash}\} \subseteq \{\text{CE}\} \subseteq \{\text{CCE}\}$, so that they form a nested set of notions of equilibria (Nisan et al., 2007). Indeed, one can easily verify that if we restrict the choice of strategy modification $\phi$ to those consisting of only constant functions, i.e., $\phi_{h,s}(a)$ being independent of $a$, Definition 12 will reduce to the definition of CCE policy. In addition, any Nash equilibrium is a CE by definition. Finally, since a Nash equilibrium always exists, so does CE and CCE.

## C.2 MULTIPLAYER OPTIMISTIC NASH VALUE ITERATION

Here we present the Multi-Nash-VI algorithm, which is an extension of Algorithm 1 for multi-player general-sum Markov games.

**The EQUILIBRIUM Subroutine.** Our EQUILIBRIUM subroutine in Line 11 could be taken from either one of the $\{\text{NASH}, \text{CE}, \text{CCE}\}$ subroutines for *one-step* games. When using NASH, we compute the Nash equilibrium of a one-step multi-player game (see, e.g., Berg & Sandholm (2016) for an overview of the available algorithms); the worst-case computational complexity of such a subroutine will be PPAD-hard (Daskalakis, 2013). When using CE or CCE, we find CEs or CCEs of the one-step games respectively, which can be solved in polynomial time using linear programming. However, the policies found are not guaranteed to be a product policy. We remark that in Algorithm 1 we used the CCE subroutine for finding Nash in two-player zero-sum games, which seemingly contrasts the principle of using the right subroutine for finding the right equilibrium, but nevertheless works as the Nash equilibrium and CCE are equivalent in zero-sum games.

Now we are ready to present the theoretical guarantees for Algorithm 3. We let $\pi^k$ denote the policy computed in line 11 of Algorithm 3 in the $k^{\text{th}}$ episode.

**Theorem 15** (Multi-Nash-VI). *There exists an absolute constant $c$, for any $p \in (0, 1]$, let $\iota = \log(SABT/p)$, then with probability at least $1 - p$, Algorithm 3 with bonus $\beta_t = c\sqrt{SH^2\iota/t}$ and EQUILIBRIUM being one of $\{\text{NASH}, \text{CE}, \text{CCE}\}$ satisfies (repsectively):*

---

**Algorithm 3** Multiplayer Optimistic Nash Value Iteration (Multi-Nash-VI)

---

1: **Initialize:** for any $(s, \boldsymbol{a}, h, i), \overline{Q}_{h,i}(s, \boldsymbol{a}) \leftarrow H, \underline{Q}_{h,i}(s, \boldsymbol{a}) \leftarrow 0, \Delta \leftarrow H, N_h(s, \boldsymbol{a}) \leftarrow 0.$
2: **for** episode $k = 1, \ldots, K$ **do**
3:    **for** step $h = H, H - 1, \ldots, 1$ **do**
4:       **for** $(s, \boldsymbol{a}) \in \mathcal{S} \times \mathcal{A}_1 \times \cdots \times \mathcal{A}_m$ **do**
5:          $t \leftarrow N_h(s, \boldsymbol{a})$;
6:          **if** $t > 0$ **then**
7:             **for** player $i = 1, 2, \ldots, m$ **do**
8:                $\overline{Q}_{h,i}(s, \boldsymbol{a}) \leftarrow \min\{(r_{h,i} + \widehat{\mathbb{P}}_h \overline{V}_{h+1,i})(s, \boldsymbol{a}) + \beta_t, H\}.$
9:                $\underline{Q}_{h,i}(s, \boldsymbol{a}) \leftarrow \max\{(r_{h,i} + \widehat{\mathbb{P}}_h \underline{V}_{h+1,i})(s, \boldsymbol{a}) - \beta_t, 0\}.$
10:       **for** $s \in \mathcal{S}$ **do**
11:          $\pi_h(\cdot|s) \leftarrow \text{EQUILIBRIUM}(\overline{Q}_{h,1}(s, \cdot), \overline{Q}_{h,2}(s, \cdot), \cdots, \overline{Q}_{h,M}(s, \cdot)).$
12:          **for** player $i = 1, 2, \ldots, m$ **do**
13:             $\overline{V}_{h,i}(s) \leftarrow (\mathbb{D}_{\pi_h} \overline{Q}_{h,i})(s); \quad \underline{V}_{h,i}(s) \leftarrow (\mathbb{D}_{\pi_h} \underline{Q}_{h,i})(s).$
14:    **if** $\max_{i \in [m]}(\overline{V}_{1,i} - \underline{V}_{1,i})(s_1) < \Delta$ **then**
15:       $\Delta \leftarrow \max_{i \in [m]}(\overline{V}_{1,i} - \underline{V}_{1,i})(s_1)$ and $\pi^{\text{out}} \leftarrow \pi.$
16:    **for** step $h = 1, \ldots, H$ **do**
17:       take action $\boldsymbol{a}_h \sim \pi_h(\cdot|s_h)$, observe reward $r_h$ and next state $s_{h+1}$.
18:       add 1 to $N_h(s_h, \boldsymbol{a}_h)$ and $N_h(s_h, \boldsymbol{a}_h, s_{h+1})$.
19:    $\widehat{\mathbb{P}}_h(\cdot|s_h, \boldsymbol{a}_h) \leftarrow N_h(s_h, \boldsymbol{a}_h, \cdot)/N_h(s_h, \boldsymbol{a}_h).$
20: **Output** $\pi^{\text{out}}$.

---

- $\pi^{out}$ *is an $\epsilon$-approximate* {NASH,CE,CCE}, *if the number of episodes* $K \geq \Omega(H^4 S^2 (\prod_{i=1}^m A_i)\iota/\epsilon^2).$

- $\text{Regret}_{\{\text{Nash,CE,CCE}\}}(K) \leq \mathcal{O}(\sqrt{H^3 S^2 (\prod_{i=1}^m A_i)T\iota}).$

In the situation where the EQUILIBRIUM subroutine is taken as NASH, Theorem 15 provides the sample complexity bound of Multi-Nash-VI algorithm to find a $\epsilon$-approximate Nash equilibrium and its regret bound. Compared with our earlier result in two-player zero-sum games (Theorem 3), here the sample complexity scales as $S^2 H^4$ instead of $SH^3$. This is because the auxiliary bonus and Bernstein concentration technique do not apply here. Furthermore, the sample complexity is proportional to $\prod_{i=1}^m A_i$, which increases exponentially as the number of players increases.

**Runtime of Algorithm 3**   We remark that while the Nash guarantee is the strongest among the three guarantees presented in Theorem 15, the runtime of Algorithm 3 in the Nash case is not guaranteed to be polynomial and in the worst case PPAD-hard (due to the hardness of the NASH subroutine). In contrast, the CE and CCE guarantees are weaker, but the corresponding algorithms are guaranteed to finish in polynomial time.

### C.3   MULTIPLAYER REWARD-FREE LEARNING

We can also generalize VI-Zero to the multiplayer setting and obtain Algorithm 4, Multi-VI-Zero, which is almost the same as VI-Zero except that its exploration bonus $\beta_t$ is larger than that of VI-Zero by a $\sqrt{S}$ factor.

Similar to Theorem 5, we have the following theoretical guarantee claiming that any {NASH,CCE,CE} of the $\mathcal{M}(\widehat{\mathbb{P}}, \widehat{r}^i)$ ($i \in [N]$) is also an approximate {NASH,CCE,CE} of the true Markov game $\mathcal{M}(\mathbb{P}, r^i)$, where $\widehat{\mathbb{P}}^{\text{out}}$ is the empirical transition outputted by Algorithm 4 and $\widehat{r}^i$ is the empirical estimate of $r^i$.

**Theorem 16** (Multi-VI-Zero). *There exists an absolute constant $c$, for any $p \in (0, 1]$, $\epsilon \in (0, H]$, $N \in \mathbb{N}$, if we choose bonus $\beta_t = c\sqrt{H^2 S\iota/t}$ with $\iota = \log(NSABT/p)$ and $K \geq c(H^4 S^2 (\prod_{i=1}^m A_i)\iota/\epsilon^2)$, then with probability at least $1 - p$, the output $\widehat{\mathbb{P}}^{out}$ of Algorithm 4 has the following property: for any $N$ fixed reward functions $r^1, \ldots, r^N$, any {NASH,CCE,CE} of*

---

**Algorithm 4** Multiplayer Optimistic Value Iteration with Zero Reward (Multi-VI-Zero)

---

1: **Initialize:** for any $(s, \boldsymbol{a}, h)$, $\widetilde{V}_h(s, \boldsymbol{a}) \leftarrow H$, $\Delta \leftarrow H$, $N_h(s, \boldsymbol{a}) \leftarrow 0$.
2: **for** episode $k = 1, \ldots, K$ **do**
3:     **for** step $h = H, H-1, \ldots, 1$ **do**
4:         **for** $(s, \boldsymbol{a}) \in \mathcal{S} \times \mathcal{A}_1 \times \cdots \times \mathcal{A}_m$ **do**
5:             $t \leftarrow N_h(s, \boldsymbol{a})$.
6:             **if** $t > 0$ **then**
7:                 $\widetilde{Q}_h(s, \boldsymbol{a}) \leftarrow \min\{(\widehat{\mathbb{P}}_h \widetilde{V}_{h+1})(s, \boldsymbol{a}) + \beta_t, H\}$.
8:         **for** $s \in \mathcal{S}$ **do**
9:             $\pi_h(s) \leftarrow \arg\max_{\boldsymbol{a} \in \mathcal{A}_1 \times \cdots \times \mathcal{A}_m} \widetilde{Q}_h(s, \boldsymbol{a})$.
10:            $\widetilde{V}_h(s) \leftarrow (\mathbb{D}_{\pi_h} \widetilde{Q}_h)(s)$.
11:     **if** $\widetilde{V}_1(s_1) < \Delta$ **then**
12:         $\Delta \leftarrow \widetilde{V}_1(s_1)$ and $\widehat{\mathbb{P}}^{\text{out}} \leftarrow \widehat{\mathbb{P}}$.
13:     **for** step $h = 1, \ldots, H$ **do**
14:         take action $\boldsymbol{a}_h \sim \pi_h(\cdot, \cdot | s_h)$, observe next state $s_{h+1}$.
15:         add 1 to $N_h(s_h, \boldsymbol{a}_h)$ and $N_h(s_h, \boldsymbol{a}_h, s_{h+1})$.
16:         $\widehat{\mathbb{P}}_h(\cdot | s_h, \boldsymbol{a}_h) \leftarrow N_h(s_h, \boldsymbol{a}_h, \cdot)/N_h(s_h, \boldsymbol{a}_h)$.
17: **Output** $\widehat{\mathbb{P}}^{\text{out}}$.

---

*Markov game $\mathcal{M}(\widehat{\mathbb{P}}^{out}, \widehat{r}^i)$ is also an $\epsilon$-approximate* {NASH,CCE,CE} *of the true Markov game* $\mathcal{M}(\mathbb{P}, r^i)$ *for all $i \in [N]$.*

The proof of Theorem 16 can be found in Appendix H.2. It is worth mentioning that the empirical Markov game $\mathcal{M}(\widehat{\mathbb{P}}^{out}, \widehat{r}^i)$ may have multiple {Nash equilibria,CCEs,CEs} and Theorem 16 ensures that all of them are $\epsilon$-approximate {Nash equilibria,CCEs,CEs} of the true Markov game. Also, note that the sample complexity here is quadratic in the number of states because we are using the exploration bonus $\beta_t = \sqrt{H^2 S \iota / t}$ that is larger than usual by a $\sqrt{S}$ factor.

## D   BELLMAN EQUATIONS FOR MARKOV GAMES

In this section, we present the Bellman equations for different types of values in Markov games.

**Fixed policies.** For any pair of Markov policy $(\mu, \nu)$, by definition of their values in (1) (2), we have the following Bellman equations:

$$Q_h^{\mu,\nu}(s, a, b) = (r_h + \mathbb{P}_h V_{h+1}^{\mu,\nu})(s, a, b), \qquad V_h^{\mu,\nu}(s) = (\mathbb{D}_{\mu_h \times \nu_h} Q_h^{\mu,\nu})(s)$$

for all $(s, a, b, h) \in \mathcal{S} \times \mathcal{A} \times \mathcal{B} \times [H]$, where $V_{H+1}^{\mu,\nu}(s) = 0$ for all $s \in \mathcal{S}$.

**Best responses.** For any Markov policy $\mu$ of the max-player, by definition, we have the following Bellman equations for values of its best response:

$$Q_h^{\mu,\dagger}(s, a, b) = (r_h + \mathbb{P}_h V_{h+1}^{\mu,\dagger})(s, a, b), \qquad V_h^{\mu,\dagger}(s) = \inf_{\nu \in \Delta_\mathcal{B}} (\mathbb{D}_{\mu_h \times \nu} Q_h^{\mu,\dagger})(s),$$

for all $(s, a, b, h) \in \mathcal{S} \times \mathcal{A} \times \mathcal{B} \times [H]$, where $V_{H+1}^{\mu,\dagger}(s) = 0$ for all $s \in \mathcal{S}$.

Similarly, for any Markov policy $\nu$ of the min-player, we also have the following symmetric version of Bellman equations for values of its best response:

$$Q_h^{\dagger,\nu}(s, a, b) = (r_h + \mathbb{P}_h V_{h+1}^{\dagger,\nu})(s, a, b), \qquad V_h^{\dagger,\nu}(s) = \sup_{\mu \in \Delta_\mathcal{A}} (\mathbb{D}_{\mu \times \nu_h} Q_h^{\dagger,\nu})(s).$$

for all $(s, a, b, h) \in \mathcal{S} \times \mathcal{A} \times \mathcal{B} \times [H]$, where $V_{H+1}^{\dagger,\nu}(s) = 0$ for all $s \in \mathcal{S}$.

**Nash equilibria.** Finally, by definition of Nash equilibria in Markov games, we have the following Bellman optimality equations:

$$Q_h^\star(s, a, b) = (r_h + \mathbb{P}_h V_{h+1}^\star)(s, a, b)$$
$$V_h^\star(s) = \sup_{\mu \in \Delta_\mathcal{A}} \inf_{\nu \in \Delta_\mathcal{B}} (\mathbb{D}_{\mu \times \nu} Q_h^\star)(s) = \inf_{\nu \in \Delta_\mathcal{B}} \sup_{\mu \in \Delta_\mathcal{A}} (\mathbb{D}_{\mu \times \nu} Q_h^\star)(s).$$

for all $(s, a, b, h) \in \mathcal{S} \times \mathcal{A} \times \mathcal{B} \times [H]$, where $V_{H+1}^\star(s) = 0$ for all $s \in \mathcal{S}$.

## E  PROPERTIES OF COARSE CORRELATED EQUILIBRIUM

Recall the definition for CCE in our main paper (4), we restate it here after rescaling. For any pair of matrices $P, Q \in [0, 1]^{n \times m}$, the subroutine $\text{CCE}(P, Q)$ returns a distribution $\pi \in \Delta_{n \times m}$ that satisfies:

$$\mathbb{E}_{(a,b) \sim \pi} P(a, b) \geq \max_{a^\star} \mathbb{E}_{(a,b) \sim \pi} P(a^\star, b) \tag{5}$$
$$\mathbb{E}_{(a,b) \sim \pi} Q(a, b) \leq \min_{b^\star} \mathbb{E}_{(a,b) \sim \pi} Q(a, b^\star)$$

We make three remarks on CCE. First, a CCE always exists since a Nash equilibrium for a general-sum game with payoff matrices $(P, Q)$ is also a CCE defined by $(P, Q)$, and a Nash equilibrium always exists. Second, a CCE can be efficiently computed, since above constraints (5) for CCE can be rewritten as $n + m$ linear constraints on $\pi \in \Delta_{n \times m}$, which can be efficiently resolved by standard linear programming algorithm. Third, a CCE in general-sum games needs not to be a Nash equilibrium. However, a CCE in zero-sum games is guaranteed to be a Nash equalibrium.

**Proposition 17.** *Let $\pi = \text{CCE}(Q, Q)$, and $(\mu, \nu)$ be the marginal distribution over both players' actions induced by $\pi$. Then $(\mu, \nu)$ is a Nash equilibrium for payoff matrix $Q$.*

*Proof of Proposition 17.* Let $N^\star$ be the value of Nash equilibrium for $Q$. Since $\pi = \text{CCE}(Q, Q)$, by definition, we have:

$$\mathbb{E}_{(a,b) \sim \pi} Q(a, b) \geq \max_{a^\star} \mathbb{E}_{(a,b) \sim \pi} Q(a^\star, b) = \max_{a^\star} \mathbb{E}_{b \sim \nu} Q(a^\star, b) \geq N^\star$$
$$\mathbb{E}_{(a,b) \sim \pi} Q(a, b) \leq \min_{b^\star} \mathbb{E}_{(a,b) \sim \pi} Q(a, b^\star) = \min_{b^\star} \mathbb{E}_{a \sim \mu} Q(a, b^\star) \leq N^\star$$

This gives:

$$\max_{a^\star} \mathbb{E}_{b \sim \nu} Q(a^\star, b) = \min_{b^\star} \mathbb{E}_{a \sim \mu} Q(a, b^\star) = N^\star$$

which finishes the proof. $\qquad\square$

Intuitively, a CCE procedure can be used in Nash Q-learning for finding an approximate Nash equilibrium, because the values of upper confidence and lower confidence ($\overline{Q}$ and $\underline{Q}$) will be eventually very close, so that the preconditions of Proposition 17 becomes approximately satisfied.

## F  PROOF FOR SECTION 3 – OPTIMISTIC NASH VALUE ITERATION

### F.1  PROOF OF THEOREM 3

We denote $V^k$, $Q^k$, $\pi^k$, $\mu^k$ and $\nu^k$ [4] for values and policies at the *beginning* of the $k$-th episode. In particular, $N_h^k(s, a, b)$ is the number we have visited the state-action tuple $(s, a, b)$ at the $h$-th step before the $k$-th episode. $N_h^k(s, a, b, s')$ is defined by the same token. Using this notation, we can further define the empirical transition by $\widehat{\mathbb{P}}_h^k(s'|s, a, b) := N_h^k(s, a, b, s')/N_h^k(s, a, b)$. If $N_h^k(s, a, b) = 0$, we set $\widehat{\mathbb{P}}_h^k(s'|s, a, b) = 1/S$.

As a result, the bonus terms can be written as

$$\beta_h^k(s, a, b) := C \left( \sqrt{\frac{\iota H^2}{\max\{N_h^k(s, a, b), 1\}}} + \frac{H^2 S \iota}{\max\{N_h^k(s, a, b), 1\}} \right) \tag{6}$$

---

[4] recall that $(\mu_h^k, \nu_h^k)$ are the marginal distributions of $\pi_h^k$.

$$\gamma_h^k(s,a,b) := \frac{C}{H}\widehat{\mathbb{P}}_h(\overline{V}_{h+1}^k - \underline{V}_{h+1}^k)(s,a,b) \tag{7}$$

for some large absolute constant $C > 0$.

**Lemma 18.** *Let $c_1$ be some large absolute constant. Define event $E_0$ to be: for all $h, s, a, b, s'$ and $k \in [K]$,*

$$
\begin{cases}
|[(\widehat{\mathbb{P}}_h^k - \mathbb{P}_h)V_{h+1}^\star](s,a,b)| \le c_1 \sqrt{\dfrac{H^2\iota}{\max\{N_h^k(s,a,b),1\}}}, \\[2ex]
|(\widehat{\mathbb{P}}_h^k - \mathbb{P}_h)(s' \mid s,a,b)| \le c_1\left(\sqrt{\dfrac{\min\{\mathbb{P}_h(s' \mid s,a,b), \widehat{\mathbb{P}}_h^k(s' \mid s,a,b)\}\iota}{\max\{N_h^k(s,a,b),1\}}} + \dfrac{\iota}{\max\{N_h^k(s,a,b),1\}}\right).
\end{cases}
$$

*We have $\mathbb{P}(E_1) \ge 1 - p$.*

*Proof.* The proof is standard and folklore: apply standard concentration inequalities and then take a union bound. For completeness, we provide the proof of the second one here.

Consider a fixed $(s,a,b,h)$ tuple.

Let's consider the following equivalent random process: (a) before the agent starts, the environment samples $\{s^{(1)}, s^{(2)}, \dots, s^{(K)}\}$ independently from $\mathbb{P}_h(\cdot \mid s,a,b)$; (b) during the interaction between the agent and environment, the $i^{\text{th}}$ time the agent reaches $(s,a,b,h)$, the environment will make the agent transit to $s^{(i)}$. Note that the randomness induced by this interaction procedure is exactly the same as the original one, which means the probability of any event in this context is the same as in the original problem. Therefore, it suffices to prove the target concentration inequality in this 'easy' context. Denote by $\widehat{\mathbb{P}}_h^{(t)}(\cdot \mid s,a,b)$ the empirical estimate of $\mathbb{P}_h(\cdot \mid s,a,b)$ calculated using $\{s^{(1)}, s^{(2)}, \dots, s^{(t)}\}$. For a fixed $t$ and $s'$, by applying the Bernstein inequality and its empirical version, we have with probability at least $1 - p/S^2ABT$,

$$|(\mathbb{P}_h - \widehat{\mathbb{P}}_h^{(t)})(s' \mid s,a,b)| \le \mathcal{O}\left(\sqrt{\frac{\min\{\mathbb{P}_h(s' \mid s,a,b), \widehat{\mathbb{P}}_h^{(t)}(s' \mid s,a,b)\}\iota}{t}} + \frac{\iota}{t}\right).$$

Now we can take a union bound over all $s, a, b, h, s'$ and $t \in [K]$, and obtain that with probability at least $1 - p$, for all $s, a, b, h, s'$ and $t \in [K]$,

$$|(\mathbb{P}_h - \widehat{\mathbb{P}}_h^{(t)})(s' \mid s,a,b)| \le \mathcal{O}\left(\sqrt{\frac{\min\{\mathbb{P}_h(s' \mid s,a,b), \widehat{\mathbb{P}}_h^{(t)}(s' \mid s,a,b)\}\iota}{t}} + \frac{\iota}{t}\right).$$

Note that the agent can reach each $(s,a,b,h)$ for at most $K$ times, this directly implies that the third inequality also holds with probability at least $1 - p$. $\square$

We begin with an auxiliary lemma bounding the lower-order term.

**Lemma 19.** *Suppose event $E_0$ holds, then there exists absolute constant $c_2$ such that: if function $g(s)$ satisfies $|g|(s) \le (\overline{V}_{h+1}^k - \underline{V}_{h+1}^k)(s)$ for all $s$, then*

$$
\begin{aligned}
&|(\widehat{\mathbb{P}}_h^k - \mathbb{P}_h)g(s,a,b)| \\
&\le c_2\left(\frac{1}{H}\min\{\widehat{\mathbb{P}}_h^k(\overline{V}_{h+1}^k - \underline{V}_{h+1}^k)(s,a,b), \mathbb{P}_h(\overline{V}_{h+1}^k - \underline{V}_{h+1}^k)(s,a,b)\} + \frac{H^2S\iota}{\max\{N_h^k(s,a,b),1\}}\right).
\end{aligned}
$$

*Proof.* By triangle inequality,

$$|(\widehat{\mathbb{P}}_h^k - \mathbb{P}_h)g(s, a, b)|$$

$$\leq \sum_{s'} |(\widehat{\mathbb{P}}_h^k - \mathbb{P}_h)(s'|s, a, b)||g|(s')$$

$$\leq \sum_{s'} |(\widehat{\mathbb{P}}_h^k - \mathbb{P}_h)(s'|s, a, b)|(\overline{V}_{h+1}^k - \underline{V}_{h+1}^k)(s')$$

$$\overset{(i)}{\leq} \mathcal{O}\left(\sum_{s'}(\sqrt{\frac{\iota\widehat{\mathbb{P}}_h^k(s'|s, a, b)}{\max\{N_h^k(s, a, b), 1\}}} + \frac{\iota}{\max\{N_h^k(s, a, b), 1\}})(\overline{V}_{h+1}^k - \underline{V}_{h+1}^k)(s')\right)$$

$$\overset{(ii)}{\leq} \mathcal{O}\left(\sum_{s'}(\frac{\widehat{\mathbb{P}}_h^k(s'|s, a, b)}{H} + \frac{H\iota}{\max\{N_h^k(s, a, b), 1\}})(\overline{V}_{h+1}^k - \underline{V}_{h+1}^k)(s')\right)$$

$$\leq \mathcal{O}\left(\frac{\widehat{\mathbb{P}}_h^k(\overline{V}_{h+1}^k - \underline{V}_{h+1}^k)(s, a, b)}{H} + \frac{H^2 S\iota}{\max\{N_h^k(s, a, b), 1\}}\right),$$

where $(i)$ is by the second inequality in event $E_0$ and $(ii)$ is by AM-GM inequality. This proves the empirical version. Similarly, we can show

$$|(\widehat{\mathbb{P}}_h^k - \mathbb{P}_h)g(s, a, b)| \leq \mathcal{O}\left(\frac{\mathbb{P}_h(\overline{V}_{h+1}^k - \underline{V}_{h+1}^k)(s, a, b)}{H} + \frac{H^2 S\iota}{\max\{N_h^k(s, a, b), 1\}}\right),$$

Combining the two bounds completes the proof. □

Now we can prove the upper and lower bounds are indeed upper and lower bounds of the best reponses.

**Lemma 20.** *Suppose event $E_0$ holds. Then for all $h, s, a, b$ and $k \in [K]$, we have*

$$\begin{cases} \overline{Q}_h^k(s, a, b) \geq Q_h^{\dagger, \nu^k}(s, a, b) \geq Q_h^{\mu^k, \dagger}(s, a, b) \geq \underline{Q}_h^k(s, a, b), \\ \overline{V}_h^k(s) \geq V_h^{\dagger, \nu^k}(s) \geq V_h^{\mu^k, \dagger}(s) \geq \underline{V}_h^k(s). \end{cases} \quad (8)$$

*Proof.* The proof is by backward induction. Suppose the bounds hold for the $Q$-values in the $(h + 1)^{\text{th}}$ step, we now establish the bounds for the $V$-values in the $(h + 1)^{\text{th}}$ step and $Q$-values in the $h^{\text{th}}$-step. For any state $s$:

$$\overline{V}_{h+1}^k(s) = \mathbb{D}_{\pi_{h+1}^k} \overline{Q}_{h+1}^k(s)$$

$$\geq \max_\mu \mathbb{D}_{\mu \times \nu_{h+1}^k} \overline{Q}_{h+1}^k(s) \quad (9)$$

$$\geq \max_\mu \mathbb{D}_{\mu \times \nu_{h+1}^k} Q_{h+1}^{\dagger, \nu^k}(s) = V_{h+1}^{\dagger, \nu^k}(s).$$

Similarly, we can show $\underline{V}_{h+1}^k(s) \leq V_{h+1}^{\mu^k, \dagger}(s)$. Therefore, we have: for all $s$,

$$\overline{V}_{h+1}^k(s) \geq V_{h+1}^{\dagger, \nu^k}(s) \geq V_{h+1}^\star(s) \geq V_{h+1}^{\mu^k, \dagger}(s) \geq \underline{V}_{h+1}^k(s).$$

Now consider an arbitrary triple $(s, a, b)$ in the $h^{\text{th}}$ step. We have

$$(\overline{Q}_h^k - Q_h^{\dagger, \nu^k})(s, a, b)$$

$$\geq \min\left\{(\widehat{\mathbb{P}}_h^k \overline{V}_{h+1}^k - \mathbb{P}_h V_{h+1}^{\dagger, \nu^k} + \beta_h^k + \gamma_h^k)(s, a, b), 0\right\}$$

$$\geq \min\left\{(\widehat{\mathbb{P}}_h^k V_{h+1}^{\dagger, \nu^k} - \mathbb{P}_h V_{h+1}^{\dagger, \nu^k} + \beta_h^k + \gamma_h^k)(s, a, b), 0\right\}$$

$$= \min\left\{\underbrace{(\widehat{\mathbb{P}}_h^k - \mathbb{P}_h)(V_{h+1}^{\dagger, \nu^k} - V_{h+1}^\star)(s, a, b)}_{(A)} + \underbrace{(\widehat{\mathbb{P}}_h^k - \mathbb{P}_h)V_{h+1}^\star(s, a, b)}_{(B)}\right.$$

$$\left. + (\beta_h^k + \gamma_h^k)(s, a, b), 0\right\}. \quad (10)$$

Invoking Lemma 19 with $g = V_{h+1}^{\dagger,\nu^k} - V_{h+1}^\star$,

$$|(A)| \leq \mathcal{O}\left(\frac{\widehat{\mathbb{P}}_h^k(\overline{V}_{h+1}^k - \underline{V}_{h+1}^k)(s,a,b)}{H} + \frac{H^2 S\iota}{\max\{N_h^k(s,a,b),1\}}\right).$$

By the first inequality in event $E_0$,

$$|(B)| \leq \mathcal{O}\left(\sqrt{\frac{H^2\iota}{\max\{N_h^k(s,a,b),1\}}}\right).$$

Plugging the two inequalities above back into (10) and recalling the definition of $\beta_h^k$ and $\gamma_h^k$, we obtain $\overline{Q}_h^k(s,a,b) \geq Q_h^{\dagger,\nu^k}(s,a,b)$. Similarly, we can show $\underline{Q}_h^k(s,a,b) \leq Q_h^{\mu^k,\dagger}(s,a,b)$. $\square$

Finally we come to the proof of Theorem 3.

*Proof of Theorem 3.* Suppose event $E_0$ holds. We first upper bound the regret. By Lemma 20, the regret can be upper bounded by

$$\sum_k (V_1^{\dagger,\nu^k}(s_1^k) - V_1^{\mu^k,\dagger}(s_1^k)) \leq \sum_k (\overline{V}_1^k(s_1^k) - \underline{V}_1^k(s_1^k)).$$

For brevity's sake, we define the following notations:

$$\begin{cases} \Delta_h^k := (\overline{V}_h^k - \underline{V}_h^k)(s_h^k), \\ \zeta_h^k := \Delta_h^k - (\overline{Q}_h^k - \underline{Q}_h^k)(s_h^k, a_h^k, b_h^k), \\ \xi_h^k := \mathbb{P}_h(\overline{V}_{h+1}^k - \underline{V}_{h+1}^k)(s_h^k, a_h^k, b_h^k) - \Delta_{h+1}^k. \end{cases} \tag{11}$$

Let $\mathcal{F}_h^k$ be the $\sigma$-field generated by the following random variables:

$$\{(s_i^j, a_i^j, b_i^j, r_i^j)\}_{(i,j) \in [H] \times [k-1]} \bigcup \{(s_i^k, a_i^k, b_i^k, r_i^k)\}_{i \in [h-1]} \bigcup \{s_h^k\}.$$

It's easy to check $\zeta_h^k$ and $\xi_h^k$ are martingale differences with respect to $\mathcal{F}_h^k$. With a slight abuse of notation, we use $\beta_h^k$ to refer to $\beta_h^k(s_h^k, a_h^k, b_h^k)$ and $N_h^k$ to refer to $N_h^k(s_h^k, a_h^k, b_h^k)$ in the following proof.

We have

$$\begin{aligned}
\Delta_h^k =& \zeta_h^k + \left(\overline{Q}_h^k - \underline{Q}_h^k\right)(s_h^k, a_h^k, b_h^k) \\
\leq& \zeta_h^k + 2\beta_h^k + 2\gamma_h^k + \widehat{\mathbb{P}}_h^k(\overline{V}_{h+1}^k - \underline{V}_{h+1}^k)(s_h^k, a_h^k, b_h^k) \\
\overset{(i)}{\leq}& \zeta_h^k + 2\beta_h^k + 2\gamma_h^k + \mathbb{P}_h(\overline{V}_{h+1}^k - \underline{V}_{h+1}^k)(s_h^k, a_h^k, b_h^k) \\
& + c_2\left(\frac{\mathbb{P}_h(\overline{V}_{h+1}^k - \underline{V}_{h+1}^k)(s_h^k, a_h^k, b_h^k)}{H} + \frac{H^2 S\iota}{\max\{N_h^k, 1\}}\right) \\
\overset{(ii)}{\leq}& \zeta_h^k + 2\beta_h^k + \mathbb{P}_h(\overline{V}_{h+1}^k - \underline{V}_{h+1}^k)(s_h^k, a_h^k, b_h^k) \\
& + 2c_2 C\left(\frac{\mathbb{P}_h(\overline{V}_{h+1}^k - \underline{V}_{h+1}^k)(s_h^k, a_h^k, b_h^k)}{H} + \frac{H^2 S\iota}{\max\{N_h^k, 1\}}\right) \\
\leq& \zeta_h^k + \left(1 + \frac{2c_2 C}{H}\right)\mathbb{P}_h(\overline{V}_{h+1}^k - \underline{V}_{h+1}^k)(s_h^k, a_h^k, b_h^k) + 4c_2 C\left(\sqrt{\frac{\iota H^2}{\max\{N_h^k, 1\}}} + \frac{H^2 S\iota}{\max\{N_h^k, 1\}}\right) \\
=& \zeta_h^k + \left(1 + \frac{2c_2 C}{H}\right)\xi_h^k + \left(1 + \frac{2c_2 C}{H}\right)\Delta_{h+1}^k + 4c_2 C\left(\sqrt{\frac{\iota H^2}{\max\{N_h^k, 1\}}} + \frac{H^2 S\iota}{\max\{N_h^k, 1\}}\right)
\end{aligned}$$

where $(i)$ and $(ii)$ follow from Lemma 19.

Define $c_3 := 1 + 2c_2 C$ and $\kappa := 1 + c_3/H$. Recursing this argument for $h \in [H]$ and summing over $k$,

$$\sum_{k=1}^{K} \Delta_1^k \le \sum_{k=1}^{K} \sum_{h=1}^{H} \left[ \kappa^{h-1} \zeta_h^k + \kappa^h \xi_h^k + \mathcal{O} \left( \sqrt{\frac{\iota H^2}{\max\{N_h^k, 1\}}} + \frac{H^2 S \iota}{\max\{N_h^k, 1\}} \right) \right].$$

By Azuma-Hoeffding inequality, with probability at least $1 - p$,

$$\begin{cases} \displaystyle\sum_{k=1}^{K} \sum_{h=1}^{H} \kappa^{h-1} \zeta_h^k \le \mathcal{O}\left(H\sqrt{HK\iota}\right) = \mathcal{O}\left(\sqrt{H^2 T \iota}\right), \\ \displaystyle\sum_{k=1}^{K} \sum_{h=1}^{H} \kappa^h \xi_h^k \le \mathcal{O}\left(H\sqrt{HK\iota}\right) = \mathcal{O}\left(\sqrt{H^2 T \iota}\right). \end{cases} \tag{12}$$

By pigeon-hole argument,

$$\sum_{k=1}^{K} \sum_{h=1}^{H} \frac{1}{\sqrt{\max\{N_h^k, 1\}}} \le \sum_{s,a,b,h:\ N_h^K(s,a,b) > 0} \sum_{n=1}^{N_h^K(s,a,b)} \frac{1}{\sqrt{n}} + HSAB$$

$$\le \mathcal{O}\left(\sqrt{HSABT} + HSAB\right),$$

$$\sum_{k=1}^{K} \sum_{h=1}^{H} \frac{1}{\max\{N_h^k, 1\}} \le \sum_{s,a,b,h:\ N_h^K(s,a,b) > 0} \sum_{n=1}^{N_h^K(s,a,b)} \frac{1}{n} + HSAB$$

$$\le \mathcal{O}(HSAB\iota).$$

Put everything together, with probability at least $1 - 2p$ (one $p$ comes from $\mathbb{P}(E_0) \ge 1 - p$ and the other is for equation (12)),

$$\sum_{k=1}^{K} (V_1^{\dagger, \nu^k}(s_1^k) - V_1^{\mu^k, \dagger}(s_1^k)) \le \mathcal{O}\left(\sqrt{H^3 SABT\iota} + H^3 S^2 AB\iota^2\right)$$

For the PAC guarantee, recall that we choose $\pi^{\text{out}} = \pi^{k^\star}$ such that $k^\star = \operatorname{argmin}_k \left(\overline{V}_1^k - \underline{V}_1^k\right)(s_1)$. As a result,

$$(V_1^{\dagger, \nu^{k^\star}} - V_1^{\mu^{k^\star}, \dagger})(s_1) \le (\overline{V}_1^{k^\star} - \underline{V}_1^{k^\star})(s_1) \le \frac{1}{K} \mathcal{O}\left(\sqrt{H^3 SABT\iota} + H^3 S^2 AB\iota^2\right),$$

which concludes the proof. $\qquad\square$

## F.2 Proof of Theorem 4

We use the same notation as in Appendix F.1 except the form of bonus. Besides, we define the empirical variance operator

$$\widehat{\mathbb{V}}_h^k V(s, a, b) := \operatorname{Var}_{s' \sim \widehat{\mathbb{P}}_h^k(\cdot | s, a, b)} V(s')$$

and the true (population) variance operator

$$\mathbb{V}_h V(s, a, b) := \operatorname{Var}_{s' \sim \mathbb{P}_h(\cdot | s, a, b)} V(s')$$

for any function $V \in \Delta^S$. If $N_h^k(s, a, b) = 0$, we simply set $\widehat{\mathbb{V}}_h^k V(s, a, b) := H^2$ regardless of the choice of $V$.

As a result, the bonus terms can be written as

$$\beta_h^k(s, a, b) := C\left( \sqrt{\frac{\iota \widehat{\mathbb{V}}_h^k[(\overline{V}_{h+1}^k + \underline{V}_{h+1}^k)/2](s, a, b)}{\max\{N_h^k(s, a, b), 1\}}} + \frac{H^2 S \iota}{\max\{N_h^k(s, a, b), 1\}} \right) \tag{13}$$

for some absolute constant $C > 0$.

**Lemma 21.** *Let $c_1$ be some large absolute constant. Define event $E_1$ to be: for all $h, s, a, b, s'$ and $k \in [K]$,*

$$
\begin{cases}
|[(\widehat{\mathbb{P}}_h^k - \mathbb{P}_h)V_{h+1}^{\star}](s, a, b)| \le c_1 \left( \sqrt{\dfrac{\widehat{\mathbb{V}}_h^k V_{h+1}^{\star}(s, a, b)\iota}{\max\{N_h^k(s, a, b), 1\}}} + \dfrac{H\iota}{\max\{N_h^k(s, a, b), 1\}} \right), \\[4ex]
|(\widehat{\mathbb{P}}_h^k - \mathbb{P}_h)(s' \mid s, a, b)| \le c_1 \left( \sqrt{\dfrac{\min\{\mathbb{P}_h(s' \mid s, a, b), \widehat{\mathbb{P}}_h^k(s' \mid s, a, b)\}\iota}{\max\{N_h^k(s, a, b), 1\}}} + \dfrac{\iota}{\max\{N_h^k(s, a, b), 1\}} \right), \\[4ex]
\|(\widehat{\mathbb{P}}_h^k - \mathbb{P}_h)(\cdot \mid s, a, b)\|_1 \le c_1 \sqrt{\dfrac{S\iota}{\max\{N_h^k(s, a, b), 1\}}}.
\end{cases}
$$

*We have $\mathbb{P}(E_1) \ge 1 - p$.*

The proof of Lemma 21 is highly similar to that of Lemma 18. Specifically, the first two can be proved by following basically the same argument in Lemma 18; the third one is standard (e.g., equation (12) in Azar et al. (2017)). We omit the proof here.

Since the proof of Lemma 19 does not depend on the form of the bonus, it can also be applied in this section. As in Appendix F.1, we will prove the upper and lower bounds are indeed upper and lower bounds of the best reponses.

**Lemma 22.** *Suppose event $E_1$ holds. Then for all $h, s, a, b$ and $k \in [K]$, we have*

$$
\begin{cases}
\overline{Q}_h^k(s, a, b) \ge Q_h^{\dagger, \nu^k}(s, a, b) \ge Q_h^{\mu^k, \dagger}(s, a, b) \ge \underline{Q}_h^k(s, a, b), \\[2ex]
\overline{V}_h^k(s) \ge V_h^{\dagger, \nu^k}(s) \ge V_h^{\mu^k, \dagger}(s) \ge \underline{V}_h^k(s).
\end{cases}
\tag{14}
$$

*Proof.* The proof is by backward induction and very similar to that of Lemma 20. Suppose the bounds hold for the $Q$-values in the $(h+1)^{\text{th}}$ step, we now establish the bounds for the $V$-values in the $(h+1)^{\text{th}}$ step and $Q$-values in the $h^{\text{th}}$-step.

The proof for the $V$-values is the same as (9).

For the $Q$-values, the decomposition (10) still holds and $(A)$ is bounded using Lemma 19 as before. The only difference is that we need to bound $(B)$ more carefully.

First, by the first inequality in event $E_1$,

$$
|(B)| \le \mathcal{O}\left( \sqrt{\frac{\widehat{\mathbb{V}}_h^k V_{h+1}^{\star}(s, a, b)\iota}{\max\{N_h^k(s, a, b), 1\}}} + \frac{H\iota}{\max\{N_h^k(s, a, b), 1\}} \right).
$$

By the relation of $V$-values in the $(h+1)^{\text{th}}$ step,

$$
\begin{aligned}
&|[\widehat{\mathbb{V}}_h^k(\overline{V}_{h+1}^k + \underline{V}_{h+1}^k)/2] - \widehat{\mathbb{V}}_h^k V_{h+1}^{\star}|(s, a, b) \\
&\le |[\widehat{\mathbb{P}}_h^k(\overline{V}_{h+1}^k + \underline{V}_{h+1}^k)/2]^2 - (\widehat{\mathbb{P}}_h^k V_{h+1}^{\star})^2|(s, a, b) \\
&\quad + |\widehat{\mathbb{P}}_h^k[(\overline{V}_{h+1}^k + \underline{V}_{h+1}^k)/2]^2 - \widehat{\mathbb{P}}_h^k(V_{h+1}^{\star})^2|(s, a, b) \\
&\le 4H\widehat{\mathbb{P}}_h^k|(\overline{V}_{h+1}^k + \underline{V}_{h+1}^k)/2 - V_{h+1}^{\star}|(s, a, b) \\
&\le 4H\widehat{\mathbb{P}}_h^k(\overline{V}_{h+1}^k - \underline{V}_{h+1}^k)(s, a, b),
\end{aligned}
\tag{15}
$$

which implies

$$
\sqrt{\frac{\iota \widehat{\mathbb{V}}_h^k V_{h+1}^\star(s,a,b)}{\max\{N_h^k(s,a,b),1\}}}
$$

$$
\leq \sqrt{\frac{\iota [\widehat{\mathbb{V}}_h^k[(\overline{V}_{h+1}^k + \underline{V}_{h+1}^k)/2] + 4H\widehat{\mathbb{P}}_h^k(\overline{V}_{h+1}^k - \underline{V}_{h+1}^k)](s,a,b)}{\max\{N_h^k(s,a,b),1\}}}
$$

$$
\leq \sqrt{\frac{\iota \widehat{\mathbb{V}}_h^k[(\overline{V}_{h+1}^k + \underline{V}_{h+1}^k)/2](s,a,b)}{\max\{N_h^k(s,a,b),1\}}} + \sqrt{\frac{4\iota H\widehat{\mathbb{P}}_h^k(\overline{V}_{h+1}^k - \underline{V}_{h+1}^k)](s,a,b)}{\max\{N_h^k(s,a,b),1\}}}
$$

$$
\overset{(i)}{\leq} \sqrt{\frac{\iota \widehat{\mathbb{V}}_h^k[(\overline{V}_{h+1}^k + \underline{V}_{h+1}^k)/2](s,a,b)}{\max\{N_h^k(s,a,b),1\}}} + \frac{\widehat{\mathbb{P}}_h^k(\overline{V}_{h+1}^k - \underline{V}_{h+1}^k)}{H} + \frac{4H^2\iota}{\max\{N_h^k(s,a,b),1\}},
$$

(16)

where $(i)$ is by AM-GM inequality.

Plugging the above inequalities back into (10) and recalling the definition of $\beta_h^k$ and $\gamma_h^k$ completes the proof. $\qquad\square$

We need one more lemma to control the error of the empirical variance estimator:

**Lemma 23.** *Suppose event $E_1$ holds. Then for all $h, s, a, b$ and $k \in [K]$, we have*

$$
|\widehat{\mathbb{V}}_h^k[(\overline{V}_{h+1}^k + \underline{V}_{h+1}^k)/2] - \mathbb{V}_h V_{h+1}^{\pi^k}|(s,a,b)
$$
$$
\leq 4H\mathbb{P}_h(\overline{V}_{h+1}^k - \underline{V}_{h+1}^k)(s,a,b) + \mathcal{O}\left(1 + \frac{H^4 S\iota}{\max\{N_h^k(s,a,b),1\}}\right).
$$

*Proof.* By Lemma 22, we have $\overline{V}_h^k(s) \geq V_h^{\pi^k}(s) \geq \underline{V}_h^k(s)$. As a result,

$$
|\widehat{\mathbb{V}}_h^k[(\overline{V}_{h+1}^k + \underline{V}_{h+1}^k)/2] - \mathbb{V}_h V_{h+1}^{\pi^k}|(s,a,b)
$$
$$
= |[\widehat{\mathbb{P}}_h^k(\overline{V}_{h+1}^k + \underline{V}_{h+1}^k)^2/4 - \mathbb{P}_h(V_{h+1}^{\pi^k})^2](s,a,b) - [(\widehat{\mathbb{P}}_h^k(\overline{V}_{h+1}^k + \underline{V}_{h+1}^k))^2/4 - (\mathbb{P}_h V_{h+1}^{\pi^k})^2](s,a,b)|
$$
$$
\leq [\widehat{\mathbb{P}}_h^k(\overline{V}_{h+1}^k)^2 - \mathbb{P}_h(\underline{V}_{h+1}^k)^2 - (\widehat{\mathbb{P}}_h^k \underline{V}_{h+1}^k)^2 + (\mathbb{P}_h \overline{V}_{h+1}^k)^2](s,a,b)
$$
$$
\leq [|(\widehat{\mathbb{P}}_h^k - \mathbb{P}_h)(\overline{V}_{h+1}^k)^2| + |\mathbb{P}_h[(\overline{V}_{h+1}^k)^2 - (\underline{V}_{h+1}^k)^2]|
$$
$$
\quad + |(\widehat{\mathbb{P}}_h^k \underline{V}_{h+1}^k)^2 - (\mathbb{P}_h \underline{V}_{h+1}^k)^2| + |(\mathbb{P}_h \underline{V}_{h+1}^k)^2 - (\mathbb{P}_h \overline{V}_{h+1}^k)^2|](s,a,b)
$$

These terms can be bounded separately by using event $E_1$:

$$
|(\widehat{\mathbb{P}}_h^k - \mathbb{P}_h)(\overline{V}_{h+1}^k)^2|(s,a,b) \leq H^2 \|(\widehat{\mathbb{P}}_h^k - \mathbb{P}_h)(\cdot \mid s,a,b)\|_1 \leq \mathcal{O}(H^2 \sqrt{\frac{S\iota}{\max\{N_h^k(s,a,b),1\}}}),
$$

$$
|\mathbb{P}_h[(\overline{V}_{h+1}^k)^2 - (\underline{V}_{h+1}^k)^2]|(s,a,b) \leq 2H[\mathbb{P}_h(\overline{V}_{h+1}^k - \underline{V}_{h+1}^k)](s,a,b),
$$

$$
|(\widehat{\mathbb{P}}_h^k \underline{V}_{h+1}^k)^2 - (\mathbb{P}_h \underline{V}_{h+1}^k)^2|(s,a,b) \leq 2H[(\widehat{\mathbb{P}}_h^k - \mathbb{P}_h)\underline{V}_{h+1}^k](s,a,b) \leq \mathcal{O}(H^2 \sqrt{\frac{S\iota}{\max\{N_h^k(s,a,b),1\}}}),
$$

$$
|(\mathbb{P}_h \underline{V}_{h+1}^k)^2 - (\mathbb{P}_h \overline{V}_{h+1}^k)^2|(s,a,b) \leq 2H[\mathbb{P}_h(\overline{V}_{h+1}^k - \underline{V}_{h+1}^k)](s,a,b).
$$

Combining with $H^2 \sqrt{\frac{S\iota}{\max\{N_h^k(s,a,b),1\}}} \leq 1 + \frac{H^4 S\iota}{\max\{N_h^k(s,a,b),1\}}$ completes the proof. $\qquad\square$

Finally we come to the proof of Theorem 4.

*Proof of Theorem 4.* Suppose event $E_1$ holds. We define $\Delta_h^k$, $\zeta_h^k$ abd $\xi_h^k$ as in the proof of Theorem 3. As before we have

$$
\begin{aligned}
\Delta_h^k \leq & \zeta_h^k + \left(1 + \frac{c_3}{H}\right) \mathbb{P}_h(\overline{V}_{h+1}^k - \underline{V}_{h+1}^k)\left(s_h^k, a_h^k, b_h^k\right) \\
& + 4c_2 C \left( \sqrt{\frac{\iota \widehat{\mathbb{V}}_h^k[(\overline{V}_{h+1}^k + \underline{V}_{h+1}^k)/2](s_h^k, a_h^k, b_h^k)}{\max\{N_h^k(s_h^k, a_h^k, b_h^k), 1\}}} + \frac{H^2 S \iota}{\max\{N_h^k(s_h^k, a_h^k, b_h^k), 1\}} \right).
\end{aligned}
\tag{17}
$$

By Lemma 23,

$$
\begin{aligned}
& \sqrt{\frac{\iota \widehat{\mathbb{V}}_h^k[(\overline{V}_{h+1}^k + \underline{V}_{h+1}^k)/2](s, a, b)}{\max\{N_h^k(s, a, b), 1\}}} \\
& \leq \mathcal{O}\left( \sqrt{\frac{\iota \mathbb{V}_h V_{h+1}^{\pi^k}(s, a, b) + \iota}{\max\{N_h^k(s, a, b), 1\}}} + \sqrt{\frac{H \iota \mathbb{P}_h(\overline{V}_{h+1}^k - \underline{V}_{h+1}^k)(s, a, b)}{\max\{N_h^k(s, a, b), 1\}}} + \frac{H^2 \sqrt{S} \iota}{\max\{N_h^k(s, a, b), 1\}} \right) \\
& \leq c_4 \left( \sqrt{\frac{\iota \mathbb{V}_h V_{h+1}^{\pi^k}(s, a, b) + \iota}{\max\{N_h^k(s, a, b), 1\}}} + \frac{\mathbb{P}_h(\overline{V}_{h+1}^k - \underline{V}_{h+1}^k)(s, a, b)}{H} + \frac{H^2 \sqrt{S} \iota}{\max\{N_h^k(s, a, b), 1\}} \right),
\end{aligned}
\tag{18}
$$

where $c_4$ is some absolute constant. Define $c_5 := 4c_2 c_4 C + c_3$ and $\kappa := 1 + c_5/H$. Plugging (18) back into (17), we have

$$
\begin{aligned}
\Delta_h^k \leq & \kappa \Delta_{h+1}^k + \kappa \xi_h^k + \zeta_h^k \\
& + \mathcal{O}\left( \sqrt{\frac{\iota \mathbb{V}_h V_{h+1}^{\pi^k}(s_h^k, a_h^k, b_h^k)}{N_h^k(s_h^k, a_h^k, b_h^k)}} + \sqrt{\frac{\iota}{N_h^k(s_h^k, a_h^k, b_h^k)}} + \frac{H^2 S \iota}{N_h^k(s_h^k, a_h^k, b_h^k)} \right).
\end{aligned}
\tag{19}
$$

Recursing this argument for $h \in [H]$ and summing over $k$,

$$
\begin{aligned}
\sum_{k=1}^K \Delta_1^k \leq \sum_{k=1}^K \sum_{h=1}^H \Bigg[ & \kappa^{h-1} \zeta_h^k + \kappa^h \xi_h^k \\
& + \mathcal{O}\left( \sqrt{\frac{\iota \mathbb{V}_h V_{h+1}^{\pi^k}(s_h^k, a_h^k, b_h^k)}{\max\{N_h^k, 1\}}} + \sqrt{\frac{\iota}{\max\{N_h^k, 1\}}} + \frac{H^2 S \iota}{\max\{N_h^k, 1\}} \right) \Bigg].
\end{aligned}
$$

The remaining steps are the same as that in the proof of Theorem 3 except that we need to bound the sum of variance term.

By Cauchy-Schwarz,

$$
\sum_{k=1}^K \sum_{h=1}^H \sqrt{\frac{\mathbb{V}_h V_{h+1}^{\pi^k}(s_h^k, a_h^k, b_h^k)}{\max\{N_h^k(s_h^k, a_h^k, b_h^k), 1\}}} \leq \sqrt{\sum_{k=1}^K \sum_{h=1}^H \mathbb{V}_h V_{h+1}^{\pi^k}(s_h^k, a_h^k, b_h^k) \cdot \sum_{k=1}^K \sum_{h=1}^H \frac{1}{\max\{N_h^k(s_h^k, a_h^k, b_h^k), 1\}}}.
$$

By the Law of total variation and standard martingale concentration (see Lemma C.5 in Jin et al. (2018) for a formal proof), with probability at least $1 - p$, we have

$$
\sum_{k=1}^K \sum_{h=1}^H \mathbb{V}_h V_{h+1}^{\pi^k}(s_h^k, a_h^k, b_h^k) \leq \mathcal{O}(HT + H^3 \iota).
$$

Putting all relations together, we obtain that with probability at least $1 - 2p$ (one $p$ comes from $\mathbb{P}(E_1) \geq 1 - p$ and the other comes from the inequality for bounding the variance term),

$$
\text{Regret}(K) = \sum_{k=1}^K (V_1^{\dagger, \nu^k} - V_1^{\mu^k, \dagger})(s_1) \leq \mathcal{O}(\sqrt{H^2 SABT \iota} + H^3 S^2 AB \iota^2).
$$

Rescaling $p$ completes the proof. $\qquad\square$

# G    PROOF FOR SECTION 4 – REWARD-FREE LEARNING

## G.1    PROOF OF THEOREM 5

In this section, we prove Theorem 5 for the single reward function case, i.e., $N = 1$. The proof for multiple reward functions ($N > 1$) simply follows from taking a union bound, that is, replacing the failure probability $p$ by $Np$.

Let $(\mu^k, \nu^k)$ be an arbitrary Nash-equilibrium policy of $\widehat{\mathcal{M}}^k := (\widehat{\mathbb{P}}^k, \widehat{r}^k)$, where $\widehat{\mathbb{P}}^k$ and $\widehat{r}^k$ are our empirical estimate of the transition and the reward at the beginning of the $k$'th episode in Algorithm 2, respectively. We use $N_h^k(s, a, b)$ to denote the number we have visited the state-action tuple $(s, a, b)$ at the $h$-th step before the $k$'th episode. And the bonus used in the $k$'th episode can be written as

$$\beta_h^k(s, a, b) := C\left(\sqrt{\frac{H^2 \iota}{\max\{N_h^k(s, a, b), 1\}}} + \frac{H^2 S \iota}{\max\{N_h^k(s, a, b), 1\}}\right), \tag{20}$$

where $\iota = \log(SABT/p)$ and $C$ is some large absolute constant.

We use $\widehat{Q}^k$ and $\widehat{V}^k$ to denote the empirical optimal value functions of $\widehat{\mathcal{M}}^k$ as following.

$$\begin{cases} \widehat{Q}_h^k(s, a, b) = (\widehat{\mathbb{P}}_h^k \widehat{V}_{h+1}^k)(s, a, b) + \widehat{r}_h^k(s, a, b), \\ \widehat{V}_h^k(s) = \max_\mu \min_\nu \mathbb{D}_{\mu \times \nu} \widehat{Q}_h^k(s). \end{cases} \tag{21}$$

Since $(\mu^k, \nu^k)$ is a Nash-equilibrium policy of $\widehat{\mathcal{M}}^k$, we also have $\widehat{V}_h^k(s) = \mathbb{D}_{\mu^k \times \nu^k} \widehat{Q}_h^k(s)$.

We begin with stating a useful property of matrix game that will be frequently used in our analysis. Since its proof is quite simple, we omit it here.

**Lemma 24.** *Let* $\mathbf{X}, \mathbf{Y}, \mathbf{Z} \in \mathbb{R}^{A \times B}$ *and* $\triangle_d$ *be the d-dimensional simplex. Suppose* $|\mathbf{X} - \mathbf{Y}| \le \mathbf{Z}$, *where the inequality is entry-wise. Then*

$$\left| \max_{\mu \in \triangle_A} \min_{\nu \in \triangle_B} \mu^\top \mathbf{X} \nu - \max_{\mu \in \triangle_A} \min_{\nu \in \triangle_B} \mu^\top \mathbf{Y} \nu \right| \le \max_{i,j} \mathbf{Z}_{ij}. \tag{22}$$

**Lemma 25.** *Let* $c_1$ *be some large absolute constant such that* $c_1^2 + c_1 \le C$. *Define event* $E_1$ *to be: for all* $h, s, a, b, s'$ *and* $k \in [K]$,

$$\begin{cases} |[(\widehat{\mathbb{P}}_h^k - \mathbb{P}_h) V_{h+1}^\star](s, a, b)| \le \dfrac{c_1}{10} \sqrt{\dfrac{H^2 \iota}{\max\{N_h^k(s, a, b), 1\}}}, \\[4mm] |(\widehat{r}_h^k - r_h)(s, a, b)| \le \dfrac{c_1}{10} \sqrt{\dfrac{H^2 \iota}{\max\{N_h^k(s, a, b), 1\}}}, \\[4mm] |(\widehat{\mathbb{P}}_h^k - \mathbb{P}_h)(s' \mid s, a, b)| \le \dfrac{c_1}{10} \left( \sqrt{\dfrac{\widehat{\mathbb{P}}_h^k(s' \mid s, a, b) \iota}{\max\{N_h^k(s, a, b), 1\}}} + \dfrac{\iota}{\max\{N_h^k(s, a, b), 1\}} \right). \end{cases} \tag{23}$$

*We have* $\mathbb{P}(E_1) \ge 1 - p$.

*Proof.* The proof is standard and folklore: apply standard concentration inequalities and then take a union bound. For completeness, we provide the proof of the third one here.

Consider a fixed $(s, a, b, h)$ tuple.

Let's consider the following equivalent random process: (a) before the agent starts, the environment samples $\{s^{(1)}, s^{(2)}, \ldots, s^{(K)}\}$ independently from $\mathbb{P}_h(\cdot \mid s, a, b)$; (b) during the interaction between the agent and environment, the $i^{\text{th}}$ time the agent reaches $(s, a, b, h)$, the environment will make the agent transit to $s^{(i)}$. Note that the randomness induced by this interaction procedure is exactly the same as the original one, which means the probability of any event in this context is the same as in the original problem. Therefore, it suffices to prove the target concentration inequality in

this 'easy' context. Denote by $\widehat{\mathbb{P}}_h^{(t)}(\cdot \mid s, a, b)$ the empirical estimate of $\mathbb{P}_h(\cdot \mid s, a, b)$ calculated using $\{s^{(1)}, s^{(2)}, \ldots, s^{(t)}\}$. For a fixed $t$ and $s'$, by the empirical Bernstein inequality, we have with probability at least $1 - p/S^2 ABT$,

$$|(\mathbb{P}_h - \widehat{\mathbb{P}}_h^{(t)})(s' \mid s, a, b)| \leq \mathcal{O}\left(\sqrt{\frac{\widehat{\mathbb{P}}_h^{(t)}(s' \mid s, a, b)\iota}{t}} + \frac{\iota}{t}\right).$$

Now we can take a union bound over all $s, a, b, h, s'$ and $t \in [K]$, and obtain that with probability at least $1 - p$, for all $s, a, b, h, s'$ and $t \in [K]$,

$$|(\mathbb{P}_h - \widehat{\mathbb{P}}_h^{(t)})(s' \mid s, a, b)| \leq \mathcal{O}\left(\sqrt{\frac{\widehat{\mathbb{P}}_h^{(t)}(s' \mid s, a, b)\iota}{t}} + \frac{\iota}{t}\right).$$

Note that the agent can reach each $(s, a, b, h)$ for at most $K$ times, this directly implies that the third inequality also holds with probability at least $1 - p$. $\qquad\square$

The following lemma states that the empirical optimal value functions are close to the true optimal ones, and their difference is controlled by the exploration value functions calculated in Algorithm 2.

**Lemma 26.** *Suppose event $E_1$ (defined in Lemma 25) holds. Then for all $h, s, a, b$ and $k \in [K]$, we have,*

$$\begin{cases} \left|\widehat{Q}_h^k(s, a, b) - Q_h^\star(s, a, b)\right| \leq \widetilde{Q}_h^k(s, a, b), \\ \left|\widehat{V}_h^k(s) - V_h^\star(s)\right| \leq \widetilde{V}_h^k(s). \end{cases} \tag{24}$$

*Proof.* Let's prove by doing backward induction on $h$. The case of $h = H + 1$ holds trivially.

Assume the conclusion hold for $(h + 1)$'th step. For $h$'th step,

$$\left|\widehat{Q}_h^k(s, a, b) - Q_h^\star(s, a, b)\right|$$
$$\leq \min\left\{\left|[(\widehat{\mathbb{P}}_h^k - \mathbb{P}_h)V_{h+1}^\star](s, a, b)\right| + |(\widehat{r}_h^k - r_h)(s, a, b)| + \left|[\widehat{\mathbb{P}}_h^k(\widehat{V}_{h+1}^k - V_{h+1}^\star)](s, a, b)\right|, H\right\}$$
$$\overset{(i)}{\leq} \min\left\{\beta_h^k(s, a, b) + (\widehat{\mathbb{P}}_h^k\widetilde{V}_{h+1}^k)(s, a, b), H\right\} \overset{(ii)}{=} \widetilde{Q}_h^k(s, a, b),$$

$$\tag{25}$$

where $(i)$ follows from the induction hypothesis and event $E_1$, and $(ii)$ follows from the definition of $\widetilde{Q}_h^k$. By Lemma 24, we immediately obtain $|\widehat{V}_h^k(s) - V_h^\star(s)| \leq \widetilde{V}_h^k(s)$. $\qquad\square$

Now, we are ready to establish the key lemma in our analysis using Lemma 26.

**Lemma 27.** *Suppose event $E_1$ (defined in Lemma 25) holds. Then for all $h, s, a, b$ and $k \in [K]$, we have*

$$\begin{cases} |\widehat{Q}_h^k(s, a, b) - Q_h^{\dagger, \nu^k}(s, a, b)| \leq \alpha_h \widetilde{Q}_h^k(s, a, b), \\ |\widehat{V}_h^k(s) - V_h^{\dagger, \nu^k}(s)| \leq \alpha_h \widetilde{V}_h^k(s), \end{cases} \tag{26}$$

*and*

$$\begin{cases} |\widehat{Q}_h^k(s, a, b) - Q_h^{\mu^k, \dagger}(s, a, b)| \leq \alpha_h \widetilde{Q}_h^k(s, a, b), \\ |\widehat{V}_h^k(s) - V_h^{\mu^k, \dagger}(s)| \leq \alpha_h \widetilde{V}_h^k(s), \end{cases} \tag{27}$$

*where $\alpha_{H+1} = 0$ and $\alpha_h = [(1 + \frac{1}{H})\alpha_{h+1} + \frac{1}{H}] \leq 4$.*

*Proof.* We only prove the first set of inequalities. The second one follows exactly the same. Again, the proof is by performing backward induction on $h$. It is trivial to see the conclusion holds for

$(H + 1)$'th step with $\alpha_{H+1} = 0$. Now, assume the conclusion holds for $(h + 1)$'th step. For $h$'th step,

$$
\begin{aligned}
&|\widehat{Q}_h^k(s, a, b) - Q_h^{\dagger, \nu^k}(s, a, b)| \\
&\leq \min \Big\{ |[(\widehat{\mathbb{P}}_h^k - \mathbb{P}_h)(V_{h+1}^{\dagger, \nu^k} - V_{h+1}^\star)](s, a, b)| + |(\widehat{\mathbb{P}}_h^k - \mathbb{P}_h)V_{h+1}^\star(s, a, b)| \\
&\quad + |(\widehat{r}_h^k - r_h)(s, a, b)| + |[\widehat{\mathbb{P}}_h(\widehat{V}_{h+1}^k - V_{h+1}^{\dagger, \nu^k})](s, a, b)|, H \Big\} \\
&\leq \min \Big\{ \underbrace{|[(\widehat{\mathbb{P}}_h^k - \mathbb{P}_h)(V_{h+1}^{\dagger, \nu^k} - V_{h+1}^\star)](s, a, b)|}_{(T_1)} + c_1 \sqrt{\frac{H^2 \iota}{\max\{N_h^k(s, a, b), 1\}}} \\
&\quad + \underbrace{|[\widehat{\mathbb{P}}_h(\widehat{V}_{h+1}^k - V_{h+1}^{\dagger, \nu^k})](s, a, b)|}_{(T_2)}, H \Big\},
\end{aligned}
\tag{28}
$$

where the second inequality follows from the definition of event $E_1$.

We can control the term $(T_1)$ by combining Lemma 26 and the induction hypothesis to bound $|V_{h+1}^{\dagger, \nu^k} - V_{h+1}^\star|$, and then applying the third inequality in event $E_1$:

$$
\begin{aligned}
(T_1) &\leq \sum_{s'} |\widehat{\mathbb{P}}_h^k(s' \mid s, a, b) - \mathbb{P}_h(s' \mid s, a, b)||V_{h+1}^{\dagger, \nu^k} - V_{h+1}^\star(s')| \\
&\leq \sum_{s'} |\widehat{\mathbb{P}}_h^k(s' \mid s, a, b) - \mathbb{P}_h(s' \mid s, a, b)|\Big(|V_{h+1}^{\dagger, \nu^k} - \widehat{V}_{h+1}^k(s')| + |\widehat{V}_{h+1}^k - V_{h+1}^\star(s')|\Big) \\
&\leq \sum_{s'} |\widehat{\mathbb{P}}_h^k(s' \mid s, a, b) - \mathbb{P}_h(s' \mid s, a, b)|(\alpha_{h+1} + 1)\widetilde{V}_{h+1}^k \\
&\leq \frac{(\alpha_{h+1} + 1)}{H}(\widehat{\mathbb{P}}_h^k \widetilde{V}_{h+1}^k)(s, a, b) + \frac{c_1^2(\alpha_{h+1} + 1)H^2 S\iota}{\max\{N_h^k(s, a, b), 1\}}.
\end{aligned}
\tag{29}
$$

The term $(T_2)$ is bounded by directly applying the induction hypothesis

$$
|[\widehat{\mathbb{P}}_h(\widehat{V}_{h+1}^k - V_{h+1}^{\dagger, \nu^k})](s, a, b)| \leq \alpha_{h+1}[\widehat{\mathbb{P}}_h \widetilde{V}_{h+1}^k](s, a, b).
\tag{30}
$$

Plugging (29) and (30) into (28), we obtain

$$
\begin{aligned}
&\left|\widehat{Q}_h^k(s, a, b) - Q_h^{\dagger, \nu^k}(s, a, b)\right| \\
&\leq \min \Big\{ (1 + \frac{1}{H})\alpha_{h+1} + \frac{1}{H}[\widehat{\mathbb{P}}_h^k \widetilde{V}_{h+1}^k](s, a, b) + c_1 \sqrt{\frac{H^2 \iota}{\max\{N_h^k(s, a, b), 1\}}} \\
&\quad + \frac{c_1^2(\alpha_{h+1} + 1)H^2 S\iota}{\max\{N_h^k(s, a, b), 1\}}, H \Big\} \\
&\overset{(i)}{\leq} \min \Big\{ \Big((1 + \frac{1}{H})\alpha_{h+1} + \frac{1}{H}\Big)[\widehat{\mathbb{P}}_h^k \widetilde{V}_{h+1}^k](s, a, b) + \beta_h^k(s, a, b), H \Big\} \\
&\overset{(ii)}{\leq} \Big((1 + \frac{1}{H})\alpha_{h+1} + \frac{1}{H}\Big)\widetilde{Q}_h^k(s, a, b),
\end{aligned}
\tag{31}
$$

where $(i)$ follows from the definition of $\beta_h^k$, and $(ii)$ follows from the definition of $\widetilde{Q}_h^k$. Therefore, by (31), choosing $\alpha_h = [(1 + \frac{1}{H})\alpha_{h+1} + \frac{1}{H}]$ suffices for the purpose of induction.

Now, let's prove the inequality for $V$ functions.

$$
\begin{aligned}
|(\widehat{V}_h^k - V_h^{\dagger, \nu^k})(s)| &\overset{(i)}{=} |\max_{\mu \in \triangle_A} (\mathbb{D}_{\mu, \nu^k} \widehat{Q}_h^k)(s) - \max_{\mu \in \triangle_A} (\mathbb{D}_{\mu, \nu^k} Q_h^{\dagger, \nu^k})(s)| \\
&\overset{(ii)}{\leq} \max_{a, b} \Big[\alpha_h \widetilde{Q}_h^k(s, a, b)\Big] = \alpha_h \widetilde{V}_h^k(s),
\end{aligned}
\tag{32}
$$

where $(i)$ follows from the definition of $\widehat{V}_h^k$ and $V_h^{\dagger, \nu^k}$, and $(ii)$ uses (31) and Lemma 24. $\qquad \square$

**Theorem 28** (Guarantee for UCB-VI from Azar et al. (2017)). *For any $p \in (0, 1]$, choose the exploration bonus $\beta_t$ in Algrothm 2 as (20). Then, with probability at least $1 - p$,*

$$\sum_{k=1}^{K} \widetilde{V}_1^k(s_1) \leq \mathcal{O}(\sqrt{H^4 SAK\iota} + H^3 S^2 A\iota^2).$$

*Proof of Theorem 5.* Recall that out $= \arg\min_{k \in [K]} \widetilde{V}_h^k(s)$. By Lemma 27 and Theorem 28, with probability at least $1 - 2p$,

$$\begin{aligned}
V_h^{\dagger, \nu^{\text{out}}}(s) - V_h^{\mu^{\text{out}}, \dagger}(s) &\leq |V_h^{\dagger, \nu^{\text{out}}}(s) - \widehat{V}_h^{\text{out}}(s)| + |\widehat{V}_h^{\text{out}}(s) - V_h^{\mu^{\text{out}}, \dagger}(s)| \\
&\leq 8\widetilde{V}_h^{\text{out}}(s) \leq \mathcal{O}\left(\sqrt{\frac{H^4 SA\iota}{K}} + \frac{H^3 S^2 A\iota^2}{K}\right).
\end{aligned} \tag{33}$$

Rescaling $p$ completes the proof. □

### G.2  VANILLA NASH VALUE ITERATION

Here, we provide one optional algorithm, Vanilla Nash VI, for computing the Nash equilibrium policy for a *known* model. Its only difference from the value iteration algorithm for MDPs is that the maximum operator is replaced by the minimax operator in Line 7. We remark that the Nash equilibrium for a two-player zero-sum game can be computed in polynomial time.

---

**Algorithm 5** Vanilla Nash Value Iteration

---

1: **Input**: model $\widehat{\mathcal{M}} = (\widehat{\mathbb{P}}, \widehat{r})$.
2: **Initialize:** for all $(s, a, b)$, $V_{H+1}(s, a, b) \leftarrow 0$.
3: **for** step $h = H, H-1, \ldots, 1$ **do**
4:    **for** $(s, a, b) \in \mathcal{S} \times \mathcal{A} \times \mathcal{B}$ **do**
5:       $Q_h(s, a, b) \leftarrow [\widehat{\mathbb{P}}_h V_{h+1}](s, a, b) + \widehat{r}_h(s, a, b)$.
6:    **for** $s \in \mathcal{S}$ **do**
7:       $(\hat{\mu}_h(\cdot \mid s), \hat{\nu}_h(\cdot \mid s)) \leftarrow \text{NASH-ZERO-SUM}(Q_h(s, \cdot, \cdot))$.
8:       $V_h(s) \leftarrow \hat{\mu}_h(\cdot \mid s)^\top Q_h(s, \cdot, \cdot)\hat{\nu}_h(\cdot \mid s)$.
9: **Output** $(\hat{\mu}, \hat{\nu}) \leftarrow \{(\hat{\mu}_h(\cdot \mid s), \hat{\nu}_h(\cdot \mid s))\}_{(h, s) \in [H] \times \mathcal{S}}$.

---

By recalling the definition of best responses in Appendix D, one can directly see that the output policy $(\hat{\mu}, \hat{\nu})$ is a Nash equilibrium for $\widehat{\mathcal{M}}$.

### G.3  PROOF OF THEOREM 6

In this section, we first prove a $\Theta(AB/\epsilon^2)$ lower bound for reward-free matrix games, i.e., $S = H = 1$, and then generalize it to $\Theta(SABH^2/\epsilon^2)$ for the Markov games setting.

#### G.3.1  REWARD-FREE MATRIX GAMES

In the matrix game, let the max-player pick row and the min-player pick column. We consider the following family of Bernoulli matrix games:

$$\mathfrak{M}(\epsilon) = \left\{ \mathcal{M} \in \mathbb{R}^{A \times B} \text{ with } \mathcal{M}_{ab} = \frac{1}{2} + (1 - 2 \cdot \mathbf{1}\{a \neq a^\star \& b = b^\star\})\epsilon : \quad (a^\star, b^\star) \in [A] \times [B] \right\}, \tag{34}$$

where in matrix game $\mathcal{M}$, the reward is sampled from $\text{Bernoulli}(\mathcal{M}_{ab})$ if the max-player picks the $a$'th row and the min-player picks the $b$'th column.

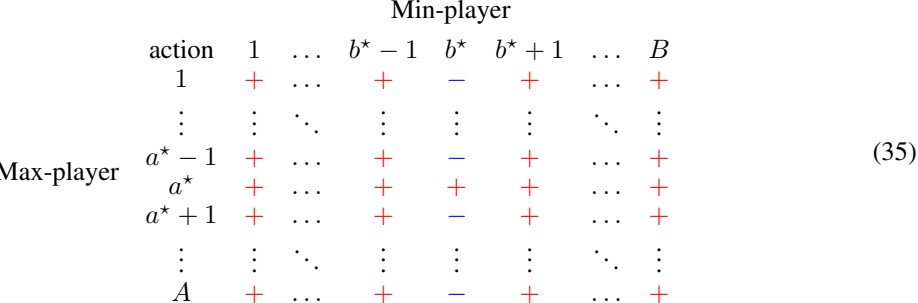

$$(35)$$

Above, we visualize the hard instance by using $+$ and $-$ to represent $1/2+\epsilon$ and $1/2-\epsilon$, respectively. It is direct to see that the optimal policy for the max-player is always picking the $a^\star$'th row and the optimal policy for the min-player is always picking the $b^\star$'th column. If the max-player picks the $a^\star$'th row with probability smaller than $2/3$, it is at least $\epsilon/10$ suboptimal.

**Lemma 29.** *For any fixed matrix game $\mathcal{M}$ from $\mathfrak{M}(\epsilon)$ and $N \in \mathbb{N}$, if an algorithm $\mathcal{A}$ can output a policy that is at most $\epsilon/10$ suboptimal with probability at least $p$ using at most $N$ samples, then there exists an algorithm $\hat{\mathcal{A}}$ that can identify the best row in $\mathcal{M}$ with probability at least $p$ using at most $N$ samples.*

*Proof.* We simply define $\hat{\mathcal{A}}$ as running algorithm $\mathcal{A}$ and choosing the most played row by its outputted policy as the guess for the best row. By simple calculation, one can show $\hat{\mathcal{A}}$ will output the best row in $\mathcal{M}$ with probability at least $p$. □

Lemma 29 directly implies that in order to prove the desired lower bound for matrix games:

**Claim 30.** *for any algorithm $\mathcal{A}$ using at most $N = AB/(10^3\epsilon^2)$ samples, there exists a matrix game $\mathcal{M}$ in $\mathfrak{M}(\epsilon)$ such that when running $\mathcal{A}$ on $\mathcal{M}$, it will output a policy that is at least $\epsilon/10$ suboptimal for the max-player with probability at least $1/4$,*

it suffices to prove the following claim:

**Claim 31.** *for any algorithm $\hat{\mathcal{A}}$ using at most $N = AB/(10^3\epsilon^2)$ samples, there exists a matrix game $\mathcal{M}$ in $\mathfrak{M}(\epsilon)$ such that when running $\hat{\mathcal{A}}$ on $\mathcal{M}$, it will fail to identify the optimal row with probability at least $1/4$.*

*Proof of Claim 31.* WLOG, we assume $\hat{\mathcal{A}}$ is deterministic. Since this is the reward-free setting, being deterministic means that algorithm $\hat{\mathcal{A}}$ will always pull each arm $(a, b)$ for some *fixed* $n(a, b)$ times and then output a guess for $a^\star$ which is a function of the reward revealed.

Denote by $L$ the reward revealed after algorithm $\hat{\mathcal{A}}$'s pulling. Denote by $\mathbb{P}_\star$ the probability induced by picking $\mathcal{M}$ uniformly at random from $\mathfrak{M}(\epsilon)$ and running $\hat{\mathcal{A}}$ on $\mathcal{M}$. Denote by $\mathbb{P}_{a,b}$ the probability induced by running $\hat{\mathcal{A}}$ on $\mathcal{M}$, whose indices of the special row and the special column are $a$ and $b$, respectively. Denote by $\mathbb{P}_{0,b}$ the probability induced by running $\mathcal{A}$ on $\mathcal{M}$, whose $b$'th column are all $1/2 - \epsilon$ and other columns are all $1/2 + \epsilon$. We want to mention that the $\mathcal{M}$ we use to define $\mathbb{P}_{0,b}$ does not belong to $\mathfrak{M}(\epsilon)$.

We have

$$
\begin{aligned}
\mathbb{P}_\star(\hat{\mathcal{A}}(L) \neq a^\star) &\geq \frac{1}{AB} \sum_{a,b} \mathbb{P}_{0,b}(\hat{\mathcal{A}}(L) \neq a) - \frac{1}{AB} \sum_{a,b} \|\mathbb{P}_{a,b} - \mathbb{P}_{0,b}\|_1 \\
&\geq 1 - \frac{1}{A} - \frac{1}{AB} \sum_{a,b} \sqrt{2\mathrm{KL}(\mathbb{P}_{0,b}\|\mathbb{P}_{a,b})} \\
&= 1 - \frac{1}{A} - \frac{1}{AB} \sum_{a,b} \sqrt{2n(a,b)[(\frac{1}{2} - \epsilon) \log \frac{\frac{1}{2} - \epsilon}{\frac{1}{2} + \epsilon} + (\frac{1}{2} + \epsilon) \log \frac{\frac{1}{2} + \epsilon}{\frac{1}{2} - \epsilon}]} \quad (36) \\
&\geq 1 - \frac{1}{A} - \frac{10}{AB} \sum_{a,b} \sqrt{n(a,b)\epsilon^2} \\
&\geq 1 - \frac{1}{A} - \sqrt{\frac{100N\epsilon^2}{AB}}.
\end{aligned}
$$

Choosing $N = AB/(10^3\epsilon^2)$ concludes the proof. $\qquad\square$

### G.3.2 REWARD-FREE MARKOV GAMES

Now let's generalize the $\Theta(AB/\epsilon^2)$ lower bound to $\Theta(SABH^2/\epsilon^2)$ for reward-free Markov games.

We define the following family of MDPs:

$$
\mathfrak{J}(\epsilon) := \left\{ \mathcal{J}(\boldsymbol{a}^\star, \boldsymbol{b}^\star) : (\boldsymbol{a}^\star, \boldsymbol{b}^\star) \in [A]^{H \times S} \times [B]^{H \times S} \right\}, \quad (37)
$$

where MDP $\mathcal{J}(\boldsymbol{a}^\star, \boldsymbol{b}^\star)$ is defined as below:

- **States and actions:** $\mathcal{J}(\boldsymbol{a}^\star, \boldsymbol{b}^\star)$ is a finite-horizon MDP with $S + 1$ states of length $H + 1$. There is a fixed initial state $s_0$ in the first step, $S$ states $\{s_1, \ldots, s_S\}$ for the remaining steps. The two players have $A$ and $B$ actions, respectively.

- **Rewards:** there is no reward in the first step. For the remaining steps $h \in \{2, \ldots, H + 1\}$, if the agent takes action $(a, b)$ at state $s_i$ in the $h^{\text{th}}$ step, it will receive a binary reward sampled from

$$
\text{Bernoulli}\left(\frac{1}{2} + (1 - 2 \cdot \mathbf{1}\{a \neq \boldsymbol{a}^\star_{h-1,i} \& b = \boldsymbol{b}^\star_{h-1,i}\}) \frac{\epsilon}{H}\right)
$$

- **Transitions:** Regardless of the current state, actions and index of steps, the agent will always transit to one of $s_1, \ldots, s_S$ uniformly at random.

It is direct to see that $\mathcal{J}(\boldsymbol{a}^\star, \boldsymbol{b}^\star)$ is a collection of $SH$ independent matrix games from $\mathfrak{M}(\epsilon/H)$. Therefore, the optimal policy for the max player is to always pick action $\boldsymbol{a}^\star_{h-1,i}$ whenever it reaches state $s_i$ in the $h^{\text{th}}$ step ($h \geq 2$). In other words, $\boldsymbol{a}^\star_{h-1,i}$ is the unique optimal action for the step-state pair $(h, i)$.

At a high level, in order to find an $\epsilon$-optimal policy for the above Markov game, we need to identify at least half of the entries of $\boldsymbol{a}^\star$. Therefore, the number of episodes should be at least

$$
\Theta\left(\frac{AB}{(\epsilon/H)^2}\right) \times \frac{S}{2} = \Theta\left(\frac{ABSH^2}{\epsilon^2}\right).
$$

Below we provide a formal proof of this argument, which is almost the same as that for the setting of reward-free matrix games.

We start by proving an analogue of Lemma 29.

**Lemma 32.** *For any fixed matrix game $\mathcal{J}(\boldsymbol{a}^\star, \boldsymbol{b}^\star)$ from $\mathfrak{J}(\epsilon)$ and $N \in \mathbb{N}$, if an algorithm $\mathcal{A}$ can output a policy that is at most $\epsilon/10^3$ suboptimal with probability at least $p$ using at most $N$ samples, then there exists an algorithm $\hat{\mathcal{A}}$ that can correctly identify at least $SH - \lfloor SH/500 \rfloor$ entries of $\boldsymbol{a}^\star$ with probability at least $p$ using at most $N$ samples.*

*Proof.* Denote by $\pi$ the output policy for the max player. Denote by $Z$ the collection of $(h, i)$'s in $[H] \times [S]$ such that $\pi_{h+1}(\boldsymbol{a}^\star_{h,i} \mid s_i) \leq 2/3$.

Observe that each time the max player picks a suboptimal action, it will incur an $2\epsilon/H$ suboptimality in expectation. As a result, if $\pi$ is at most $\epsilon/10^3$-suboptimal, we must have

$$\frac{1}{S} \sum_{(h,i)\in Z} (1 - \pi_{h+1}(\boldsymbol{a}^\star_{h,i} \mid s_i)) \times \frac{2\epsilon}{H} \leq \frac{\epsilon}{10^3},$$

which implies $|Z| \leq SH/500$, that is, for at most $\lfloor SH/500 \rfloor$ different $i$'s, $\pi(\boldsymbol{a}^\star_i \mid s_i) \leq 2/3$. Therefore, we can simply pick $\arg\max_a \pi_{h+1}(a \mid s_i)$ as the guess for $\boldsymbol{a}^\star_{h,i}$. Since policy $\pi$ is at most $\epsilon/10^3$ suboptimal with probability at least $p$, we can correctly identify the optimal actions for at least $SH - \lfloor SH/500 \rfloor$ different $(s, h)$ pairs also with probability no smaller than $p$. $\square$

Lemma 32 directly implies that in order to prove the desired lower bound for reward-free Markov games:

**Claim 33.** *for any algorithm $\mathcal{A}$ interacting with the environment for at most $K = ABSH^2/(10^4\epsilon^2)$ episodes, there exists $\mathcal{J} \in \mathfrak{J}(\epsilon)$ such that when running $\mathcal{A}$ on $\mathcal{J}$, it will output a policy that is at least $\epsilon/10^3$ suboptimal for the max-player with probability at least $1/4$,*

it suffices to prove the following claim:

**Claim 34.** *for any algorithm $\hat{\mathcal{A}}$ interacting with the environment for at most $K = ABSH^2/(10^4\epsilon^2)$ episodes, there exists $\mathcal{J} \in \mathfrak{J}(\epsilon)$ such that when running $\hat{\mathcal{A}}$ on $\mathcal{J}$, it will fail to identify the optimal actions for at least $\lfloor SH/500 \rfloor + 1$ different $(s, h)$ pairs with probability at least $1/4$.*

*Proof of Claim 34.* Denote by $\mathbb{P}_\star$ ($\mathbb{E}_\star$) the probability (expectation) induced by picking $\mathcal{J}(\boldsymbol{a}^\star, \boldsymbol{b}^\star)$ uniformly at random from $\mathfrak{J}(\epsilon)$ and running $\hat{\mathcal{A}}$ on $\mathcal{J}$. Denote by $n_{\text{wrong}}$ the number of $(s, h)$ pairs for which $\hat{\mathcal{A}}$ fails to identify the optimal actions. Denote by $\text{error}_{h,i}$ the indicator function of the event that $\hat{\mathcal{A}}$ fails to identify the optimal action for $(h + 1, i)$.

We prove by contradiction. Suppose for any $\mathcal{J} \in \mathfrak{J}(\epsilon)$, $\hat{\mathcal{A}}$ can identify the optimal actions for at least $SH - \lfloor SH/500 \rfloor$ different $(s, h)$ pairs with probability larger than $3/4$. Then we have

$$\mathbb{E}_\star[n_{\text{wrong}}] \leq \frac{1}{4} \times SH + \frac{3}{4} \times \left\lfloor \frac{SH}{500} \right\rfloor \leq \frac{101SH}{400}.$$

Since $\sum_{(h,i)\in[H]\times[S]} \mathbb{E}_\star[\text{error}_{h,i}] = \mathbb{E}_\star[n_{\text{wrong}}]$, there must exists $(h', i') \in [H] \times [S]$ such that $\mathbb{E}_\star[\text{error}_{h',i'}] \leq 101/400$. However, in the following, we show that for every $(h, i) \in [H] \times [S]$, $\hat{\mathcal{A}}$ fails to identify the optimal action for the step-state pair $(h+1, i)$ with probability at least $1/3$, which directly implies $\mathbb{E}_\star[\text{error}_{h,i}] \geq 1/3$ for all $(h, i) \in [H] \times [S]$. As a result, we obtain a contraction and Claim 34 holds.

Now, let us prove that for every $(h, i) \in [H] \times [S]$, $\mathbb{E}_\star[\text{error}_{h,i}] \geq 1/6$. WLOG, we assume $\hat{\mathcal{A}}$ is deterministic and it runs for exactly $K = ABSH^2/(10^3\epsilon^2)$ episodes. In the following, we consider a *fixed* $(h', i')$ pair. For technical reason, we define MDP $\mathcal{J}_{-(h',i')}(\boldsymbol{a}^\star, \boldsymbol{b}^\star)$ as below:

- **States, actions and transitions:** same as $\mathcal{J}(\boldsymbol{a}^\star, \boldsymbol{b}^\star)$.

- **Rewards:** there is no reward in the first step. For the remaining steps $h \in \{2, \ldots, H+1\}$, if the agent takes action $(a, b)$ at state $s_i$ in the $h^{\text{th}}$ step such that $(h-1, i) \neq (h', i')$, it will receive a binary reward sampled from

$$\text{Bernoulli}\left(\frac{1}{2} + (1 - 2 \cdot \mathbf{1}\{a \neq \boldsymbol{a}^\star_{h-1,i} \& b = \boldsymbol{b}^\star_{h-1,i}\})\frac{\epsilon}{H}\right),$$

otherwise it will receive a binary reward sampled from

$$\text{Bernoulli}\left(\frac{1}{2} + (1 - 2 \cdot \mathbf{1}\{b = \boldsymbol{b}^\star_{h-1,i}\})\frac{\epsilon}{H}\right).$$

Intuitively, $\mathcal{J}_{-(h',i')}(\boldsymbol{a}^\star, \boldsymbol{b}^\star)$ is the same as $\mathcal{J}(\boldsymbol{a}^\star, \boldsymbol{b}^\star)$ except that for the max player, all its actions at state $s_{i'}$ in the $h'^{\text{th}}$ step are equivalent. In other words, $\mathcal{J}_{-(h',i')}(\boldsymbol{a}^\star, \boldsymbol{b}^\star)$ is independent of $\boldsymbol{a}^\star_{h',i'}$.

To proceed, we need to define the following notations: denote by $n(a,b)$ the number of times $\hat{\mathcal{A}}$ picks action $(a,b)$ at state $s_{i'}$ in the $(h'+1)^{\text{th}}$ step; denote by $\mathbb{P}(\cdot \mid \mathcal{J}(\boldsymbol{a}^\star, \boldsymbol{b}^\star))$ $(\mathbb{E}[\cdot \mid \mathcal{J}(\boldsymbol{a}^\star, \boldsymbol{b}^\star)])$ the probability (expectation) induced by running algorithm $\hat{\mathcal{A}}$ on $\mathcal{J}(\boldsymbol{a}^\star, \boldsymbol{b}^\star)$; similarly, we define $\mathbb{P}(\cdot \mid \mathcal{J}_{-(h',i')}(\boldsymbol{a}^\star, \boldsymbol{b}^\star))$ $(\mathbb{E}[\cdot \mid \mathcal{J}_{-(h',i')}(\boldsymbol{a}^\star, \boldsymbol{b}^\star)])$; also recall that we denote by $\mathbb{P}_\star$ $(\mathbb{E}_\star)$ the probability (expectation) induced by picking $\mathcal{J}(\boldsymbol{a}^\star, \boldsymbol{b}^\star)$ uniformly at random from $\mathfrak{J}(\epsilon)$ and running $\hat{\mathcal{A}}$ on $\mathcal{J}$; denote by $L$ the whole trajectory of states, actions and rewards produced by algorithm $\hat{\mathcal{A}}$ in $N$ episodes; with slight abuse of notation, denote by $\hat{\mathcal{A}}(L)$ the guess of $\hat{\mathcal{A}}$ for $\boldsymbol{a}^\star_{h',i'}$ based on $L$.

First, note that for any $(a,b) \in [A] \times [B]$, $\mathbb{E}[n(s,a) \mid \mathcal{J}_{-(h',i')}(\boldsymbol{a}^\star, \boldsymbol{b}^\star)]$ is independent of $(h', i', \boldsymbol{a}^\star, \boldsymbol{b}^\star)$ because the agent cannot observe any reward when interacting with the environment and the transition dynamics of different $\mathcal{J}_{-(h',i')}(\boldsymbol{a}^\star, \boldsymbol{b}^\star)$'s are the same. For simplicity of notation, we denote this expectation by $m(a,b)$. Note that $\sum_{a,b} m(a,b) = K/S$ because the agent always reach state $s_{i'}$ in the $(h'+1)^{\text{th}}$ step with probability $1/S$ regardless of the actions taken.

We have

$$
\mathbb{P}_\star(\hat{\mathcal{A}}(L) \neq \boldsymbol{a}^\star_{h',i'})
$$

$$
= \frac{1}{(AB)^{SH}} \sum_{(\boldsymbol{a}^\star, \boldsymbol{b}^\star) \in [A]^{SH} \times [B]^{SH}} \mathbb{P}(\hat{\mathcal{A}}(L) \neq \boldsymbol{a}^\star_{h',i'} \mid \mathcal{J}(\boldsymbol{a}^\star, \boldsymbol{b}^\star))
$$

$$
\geq \frac{1}{(AB)^{SH}} \sum_{(\boldsymbol{a}^\star, \boldsymbol{b}^\star) \in [A]^{SH} \times [B]^{SH}} \bigg( \mathbb{P}(\hat{\mathcal{A}}(L) \neq \boldsymbol{a}^\star_{h',i'} \mid \mathcal{J}_{-(h',i')}(\boldsymbol{a}^\star, \boldsymbol{b}^\star))
$$

$$
- \big\| \mathbb{P}(L = \cdot \mid \mathcal{J}_{-(h',i')}(\boldsymbol{a}^\star, \boldsymbol{b}^\star)) - \mathbb{P}(L = \cdot \mid \mathcal{J}(\boldsymbol{a}^\star, \boldsymbol{b}^\star)) \big\|_1 \bigg)
$$

$$
= 1 - \frac{1}{A} - \frac{1}{(AB)^{SH}} \sum_{(\boldsymbol{a}^\star, \boldsymbol{b}^\star) \in [A]^{SH} \times [B]^{SH}} \big\| \mathbb{P}(L = \cdot \mid \mathcal{J}_{-(h',i')}(\boldsymbol{a}^\star, \boldsymbol{b}^\star)) - \mathbb{P}(L = \cdot \mid \mathcal{J}(\boldsymbol{a}^\star, \boldsymbol{b}^\star)) \big\|_1
$$

$$
\geq 1 - \frac{1}{A} - \frac{1}{(AB)^{SH}} \sum_{(\boldsymbol{a}^\star, \boldsymbol{b}^\star) \in [A]^{SH} \times [B]^{SH}} \sqrt{2\mathrm{KL}(\mathbb{P}(L = \cdot \mid \mathcal{J}_{-(h',i')}(\boldsymbol{a}^\star, \boldsymbol{b}^\star)), \mathbb{P}(L = \cdot \mid \mathcal{J}(\boldsymbol{a}^\star, \boldsymbol{b}^\star)))}
$$

$$
= 1 - \frac{1}{A} - \frac{1}{(AB)^{SH}} \sum_{(\boldsymbol{a}^\star, \boldsymbol{b}^\star) \in [A]^{SH} \times [B]^{SH}} \sqrt{2m(\boldsymbol{a}^\star_{h',i}, \boldsymbol{b}^\star_{h',i}) \Big[ (\frac{1}{2} - \frac{\epsilon}{H}) \log \frac{\frac{1}{2} - \frac{\epsilon}{H}}{\frac{1}{2} + \frac{\epsilon}{H}} + (\frac{1}{2} + \frac{\epsilon}{H}) \log \frac{\frac{1}{2} + \frac{\epsilon}{H}}{\frac{1}{2} - \frac{\epsilon}{H}} \Big]}
$$

$$
= 1 - \frac{1}{A} - \frac{1}{AB} \sum_{(\boldsymbol{a}^\star_{i,h}, \boldsymbol{b}^\star_{i,h}) \in [A] \times [B]} \sqrt{2m(\boldsymbol{a}^\star_{h',i}, \boldsymbol{b}^\star_{h',i}) \Big[ (\frac{1}{2} - \frac{\epsilon}{H}) \log \frac{\frac{1}{2} - \frac{\epsilon}{H}}{\frac{1}{2} + \frac{\epsilon}{H}} + (\frac{1}{2} + \frac{\epsilon}{H}) \log \frac{\frac{1}{2} + \frac{\epsilon}{H}}{\frac{1}{2} - \frac{\epsilon}{H}} \Big]}
$$

$$
\geq 1 - \frac{1}{A} - \frac{10}{AB} \sum_{a,b} \sqrt{m(a,b) \frac{\epsilon^2}{H^2}}
$$

$$
\geq 1 - \frac{1}{A} - \frac{10\epsilon}{ABH} \sqrt{AB \sum_{a,b} m(a,b)} = 1 - \frac{1}{A} - \sqrt{\frac{100K\epsilon^2}{SABH^2}}.
$$

(38)

Plugging in $K = SABH^2/(10^4 \epsilon^2)$ completes the proof. $\qquad\square$

# H PROOF FOR APPENDIX C – MULTI-PLAYER GENERAL-SUM MARKOV GAMES

## H.1 PROOF OF THEOREM 15

### H.1.1 NE VERSION

In this section, we prove Theorem 15 (NE version). As before, we begin with proving the optimistic estimations are indeed upper bounds of corresponding value and Q-value functions.

**Lemma 35.** *With probability $1 - p$, for any $(s, \boldsymbol{a}, h, k, i)$:*

$$\overline{Q}_{h,i}^{k}(s, \boldsymbol{a}) \geq Q_{h,i}^{\dagger, \pi_{-i}^{k}}(s, \boldsymbol{a}), \quad \underline{Q}_{h,i}^{k}(s, \boldsymbol{a}) \leq Q_{h,i}^{\pi^{k}}(s, \boldsymbol{a}), \tag{39}$$

$$\overline{V}_{h,i}^{k}(s) \geq V_{h,i}^{\dagger, \pi_{-i}^{k}}(s), \quad \underline{V}_{h,i}^{k}(s) \leq V_{h,i}^{\pi^{k}}(s). \tag{40}$$

*Proof.* For each fixed $k$, we prove this by induction from $h = H + 1$ to $h = 1$. For base case, we know at the $(H + 1)$-th step, $V_{H+1,i}^{k}(s) = \overline{V}_{H+1,i}^{\dagger, \pi_{-i}^{k}}(s) = 0$. Now, assume the inequality (40) holds for the $(h + 1)$-th step, for the $h$-th step, by definition of the $Q$ functions,

$$\overline{Q}_{h,i}^{k}(s, \boldsymbol{a}) - Q_{h,i}^{\dagger, \pi_{-i}^{k}}(s, \boldsymbol{a}) = \left[\widehat{\mathbb{P}}_{h}^{k} \overline{V}_{h+1,i}^{k}\right](s, \boldsymbol{a}) - \left[\mathbb{P}_{h} V_{h+1,i}^{\dagger, \pi_{-i}^{k}}\right](s, \boldsymbol{a}) + \beta_{t}$$

$$= \underbrace{\widehat{\mathbb{P}}_{h}^{k}\left(\overline{V}_{h+1,i}^{k} - V_{h+1,i}^{\dagger, \pi_{-i}^{k}}\right)(s, \boldsymbol{a})}_{(A)} + \underbrace{\left(\widehat{\mathbb{P}}_{h}^{k} - \mathbb{P}_{h}\right) V_{h+1,i}^{\dagger, \pi_{-i}^{k}}(s, \boldsymbol{a})}_{(B)} + \beta_{t}.$$

By induction hypothesis, for any $s'$, $\left(\overline{V}_{h+1,i}^{k} - V_{h+1,i}^{\dagger, \pi_{-i}^{k}}\right)(s') \geq 0$, and thus $(A) \geq 0$. By uniform concentration (e.g., Lemma 12 in Bai & Jin (2020)), $(B) \leq C\sqrt{SH^{2}\iota/N_{h}^{k}(s, \boldsymbol{a})} = \beta_{t}$. Putting everything together we have $Q_{h,i}^{k}(s, \boldsymbol{a}) - Q_{h,i}^{\dagger, \pi_{-i}^{k}}(s, \boldsymbol{a}) \geq 0$. The second inequality can be proved similarly.

Now assume inequality (39) holds for the $h$-th step, by definition of value functions and Nash equilibrium,

$$\overline{V}_{h,i}^{k}(s) = \mathbb{D}_{\pi^{k}} \overline{Q}_{h,i}^{k}(s) = \max_{\mu} \mathbb{D}_{\mu \times \pi_{-i}^{k}} \overline{Q}_{h,i}^{k}(s).$$

By Bellman equation,

$$V_{h,i}^{\dagger, \pi_{-i}^{k}}(s) = \max_{\mu} \mathbb{D}_{\mu \times \pi_{-i}^{k}} Q_{h,i}^{\dagger, \pi_{-i}^{k}}(s).$$

Since by induction hypothesis, for any $(s, \boldsymbol{a})$, $\overline{Q}_{h,i}^{k}(s, \boldsymbol{a}) \geq Q_{h,i}^{\dagger, \pi_{-i}^{k}}(s, \boldsymbol{a})$. As a result, we also have $\overline{V}_{h,i}^{k}(s) \geq V_{h,i}^{\dagger, \pi_{-i}^{k}}(s)$, which is exactly inequality (40) for the $h$-th step. The second inequality can be proved similarly. $\square$

*Proof of Theorem 15.* Let us focus on the $i$-th player and ignore the subscript when there is no confusion. To bound

$$\max_{i}\left(V_{1,i}^{\dagger, \pi_{-i}^{k}} - V_{1,i}^{\pi^{k}}\right)(s_{h}^{k}) \leq \max_{i}\left(\overline{V}_{1,i}^{k} - \underline{V}_{1,i}^{k}\right)(s_{h}^{k}),$$

we notice the following propogation:

$$\begin{cases} (\overline{Q}_{h,i}^{k} - \underline{Q}_{h,i}^{k})(s, \boldsymbol{a}) \leq \widehat{\mathbb{P}}_{h}^{k}(\overline{V}_{h+1,i}^{k} - \underline{V}_{h+1,i}^{k})(s, \boldsymbol{a}) + 2\beta_{h}^{k}(s, \boldsymbol{a}), \\ (\overline{V}_{h,i}^{k} - \underline{V}_{h,i}^{k})(s) = [\mathbb{D}_{\pi_{h}}(\overline{Q}_{h,i}^{k} - \underline{Q}_{h,i}^{k})](s). \end{cases} \tag{41}$$

We can define $\widetilde{Q}_h^k$ and $\widetilde{V}_h^k$ recursively by $\widetilde{V}_{H+1}^k = 0$ and

$$
\begin{cases}
\widetilde{Q}_h^k(s,\boldsymbol{a}) = \widehat{\mathbb{P}}_h^k \widetilde{V}_{h+1}^k(s,\boldsymbol{a}) + 2\beta_h^k(s,\boldsymbol{a}), \\
\widetilde{V}_h^k(s) = [\mathbb{D}_{\pi_h} \widetilde{Q}_h^k](s).
\end{cases}
\tag{42}
$$

Then we can prove inductively that for any $k$, $h$, $s$ and $\boldsymbol{a}$ we have

$$
\begin{cases}
\max_i (\overline{Q}_{h,i}^k - \underline{Q}_{h,i}^k)(s,\boldsymbol{a}) \le \widetilde{Q}_h^k(s,\boldsymbol{a}), \\
\max_i (\overline{V}_{h,i} - \underline{V}_{h,i})(s) \le \widetilde{V}_h^k(s).
\end{cases}
\tag{43}
$$

Thus we only need to bound $\sum_{k=1}^K \widetilde{V}_1^k(s)$. Define the shorthand notation

$$
\begin{cases}
\beta_h^k := \beta_h^k(s_h^k, \boldsymbol{a}_h^k), \\
\Delta_h^k := \widetilde{V}_h^k(s_h^k), \\
\zeta_h^k := \mathbb{D}_{\pi^k} \widetilde{Q}_h^k(s_h^k) - \widetilde{Q}_h^k(s_h^k, \boldsymbol{a}_h^k), \\
\xi_h^k := \mathbb{P}_h \widetilde{V}_h^k(s_h^k, \boldsymbol{a}_h^k) - \Delta_{h+1}^k.
\end{cases}
\tag{44}
$$

We can check $\zeta_h^k$ and $\xi_h^k$ are martingale difference sequences. As a result,

$$
\begin{aligned}
\Delta_h^k &= \mathbb{D}_{\pi^k} \widetilde{Q}_h^k(s_h^k) \\
&= \zeta_h^k + \widetilde{Q}_h^k(s_h^k, \boldsymbol{a}_h^k) \\
&= \zeta_h^k + 2\beta_h^k + \widehat{\mathbb{P}}_h^k \widetilde{V}_{h+1}^k(s_h^k, \boldsymbol{a}_h^k) \\
&\le \zeta_h^k + 3\beta_h^k + \mathbb{P}_h \widetilde{V}_{h+1}^k(s_h^k, \boldsymbol{a}_h^k) \\
&= \zeta_h^k + 3\beta_h^k + \xi_h^k + \Delta_{h+1}^k.
\end{aligned}
$$

Recursing this argument for $h \in [H]$ and taking the sum,

$$
\sum_{k=1}^K \Delta_1^k \le \sum_{k=1}^K \left( \zeta_h^k + 3\beta_h^k + \xi_h^k \right) \le O\left( S \sqrt{H^3 T \iota \prod_{i=1}^M A_i} \right).
$$

$\square$

### H.1.2 CCE Version

The proof is very similar to the NE version. Specifically, the only part that uses the properties of NE there is Lemma 35. We prove a counterpart here.

**Lemma 36.** *With probability $1 - p$, for any $(s, \boldsymbol{a}, h, k, i)$:*

$$
\overline{Q}_{h,i}^k(s,\boldsymbol{a}) \ge Q_{h,i}^{\dagger, \pi_{-i}^k}(s,\boldsymbol{a}), \quad \underline{Q}_{h,i}^k(s,\boldsymbol{a}) \le Q_{h,i}^{\pi^k}(s,\boldsymbol{a}),
\tag{45}
$$

$$
\overline{V}_{h,i}^k(s) \ge V_{h,i}^{\dagger, \pi_{-i}^k}(s), \quad \underline{V}_{h,i}^k(s) \le V_{h,i}^{\pi^k}(s).
\tag{46}
$$

*Proof.* For each fixed $k$, we prove this by induction from $h = H + 1$ to $h = 1$. For base case, we know at the $(H+1)$-th step, $V_{H+1,i}^k(s) = \overline{V}_{H+1,i}^{\dagger, \pi_{-i}^k}(s) = 0$. Now, assume the inequality (40) holds for the $(h+1)$-th step, for the $h$-th step, by definition of the $Q$ functions,

$$
\begin{aligned}
\overline{Q}_{h,i}^k(s,\boldsymbol{a}) - Q_{h,i}^{\dagger, \pi_{-i}^k}(s,\boldsymbol{a}) &= \left[\widehat{\mathbb{P}}_h^k \overline{V}_{h+1,i}^k\right](s,\boldsymbol{a}) - \left[\mathbb{P}_h V_{h+1,i}^{\dagger, \pi_{-i}^k}\right](s,\boldsymbol{a}) + \beta_t \\
&= \underbrace{\widehat{\mathbb{P}}_h^k \left( \overline{V}_{h+1,i}^k - V_{h+1,i}^{\dagger, \pi_{-i}^k} \right)(s,\boldsymbol{a})}_{(A)} + \underbrace{\left( \widehat{\mathbb{P}}_h^k - \mathbb{P}_h \right) V_{h+1,i}^{\dagger, \pi_{-i}^k}(s,\boldsymbol{a})}_{(B)} + \beta_t.
\end{aligned}
$$

By induction hypothesis, for any $s'$, $\left( \overline{V}_{h+1,i}^{k} - V_{h+1,i}^{\dagger,\pi_{-i}^{k}} \right) (s') \geq 0$, and thus $(A) \geq 0$. By uniform concentration, $(B) \leq C\sqrt{SH^2\iota/N_h^k(s,\boldsymbol{a})} = \beta_t$. Putting everything together we have $Q_{h,i}^k(s,\boldsymbol{a}) - Q_{h,i}^{\dagger,\pi_{-i}^k}(s,\boldsymbol{a}) \geq 0$. The second inequality can be proved similarly.

Now assume inequality (45) holds for the $h$-th step, by definition of value functions and CCE,

$$\overline{V}_{h,i}^k(s) = \mathbb{D}_{\pi^k}\overline{Q}_{h,i}^k(s) = \max_\mu \mathbb{D}_{\mu \times \pi_{-i}^k}\overline{Q}_{h,i}^k(s).$$

By Bellman equation,

$$V_{h,i}^{\dagger,\pi_{-i}^k}(s) = \max_\mu \mathbb{D}_{\mu \times \pi_{-i}^k}Q_{h,i}^{\dagger,\pi_{-i}^k}(s).$$

Since by induction hypothesis, for any $(s,\boldsymbol{a})$, $\overline{Q}_{h,i}^k(s,\boldsymbol{a}) \geq Q_{h,i}^{\dagger,\pi_{-i}^k}(s,\boldsymbol{a})$. As a result, we also have $\overline{V}_{h,i}^k(s) \geq V_{h,i}^{\dagger,\pi_{-i}^k}(s)$, which is exactly inequality (40) for the $h$-th step. The second inequality can be proved similarly. □

### H.1.3 CE VERSION

The proof is very similar to the NE version. Specifically, the only part that uses the properties of NE there is Lemma 35. We prove a counterpart here.

**Lemma 37.** *With probability $1 - p$, for any $(s,\boldsymbol{a},h,k,i)$:*

$$\overline{Q}_{h,i}^k(s,\boldsymbol{a}) \geq \max_\phi Q_{h,i}^{\phi\diamond\pi^k}(s,\boldsymbol{a}), \quad \underline{Q}_{h,i}^k(s,\boldsymbol{a}) \leq Q_{h,i}^{\pi^k}(s,\boldsymbol{a}), \tag{47}$$

$$\overline{V}_{h,i}^k(s) \geq \max_\phi V_{h,i}^{\phi\diamond\pi^k}(s), \quad \underline{V}_{h,i}^k(s) \leq V_{h,i}^{\pi^k}(s). \tag{48}$$

*Proof.* For each fixed $k$, we prove this by induction from $h = H + 1$ to $h = 1$. For base case, we know at the $(H+1)$-th step, $\overline{V}_{H+1,i}^k(s) = \max_\phi \overline{V}_{H+1,i}^{\phi\diamond\pi^k}(s) = 0$. Now, assume the inequality (40) holds for the $(h+1)$-th step, for the $h$-th step, by definition of the $Q$ functions,

$$\overline{Q}_{h,i}^k(s,\boldsymbol{a}) - \max_\phi Q_{h,i}^{\phi\diamond\pi^k}(s,\boldsymbol{a})$$

$$= \left[ \widehat{\mathbb{P}}_h^k \overline{V}_{h+1,i}^k \right](s,\boldsymbol{a}) - \left[ \mathbb{P}_h \max_\phi V_{h+1,i}^{\phi\diamond\pi^k} \right](s,\boldsymbol{a}) + \beta_t$$

$$= \underbrace{\widehat{\mathbb{P}}_h^k \left( \overline{V}_{h+1,i}^k - \max_\phi V_{h+1,i}^{\phi\diamond\pi^k} \right)(s,\boldsymbol{a})}_{(A)} + \underbrace{\left( \widehat{\mathbb{P}}_h^k - \mathbb{P}_h \right) \max_\phi V_{h+1,i}^{\phi\diamond\pi^k}(s,\boldsymbol{a}) + \beta_t}_{(B)}.$$

By induction hypothesis, for any $s'$, $\left( \overline{V}_{h+1,i}^k - \max_\phi V_{h+1,i}^{\phi\diamond\pi^k} \right)(s') \geq 0$, and thus $(A) \geq 0$. By uniform concentration, $(B) \leq C\sqrt{SH^2\iota/N_h^k(s,\boldsymbol{a})} = \beta_t$. Putting everything together we have $\overline{Q}_{h,i}^k(s,\boldsymbol{a}) - \max_\phi Q_{h,i}^{\phi\diamond\pi^k}(s,\boldsymbol{a}) \geq 0$. The second inequality can be proved similarly.

Now assume inequality (47) holds for the $h$-th step, by definition of value functions and CE,

$$\overline{V}_{h,i}^k(s) = \mathbb{D}_{\pi^k}\overline{Q}_{h,i}^k(s) = \max_\phi \mathbb{D}_{\phi\diamond\pi^k}\overline{Q}_{h,i}^k(s).$$

By Bellman equation,

$$\max_\phi V_{h,i}^{\phi\diamond\pi^k}(s) = \max_\phi \mathbb{D}_{\phi\diamond\pi^k}\max_{\phi'} Q_{h,i}^{\phi'\diamond\pi^k}(s).$$

Since by induction hypothesis, for any $(s, \boldsymbol{a})$, $\overline{Q}_{h,i}^k (s, \boldsymbol{a}) \geq \max_{\phi} Q_{h,i}^{\phi \diamond \pi^k} (s, \boldsymbol{a})$. As a result, we also have $\overline{V}_{h,i}^k (s) \geq \max_{\phi} V_{h,i}^{\phi \diamond \pi^k} (s)$, which is exactly inequality (40) for the $h$-th step. The second inequality can be proved similarly. $\qquad \square$

## H.2 Proof of Theorem 16

In this section, we prove each theorem for the single reward function case, i.e., $N = 1$. The proof for the case of multiple reward functions ($N > 1$) simply follows from taking a union bound, that is, replacing the failure probability $p$ by $Np$.

### H.2.1 NE version

Let $(\mu^k, \nu^k)$ be an arbitrary Nash-equilibrium policy of $\widehat{\mathcal{M}}^k := (\widehat{\mathbb{P}}^k, \widehat{r}^k)$, where $\widehat{\mathbb{P}}^k$ and $\widehat{r}^k$ are our empirical estimate of the transition and the reward at the beginning of the $k$'th episode in Algorithm 4. Given an arbitrary Nash equilibrium $\pi^k$ of $\widehat{\mathcal{M}}^k$, we use $\widehat{Q}_{h,i}^k$ and $\widehat{V}_{h,i}^k$ to denote its value functions of the $i$'th player at the $h$'th step in $\widehat{\mathcal{M}}^k$.

We prove the following two lemmas, which together imply the conclusion about Nash equilibriums in Theorem 16 as in the proof of Theorem 5.

**Lemma 38.** *With probability $1 - p$, for any $(h, s, \boldsymbol{a}, i, k)$, we have*

$$\begin{cases} |\widehat{Q}_{h,i}^k(s, \boldsymbol{a}) - Q_{h,i}^{\pi^k}(s, \boldsymbol{a})| \leq \widetilde{Q}_h^k(s, \boldsymbol{a}), \\ |\widehat{V}_{h,i}^k(s) - V_{h,i}^{\pi^k}(s)| \leq \widetilde{V}_h^k(s). \end{cases} \tag{49}$$

*Proof.* For each fixed $k$, we prove this by induction from $h = H + 1$ to $h = 1$. For base case, we know at the $(H + 1)$-th step, $\widehat{V}_{H+1,i}^k = V_{H+1,i}^{\pi^k} = \widehat{Q}_{H+1,i}^k = Q_{H+1,i}^{\pi^k} = 0$. Now, assume the conclusion holds for the $(h + 1)$'th step, for the $h$'th step, by definition of the $Q$ functions,

$$\left| \widehat{Q}_{h,i}^k (s, \boldsymbol{a}) - Q_{h,i}^{\pi^k} (s, \boldsymbol{a}) \right|$$
$$\leq \left| \left[ \widehat{\mathbb{P}}_h^k \widehat{V}_{h+1,i}^k \right] (s, \boldsymbol{a}) - \left[ \mathbb{P}_h V_{h+1}^{\pi^k} \right] (s, \boldsymbol{a}) \right| + \left| r_h(s, a) - \widehat{r}_h^k(s, a) \right|$$
$$\leq \underbrace{\left| \widehat{\mathbb{P}}_h^k \left( \widehat{V}_{h+1,i}^k - V_{h+1,i}^{\pi^k} \right) (s, \boldsymbol{a}) \right|}_{(A)} + \underbrace{\left| \left( \widehat{\mathbb{P}}_h^k - \mathbb{P}_h \right) V_{h+1,i}^{\pi^k} (s, \boldsymbol{a}) \right| + \left| r_h(s, a) - \widehat{r}_h^k(s, a) \right|}_{(B)}$$

By the induction hypothesis,

$$(A) \leq \widehat{\mathbb{P}}_h^k \left| \widehat{V}_{h+1,i}^k - V_{h+1,i}^{\pi^k} \right| (s, \boldsymbol{a}) \leq (\widehat{\mathbb{P}}_h^k \widetilde{V}_{h+1}^k)(s, \boldsymbol{a}).$$

By uniform concentration (e.g., Lemma 12 in Bai & Jin (2020)), $(B) \leq \sqrt{SH^2 \iota / N_h^k(s, \boldsymbol{a})} = \beta_t$. Putting everything together we have

$$\left| Q_{h,i}^{\pi^k} (s, \boldsymbol{a}) - \widehat{Q}_{h,i}^k (s, \boldsymbol{a}) \right| \leq \min \left\{ (\widehat{\mathbb{P}}_h^k \widetilde{V}_{h+1}^k)(s, \boldsymbol{a}) + \beta_t, H \right\} = \widetilde{Q}_h^k(s, \boldsymbol{a}),$$

which proves the first inequality in (49). The inequality for $V$ functions follows directly by noting that the value functions are computed using the same policy $\pi^k$. $\qquad \square$

**Lemma 39.** *With probability $1 - p$, for any $(h, s, \boldsymbol{a}, i, k)$, we have*

$$\begin{cases} |\widehat{Q}_{h,i}^k(s, \boldsymbol{a}) - Q_{h,i}^{\pi_{-i}^k, \dagger}(s, \boldsymbol{a})| \leq \widetilde{Q}_h^k(s, \boldsymbol{a}), \\ |\widehat{V}_{h,i}^k(s) - V_{h,i}^{\pi_{-i}^k, \dagger}(s)| \leq \widetilde{V}_h^k(s). \end{cases} \tag{50}$$

*Proof.* For each fixed $k$, we prove this by induction from $h = H + 1$ to $h = 1$. For base case, we know at the $(H + 1)$-th step, $\widehat{V}_{H+1,i}^k = V_{H+1,i}^{\pi_{-i}^k,\dagger} = \widehat{Q}_{H+1,i}^k = Q_{H+1,i}^{\pi_{-i}^k,\dagger} = 0$. Now, assume the conclusion holds for the $(h + 1)$'th step, for the $h$'th step, by definition of the $Q$ functions,

$$
\left| \widehat{Q}_{h,i}^k(s, \boldsymbol{a}) - Q_{h,i}^{\pi_{-i}^k,\dagger}(s, \boldsymbol{a}) \right|
$$

$$
= \left| \left[ \widehat{\mathbb{P}}_h^k \widehat{V}_{h+1,i}^k \right](s, \boldsymbol{a}) - \left[ \mathbb{P}_h V_{h+1,i}^{\pi_{-i}^k,\dagger} \right](s, \boldsymbol{a}) \right| + \left| r_h(s, a) - \widehat{r}_h^k(s, a) \right|
$$

$$
\leq \underbrace{\left| \widehat{\mathbb{P}}_h^k \left( \widehat{V}_{h+1,i}^k - V_{h+1,i}^{\pi_{-i}^k,\dagger} \right)(s, \boldsymbol{a}) \right|}_{(A)} + \underbrace{\left| \left( \widehat{\mathbb{P}}_h^k - \mathbb{P}_h \right) V_{h+1,i}^{\pi_{-i}^k,\dagger}(s, \boldsymbol{a}) \right| + \left| r_h(s, a) - \widehat{r}_h^k(s, a) \right|}_{(B)}
$$

By the induction hypothesis,

$$
(A) \leq \widehat{\mathbb{P}}_h^k \left| \widehat{V}_{h+1,i}^k - V_{h+1,i}^{\pi_{-i}^k,\dagger} \right|(s, \boldsymbol{a}) \leq (\widehat{\mathbb{P}}_h^k \widetilde{V}_{h+1}^k)(s, \boldsymbol{a}).
$$

By uniform concentration, $(B) \leq \sqrt{SH^2\iota/N_h^k(s, \boldsymbol{a})} = \beta_t$. Putting everything together we have

$$
\left| Q_{h,i}^{\pi_{-i}^k,\dagger}(s, \boldsymbol{a}) - \widehat{Q}_{h,i}^k(s, \boldsymbol{a}) \right| \leq \min \left\{ (\widehat{\mathbb{P}}_h^k \widetilde{V}_{h+1}^k)(s, \boldsymbol{a}) + \beta_t, H \right\} = \widetilde{Q}_h^k(s, \boldsymbol{a}),
$$

which proves the first inequality in (50). It remains to show the inequality for $V$ functions also hold in $h$'th step.

Since $\pi^k$ is a Nash-equilibrium policy, we have

$$
\widehat{V}_{h,i}^k(s) = \max_\mu \mathbb{D}_{\mu \times \pi_{-i}^k} \widehat{Q}_{h,i}^k(s).
$$

By Bellman equation,

$$
V_{h,i}^{\pi_{-i}^k,\dagger}(s) = \max_\mu \mathbb{D}_{\mu \times \pi_{-i}^k} Q_{h,i}^{\pi_{-i}^k,\dagger}(s).
$$

Combining the two equations above, and utilizing the bound we just proved for $Q$ functions, we obtain

$$
\left| \widehat{V}_{h,i}^k(s) - V_{h,i}^{\pi_{-i}^k,\dagger}(s) \right| \leq \left| \max_\mu \mathbb{D}_{\mu \times \pi_{-i}^k} \widehat{Q}_{h,i}^k(s) - \max_\mu \mathbb{D}_{\mu \times \pi_{-i}^k} Q_{h,i}^{\pi_{-i}^k,\dagger}(s) \right|
$$

$$
\leq \max_{\boldsymbol{a}} \widetilde{Q}_h^k(s, \boldsymbol{a}) = \widetilde{V}_h^k(s),
$$

which completes the whole proof. $\qquad \square$

### H.2.2 CCE VERSION

The proof is almost the same as that for Nash equilibriums. We will reuse Lemma 38 and prove an analogue of Lemma 39. The conclusion for CCEs will follow directly by combining the two lemmas as in the proof of Theorem 5.

**Lemma 40.** *With probability $1 - p$, for any $(h, s, \boldsymbol{a}, i, k)$, we have*

$$
\begin{cases}
Q_{h,i}^{\pi_{-i}^k,\dagger}(s, \boldsymbol{a}) - \widehat{Q}_{h,i}^k(s, \boldsymbol{a}) \leq \widetilde{Q}_h^k(s, \boldsymbol{a}), \\
V_{h,i}^{\pi_{-i}^k,\dagger}(s) - \widehat{V}_{h,i}^k(s) \leq \widetilde{V}_h^k(s).
\end{cases}
\tag{51}
$$

*Proof.* For each fixed $k$, we prove this by induction from $h = H + 1$ to $h = 1$. For base case, we know at the $(H + 1)$-th step, $\widehat{V}_{H+1,i}^k = V_{H+1,i}^{\pi_{-i}^k,\dagger} = \widehat{Q}_{H+1,i}^k = Q_{H+1,i}^{\pi_{-i}^k,\dagger} = 0$. Now, assume the

conclusion holds for the $(h+1)$'th step, for the $h$'th step, by definition of the $Q$ functions,

$$
\begin{aligned}
&Q_{h,i}^{\pi_{-i}^k,\dagger}(s,\boldsymbol{a}) - \widehat{Q}_{h,i}^k(s,\boldsymbol{a}) \\
&\leq \left[\mathbb{P}_h V_{h+1,i}^{\pi_{-i}^k,\dagger}\right](s,\boldsymbol{a}) - \left[\widehat{\mathbb{P}}_h^k \widehat{V}_{h+1,i}^k\right](s,\boldsymbol{a}) + \left|r_h(s,a) - \widehat{r}_h^k(s,a)\right| \\
&\leq \underbrace{\widehat{\mathbb{P}}_h^k\left(V_{h+1,i}^{\pi_{-i}^k,\dagger} - \widehat{V}_{h+1,i}^k\right)(s,\boldsymbol{a})}_{(A)} + \underbrace{\left(\mathbb{P}_h - \widehat{\mathbb{P}}_h^k\right)V_{h+1,i}^{\pi_{-i}^k,\dagger}(s,\boldsymbol{a}) + \left|r_h(s,a) - \widehat{r}_h^k(s,a)\right|}_{(B)}.
\end{aligned}
$$

By the induction hypothesis, $(A) \leq (\widehat{\mathbb{P}}_h^k \widetilde{V}_{h+1}^k)(s,\boldsymbol{a})$.

By uniform concentration, $(B) \leq \sqrt{SH^2\iota/N_h^k(s,\boldsymbol{a})} = \beta_t$. Putting everything together we have

$$
Q_{h,i}^{\pi_{-i}^k,\dagger}(s,\boldsymbol{a}) - \widehat{Q}_{h,i}^k(s,\boldsymbol{a}) \leq \min\left\{(\widehat{\mathbb{P}}_h^k \widetilde{V}_{h+1}^k)(s,\boldsymbol{a}) + \beta_t, H\right\} = \widetilde{Q}_h^k(s,\boldsymbol{a}),
$$

which proves the first inequality in (51). It remains to show the inequality for $V$ functions also hold in $h$'th step.

Since $\pi^k$ is a CCE, we have

$$
\widehat{V}_{h,i}^k(s) \geq \max_\mu \mathbb{D}_{\mu \times \pi_{-i}^k} \widehat{Q}_{h,i}^k(s).
$$

Observe that $V_{h,i}^{\pi_{-i}^k,\dagger}$ obeys the Bellman optimality equation, so we have

$$
V_{h,i}^{\pi_{-i}^k,\dagger}(s) = \max_\mu \mathbb{D}_{\mu \times \pi_{-i}^k} Q_{h,i}^{\pi_{-i}^k,\dagger}(s).
$$

Combining the two equations above, and utilizing the bound we just proved for $Q$ functions, we obtain

$$
\begin{aligned}
V_{h,i}^{\pi_{-i}^k,\dagger}(s) - \widehat{V}_{h,i}^k(s) &\leq \max_\mu \mathbb{D}_{\mu \times \pi_{-i}^k} Q_{h,i}^{\pi_{-i}^k,\dagger}(s) - \max_\mu \mathbb{D}_{\mu \times \pi_{-i}^k} \widehat{Q}_{h,i}^k(s) \\
&\leq \max_{\boldsymbol{a}} \widetilde{Q}_h^k(s,\boldsymbol{a}) = \widetilde{V}_h^k(s),
\end{aligned}
$$

which completes the whole proof. $\qquad\square$

### H.2.3 CE VERSION

The proof is almost the same as that for Nash equilibriums. We will reuse Lemma 38 and prove an analogue of Lemma 39. The conclusion for CEs will follow directly by combining the two lemmas as in the proof of Theorem 5.

**Lemma 41.** *With probability $1-p$, for any $(h,s,\boldsymbol{a},i,k)$ and strategy modification $\phi$ for player $i$, we have*

$$
\begin{cases}
Q_{h,i}^{\phi\diamond\pi^k}(s,\boldsymbol{a}) - \widehat{Q}_{h,i}^k(s,\boldsymbol{a}) \leq \widetilde{Q}_h^k(s,\boldsymbol{a}), \\
V_{h,i}^{\phi\diamond\pi^k}(s) - \widehat{V}_{h,i}^k(s) \leq \widetilde{V}_h^k(s).
\end{cases}
\tag{52}
$$

*Proof.* For each fixed $k$, we prove this by induction from $h = H+1$ to $h = 1$. For base case, we know at the $(H+1)$-th step, $\widehat{V}_{H+1,i}^k = V_{H+1,i}^{\phi\diamond\pi^k} = \widehat{Q}_{H+1,i}^k = Q_{H+1,i}^{\phi\diamond\pi^k} = 0$. Now, assume the conclusion holds for the $(h+1)$'th step, for the $h$'th step, following exactly the same argument as Lemma 40, we can show

$$
Q_{h,i}^{\phi\diamond\pi^k}(s,\boldsymbol{a}) - \widehat{Q}_{h,i}^k(s,\boldsymbol{a}) \leq \min\left\{(\widehat{\mathbb{P}}_h^k \widetilde{V}_{h+1}^k)(s,\boldsymbol{a}) + \beta_t, H\right\} = \widetilde{Q}_h^k(s,\boldsymbol{a}),
$$

which proves the first inequality in (52). It remains to show the inequality for $V$ functions also hold in $h$'th step.

Since $\pi^k$ is a CE, we have

$$\widehat{V}_{h,i}^k\left(s\right) = \max_{\tilde{\phi}_{h,s}}\mathbb{D}_{\tilde{\phi}_{h,s}\diamond\pi^k}\widehat{Q}_{h,i}^k\left(s\right),$$

where the maximum is take over all possible injective functions from $\mathcal{A}_i$ to itself.

Observe that $V_{h,i}^{\phi\diamond\pi^k}$ obeys the Bellman optimality equation, so we have

$$V_{h,i}^{\phi\diamond\pi^k}\left(s\right) = \max_{\tilde{\phi}_{h,s}}\mathbb{D}_{\tilde{\phi}_{h,s}\diamond\pi^k}Q_{h,i}^{\phi\diamond\pi^k}\left(s\right).$$

Combining the two equations above, and utilizing the bound we just proved for $Q$ functions, we obtain

$$V_{h,i}^{\phi\diamond\pi^k}\left(s\right) - \widehat{V}_{h,i}^k\left(s\right) = \max_{\tilde{\phi}_{h,s}}\mathbb{D}_{\tilde{\phi}_{h,s}\diamond\pi^k}Q_{h,i}^{\phi\diamond\pi^k}\left(s\right) - \max_{\tilde{\phi}_{h,s}}\mathbb{D}_{\tilde{\phi}_{h,s}\diamond\pi^k}\widehat{Q}_{h,i}^k\left(s\right)$$

$$\leq \max_{\boldsymbol{a}}\widetilde{Q}_h^k(s,\boldsymbol{a}) = \widetilde{V}_h^k(s),$$

which completes the whole proof. $\qquad\square$

