# OpenReview forum: "A Sharp Analysis of Model-based Reinforcement Learning with Self-Play"
_ICLR.cc/2021/Conference — Reject_

### Official Review · AnonReviewer3 · 2020-10-28
**Review of A Sharp Analysis of Model-based Reinforcement Learning with Self-Play**

**Rating:** 4
**Confidence:** 5

**Review:**

-Summary
The authors consider self-play in tabular zero-sum episodic Markov game. In this setting, the goal is to learn an \epsilon approximate of the Nash equilibrium of the Markov game while minimizing the sample complexity, i.e. the number of episode played by the agent.  They present Optimistic Nash
Value Iteration (Nash-VI) that output with high probability a pair of policies that attains an \epsilon approximate Nash equilibrium in O(H^3SAB/\epsilon^2) episodes where H is the horizon, S the number of states,  A and B the number of actions for the max-player respectively min-player. This rate matches the lower bound of order \Omega(H^3S(A+B)/\epsilon^2) by Jin et al. (2018) up to a factor min(A,B). They extend this result to the multi-player setting with Multi-Nash-VI algorithm of sample complexity O(H^4S^2\prod_i A_i/\epsilon^2) where A_i is the size of the action space of player i. The authors also provide VI-Zero an algorithm for reward-free exploration of N-tasks in Markov game with a sample complexity of O(H^4SAB log N/\epsilon^2) and a lower bound of order \Omega(H^2SAB/\epsilon^2).


-Contributions
algorithmic: Nash-VI (significance: medium)
theoretical: Nash-VI sample complexity of order O(H^3SAB/\epsilon^2) (significance: high)
algorithmic: VI-zero (significance: low)
theoretical: VI-zeros sample complexity of order O((H^4SAB log N/\epsilon^2) (significance: medium)
algorithmic: Multi-Nash-VI (significance: medium)
theoretical:  Multi-Nash-VI sample complexity  of order O((H^4S \Prod_i A_i/\epsilon^2)  (significance: medium)
theoretical:  lower bound for the sample complexity of reward-free exploration in Markov game of order \Omega(H^2SAB/\epsilon^2)

-Score justification/Main comments
The paper is well written. The authors use the same technical tool as Azar et al. 2017 to get a sharp dependence in the horizon H and the state space size S. Precisely they use Bernstein bonuses in combination with the Law of total variance in the Markov game to obtain the H^3 and only concentrate the empirical transition along the optimal value function to push the S^2 into second-order terms.
I have mixed feelings concerning this paper. On one hand, I think the contributions deserve to be published on the other hand I not convinced by the proof. Indeed the proofs are dangerously close to a sketch of proofs (see specific comments below) when it is not the case (see proof of Theorem 6). Even if I think most of the issues are fixable, considering the number of corrections required, I would need to read the updated version to assert if the results are correct.

The algorithm that attains a dependence A+B instead of AB is almost adversarial, do you think it is possible to obtain the same result with a model-based algorithm that uses the stochastic assumption, or do you see a fundamental reason why it will be impossible?

-Specific comments
P1: What do you mean by information-theoretic lower bound?

P2, Table 1: It could be interesting to compare these algorithms on a toy example. At least implement Algorithm 1 to prove it is feasible.

P3: precise what is \pi when you introduce D_{\pi}Q

P5: Which (estimation) error the bonus \gamma is compensating precisely?

P6, Algorithm 1: The definition of the empirical transitions and the bonuses are not clear when N_h(s_h,a_h,_b_h) = 0.

P7, comparison with model-free approaches: could you also compare them in term of computational complexity.

P12, Non-asymptotic […] assumptions: precise what do you mean by highly sub-optimal because at the end Algorithm 1 is also sub-optimal by a factor min(A,B).

P15, (Approximate CE): I think that \phi \circ\pi is not a good choice of notation since it is not really composition.

P15, Th 15: Could we, as in the two-player, deduce a Nash equilibrium by only computing CCE and get a polynomial-time algorithm?

P19,Lemma 19: I do not understand the statement of the lemma, since \bar{V}^k_{h+1}-… is a random variable you need to precise in which sense the inequality |V|(s) \leq … holds. In particular, considering how you use it in the sequel it is not almost surely. Furthermore in the proof, you need to deal with the case when N_h^k = 0 and you cannot apply the empirical Bernstein Bound from Maurer and Pontil like that since you have a random number of samples. An additional union bound over the possible value of N_h^k is required and you need to prove that conditionally to this choice the samples are independent….

P19, proof of Lemma 19: you need to precise which event you consider in order to be able to invoke Lemma 18.
P20, top of the page: the two equalities are wrong in (10) they are inequality because of the clipping for the first one and because \hat{p}_h^k (\bar{V} ….) \geq 0. What do you mean by “with high probability” exactly? And since it is Hoeffding inequality the second term in 1/n is not necessary.

P21, proof of Theorem 3: Again you need to precise with the event you consider to be able to call Lemma 18 or 19 … and guarantee that this event is of probability at least 1-p. Currently, a lot of union bounds are hidden whereas they should be treated properly.
\zeta_h^k and \xi_h^k are martingales with respect to which filtration?
When you apply Azuma-Hoeffding inequality you implicitly imply that you can upper-bound all the \zeta_h^k \xi_h^k by the same constant (moving the bi O outside the sum) but you cannot because they are not necessarily positive.
For the pigeon-hole argument again you should consider the case N_h^k=0
After (12) it is \sqrt(H^3SABT).

P22, Lemma 20: same remarks as Lemma 18 and 19.

P23, Lemma 21: Because of the big O after “these terms […] separately” it seems that the constants in Lemma 21 are wrong. Furthermore, here you apply Hoeffding inequality plus a union bound over an \epsilon-net or control in KL to obtain the H^2\sqrt{S/n} bounds. You need to explain this properly.

P24, proof of Theorem 4: same remark as for the proof of Theorem 3. For (i) it is not only the Law of total variation but also because of the Hoeffding Azuma inequality ( and in Azar et al. it is only proved for stationary transition).

P27, proof of Theorem 6: since you only prove the theorem for H=S=1 you should only state this case and keep the general case as a conjecture.
S=H=1 in Markov game (where there is nothing to learn) is not equivalent to a matrix game. What do you mean exactly by reward-free matrix game? The agent does not observe the rewards, but in this case, there is nothing to learn, no? I do not think it easily generalizes to the Markov games setting. Could you also obtain the factor \log(N) in the lower bound?

---

> ### Author Response · Authors · 2020-11-20
> **Response to Reviewer 3 (1/2)**
>
> Thank you very much for your detailed feedback and  valuable comments. We have carefully revised the draft accordingly. Please see our responses below.
>
> $\mathbf{Q}$. "The authors use the same technical tool as Azar et al. 2017 to get a sharp dependence in the horizon $H$ and the state space size $S$."\
> 	$\mathbf{A}$.
> 	Naively applying the techniques in Azar et al. (2017) will not improve the $S^2$ dependence in the previous model-based results (e.g., VI-ULCB in Bai \& Jin, (2020)).
> 	In this paper, we utilize a different style of bonus term $\gamma$, in addition to the standard bonus $\beta$ to achieve the optimal dependence on $S$.
> 	Both this new bonus term and the techniques for analyzing it are not present in Azar et al. (2017).
>
> $\mathbf{Q}$. "Indeed the proofs are dangerously close to a sketch of proofs (see specific comments below)"\
> 		$\mathbf{A}$. We believe our proofs include all the key steps
> 	and we only omitted some standard arguments that have been repeatedly done in the literature.
> 	To make our proofs more accessible to the general audience, we have added more explanations in the updated version.
>
> $\mathbf{Q}$. The algorithm that attains a dependence $A+B$ instead of $AB$ is almost adversarial, do you think it is possible to obtain the same result with a model-based algorithm that uses the stochastic assumption, or do you see a fundamental reason why it will be impossible?\
> $\mathbf{A}$. So far, it remains unclear to us how to achieve a $A+B$ dependence by model-based algorithms.
> 	In this work, the $AB$ dependence comes from using the pigeon-hole principle with $HSAB$ 'holes'.
> 	We think it might require some new mechanism  to avoid this argument for model-based algorithms. We agree this is an interesting future direction worth exploring.
>
> $\mathbf{Q}$. P1, What do you mean by information-theoretic lower bound?\
> $\mathbf{A}$. By information-theoretic lower bound, we mean the sample complexity lower bound for {\bf any} possible algorithm, rather than for one specific (type of) algorithm.
>
> $\mathbf{Q}$. "P5, Which (estimation) error the bonus $\gamma$ is compensating precisely?" "And since it is Hoeffding inequality the second term in $1/n$ is not necessary."\
> $\mathbf{A}$. The bonus term $\gamma$ together with the $\mathcal{O}(1/N)$ term in
> $\beta$ controls the error $(A)$ in Lemma 20 (see the updated version). This is the reason for adding the $\mathcal{O}(1/N)$ term in the Hoeffding bonus.
>
> $\mathbf{Q}$. P6, Algorithm 1: The definition of the empirical transitions and the bonuses are not clear when $N_h(s_h,a_h,b_h)=0$.\
> $\mathbf{A}$.    Our algorithm utilizes the empirical transition and the bonus only when $N_h(s_h,a_h,b_h)>0$. This is guaranteed by the "if statement" in line 6, Algorithm 1.
>
> $\mathbf{Q}$. P7, Comparison with model-free approaches: could you also compare them in term of computational complexity.\
> $\mathbf{A}$. Thank you for the suggestion.
>     Both model-based and model-free algorithms have polynomial running time.
>      For learning Markov games,  our model-based algorithm has better space complexity while the current model-free ones have better computational complexity.
>     In Nash-VI (Algorithm 1), we need to solve LPs for computing CCEs which makes it slower than the model-free algorithms.
>      However, the output policy of model-free ones are nested mixture of Markov policies so they need  roughly  $\tilde{\Theta}(H^6S^2(A+B)^2/\epsilon^2)$ space for computing and storing an $\epsilon$-optimal policy. In contrast, our algorithm only requires $\tilde{\Theta}(HS(A+B))$ space. We have added a comparison in the updated version.
>
> $\mathbf{Q}$. P12, Non-asymptotic $[\ldots]$ assumptions: precise what do you mean by highly sub-optimal because at the end Algorithm 1 is also sub-optimal by a factor $\min(A,B)$.\
> $\mathbf{A}$. Please refer to Table 1 and the paragraph below it on page 2 for a detailed comparison of sample complexity.
>       For your convenience, the sample complexity of VI-ULCB (Bai \& Jin, 2020), OMVI-SM (Xie et al., 2020) and Nash-VI (this work) are $\tilde{\mathcal{O}}(H^4S^2AB/\epsilon^2)$,  $\tilde{\mathcal{O}}(H^4S^3A^3B^3/\epsilon^2)$ and $\tilde{\mathcal{O}}(H^3SAB/\epsilon^2)$, respectively.
>       Therefore, VI-ULCB is suboptimal by a factor $HS\min(A,B)$ and
>       OMVI-SM is suboptimal by a factor $HS^2(\max(A,B))^2(\min(A,B))^3$.
>       In contrast, our algorithm is sub-optimal only by a factor $\min(A,B)$.
>
>  $\mathbf{Q}$. P15, Th 15: Could we, as in the two-player, deduce a Nash equilibrium by only computing CCE and get a polynomial-time algorithm?\
> $\mathbf{A}$. Two-player zero-sum games, where the Nash equilibria can be efficiently computed, are very special cases of multi-player general-sum games.  Violating either 'two-player' or 'zero-sum' will lead to computational hardness for computing Nash equilibria (Daskalakis, 2013).

---

> > ### Author Response · Authors · 2020-11-20
> > **Response to Reviewer 3 (2/2)**
> >
> > $\mathbf{Q}$. P27, proof of Theorem 6: $S=H=1$ in Markov game (where there is nothing to learn) is not equivalent to a matrix game. What do you mean exactly by reward-free matrix game? The agent does not observe the rewards, but in this case, there is nothing to learn, no? \
> > $\mathbf{A}$. Thank you for the questions. We have added a detailed introduction of the reward-free (or reward-agnostic) setting at the beginning of Section 4 in the updated version. Back to your questions about reward-free matrix games, where the only thing to learn is the unknown reward function.  More specifically, in reward-free matrix games, the reward is unknown and stochastic. In the exploration phase, the agent needs to determine which action pairs to pick  without receiving any reward feedback. After the exploration phase, the rewards of the picked action pairs will be sampled from the corresponding reward distribution and revealed to the agent. The goal is to compute Nash equilibria for matrix games using the revealed stochastic reward information.
> >
> > $\mathbf{Q}$. P27, proof of Theorem 6: since you only prove the theorem for $H=S=1$ you should only state this case and keep the general case as a conjecture.  I do not think it easily generalizes to the Markov games setting. Could you also obtain the factor $\log(N)$ in the lower bound?\
> > $\mathbf{A}$.
> >      Generalizing the lower bound to the setting of reward-free Markov games
> >      follows from standard argument.
> >      For the purpose of completeness, we have included a full proof of this argument (see Appendix G.3.2) in the updated version. Our lower bound does not include the $\log(N)$ factor but we believe it is doable by further combining with the proof techniques in Zhang et al. (2020), where they  prove a $\Omega(\log(N)H^2SA/\epsilon^2)$ lower bound for reward-free learning of MDPs.

---

> > ### Author Response · Authors · 2020-11-24
> > **Acknowledgement of the reviews**
> >
> > We sincerely apologize for the misunderstanding that our original response might have brought. We highly appreciate your deep and thoughtful reviews, and have modified our paper according to many of your great suggestions. We have not responded to the 3 particular fixes raised by AC, because we thought those three issues are minor, and are already self-explained by revisions. Thank you again for your detailed reviews and constructive feedback.

---

### Official Review · AnonReviewer1 · 2020-10-29
**Impressive theoretical claims, some editing needed**

**Rating:** 7
**Confidence:** 2

**Review:**

The authors study reinforcement learning in two-player (and more) Markov games, providing new algorithms and bounds. They reduce the dependence on horizon H and states S from H^4 S^2 to H^3 S, matching information theoretic lower bound for these factors.

Overall, my impression is that the results are very valuable. In a short paper like this, it's difficult to convey (and absorb) the fundamental theoretical insights behind it.

Some additional work should be done to improve the presentation (see below).

Detailed edits:

"Model-free algorithms has also" -> "Model-free algorithms have also"

"finds eps-approximate Nash equilibrium" -> "finds eps-approximate Nash equilibria"

"multi-player general-sum Markov games": I thought that's what we've been talking about since the beginning. Explain? Does "multi-player" mean more than two? Was it clear we were talking about only two up to this point?

"the runtime is PPAD-complete" -> "the algorithm requires solving a PPAD-complete problem"??

"We remark that the current policy is random" -> "We remark that the current policy is stochastic". (To me, "is random" reads as "is chosen at random".)

"in Azar, et al. Azar et al. (2017) can not" -> "of Azar et al. (2017) cannot".

"In order to achieve so," -> "To do so,"

"needs to scale as large as" -> "needs to be as large as". Also, shouldn't the big O here be a big theta? After all big O means "is not bigger than", so "needs to be as large as not bigger than" conveys no useful information.

"a relaxation of Nash equilibrium" -> "a relaxation of the Nash equilibrium".

"Coarse Correlated Equalibirum (CCE) instead, a" -> "Coarse Correlated Equiliibrum (CCE)---instead, a".

"Therefore, executing such policy requires the cooperation of two players during the training phase.": Wouldn't it also require cooperation during any other phase as well? Also, shouldn't this observation be part of the problem definition? You are learning policies only in the setting where the algorithm is controlling both (all?) players. That's different from the earliest work (Rmax), which provided bounds even when the players are independent.

"using Bernstein bonus instead of Hoeffding bonus" -> "using a Bernstein bonus instead of a Hoeffding bonus".

"with Bernstein bonus" -> "with the Bernstein bonus"

"achieves optimal dependency" -> "achieves the optimal dependency"

"pair of Markov policy" -> "pair of Markov policies"

"can not" -> "cannot" (throughout).

"into the reward-free exploration setting" -> "for the reward-free exploration setting"

"Due to space limit" -> "Due to space limits".

"comes from the sample complexity only scales" -> "comes from the sample complexity only scaling".

"rerunned" -> "rerun".

"any more" -> "anymore"

"have regret guarantee" -> "have a regret guarantee"

"Since MDPs is" -> "Since MDPs are"

"both exploration" -> "both the exploration"

"allows arbitrary" -> "allows an arbitrary"

"in Theorem 5 scales" -> "in Theorem 5 scaling"

"we defer the detailed setups": Not just the details... you are claiming to provide Algorithms 3 and 4, which only appear there.

"How to design" -> "How can we design"

"We would like to leave these as future work." -> "We leave these problems as future work."?

"On the complexity of approximating a nash equilibrium" -> "On the complexity of approximating a Nash equilibrium".

---

> ### Author Response · Authors · 2020-11-20
> **Response to Reviewer 1**
>
> Thank you very much for your positive feedback and valuable comments. We have carefully revised the draft accordingly. Please see our responses below.
>
>  $\mathbf{Q}$. "multi-player general-sum Markov games": I thought that's what we've been talking about since the beginning. Explain? Does "multi-player" mean more than two? Was it clear we were talking about only two up to this point?\
>           $\mathbf{A}$. By "multi-player", we mean more than two players. Our Sections 2, 3, 4 consider the special case: two-player zero-sum Markov games. Our Section 5 then relaxes this to any number of players more than two, and general-sum reward functions.
>
> $\mathbf{Q}$. "Therefore, executing such policy requires the cooperation of two players during the training phase.": Wouldn't it also require cooperation during any other phase as well? \
> 	$\mathbf{A}$. The cooperation is only required during the training phase, i.e. when running Algorithm 1 to find nearly optimal policies. Executing the output policy of Algorithm 1 does not require cooperation (see the paragraph above Section 3.1.1).
>
> $\mathbf{Q}$. You are learning policies only in the setting where the algorithm is controlling both (all?) players. That's different from the earliest work (Rmax), which provided bounds even when the players are independent.\
> 	$\mathbf{A}$. The learning objectives of this paper and Rmax are different.\
> 		In this paper, the primary goal is to find Nash equilibria for Markov games, which are optimal against the best responses of the opponents (see Section 2 for the definition of best response and Nash equilibria).
> 		If an algorithm only controls one player, it is impossible to find Nash equilibria.
> 	 The reason is if the other player is dummy and always picks a fixed action, the algorithm will be unable to explore the game sufficiently. \
> 		 In the Rmax paper, the algorithm only controls one player and the goal is to perform approximately as good as the best policy in hindsight. In other words, it only cares about how good the algorithm can play against the current opponent. If the opponent is very weak, the output policy of Rmax can perform arbitrarily poorly when playing against the best response.

---

### Official Review · AnonReviewer4 · 2020-10-30
**Excellent theoretical paper**

**Rating:** 8
**Confidence:** 4

**Review:**

Summary
-------

The authors introduce new algorithms to solve two-players zero-sum Markov games, as well as two-players Markov game in the reward-free setting. The approach is model-based, based on successive episods of planning and counting for updating the model estimate. It involves solving a matrix game at each iteration, looking for a notion of equilibria that is computable in polynomial time (unlike Nash equilibria) An extension to multi-player games is proposed for both reward and reward free setting (in the appendix).

The sample complexity of the zero-sum Markov game algorithm (Nash-VI), has a
sample complexity that is closer to known lower-bounds than existing work, and which is better in the horizon dependency than the best model-free algorithm.

In this zero-sum setting, the improvement to existing work (VI-UCLB) are the following:
-  use of an auxiliary bonus that is proportional to the gap of upper-confidence and lower-confidence value function, that allows to use much-smaller standard Hoeffding/Bernstein bonuses
- use of a relaxed notion of equilibria when looking for the next policy given the Q functions of both player. This relaxed notion is introduced in Xie et al. 2020

The reward-free setting is simpler, in that it simply use greedy policies for each player, with each player maintaining artifical rewards based on Hoeffding bonuses.

Review
------

The paper is very well written and presents some exciting results. It is completely theoretical, but provides two different algorithms for two different settings, in both the two-player and n-player setting. The improvement in existing bounds is significant. I have only slight concerns:

- The algorithmic contribution (Alg. 1) could be seen as incremental, as it changes two elements in known algorithm, one of which (coarse correlated equilibria), having been used for a similar purpose in a previous paper. Similarly, Alg 2. is at the end of the day a rather naive extension of zero-reward exploration in single player MDP. Yet the notion of auxiliary bonus is very original, and the reduction of complexity non-trivial.

- The absence of experiments, even in toy setting, is regrettable. It is
especially true as the use of coarse correlated equilibria may be expensive, and
I would have appreciated seing Alg. 1 implemented. As this is ICLR, extensions
to function approximations would also be interesting. In particular, comparison
with model-free approaches would be welcome, as constants before the sample
complexities may vary.

- Some parts of the text could be further explained: in particular the
intuitions behind coarse correlated equilibria, which is introduced only mathematically.

- The paper theoretical content is rather heavy, which may make this manuscript more suitable for a journal venue, where it would be more thoroughly reviewed. I must admit that I could not proof read the entire appendix.

 I have several questions, as follow:

- I do not understand the note "Our results directly generalize to randomized reward
functions, since learning the transition is more difficult than learning the reward." Could you elaborate on this aspect.

- Is there any reason why we would like to use Hoeffding bonuses instead of Bernstein bonuses in the rewarded case ?

- In the multi-player, rewarded case, it appears that using coarse correlated equilibrium instead of Nash equilibrium yields non-product policies, which is unfortunate. Is there any way that we could obtain product policies solving a relaxed notion of equilibria that would be computable in polynomial time ? Similarly, is there a foreseeable way in which $\Pi A_i$ could be transformed in a sum ? Lower-bounds in this case are not discussed in this case, is there any ?

---

> ### Author Response · Authors · 2020-11-20
> **Response to Reviewer 4**
>
> Thank you very much for your positive feedback and valuable comments. We have carefully revised the draft accordingly.
> Please see our responses below.
>
> $\mathbf{Q}$. I do not understand the note "Our results directly generalize to randomized reward functions, since learning the transition is more difficult than learning the reward." Could you elaborate on this aspect.\
> $\mathbf{A}$. Intuitively, to learn the reward, one only needs to estimate a scalar for each $(s,a)$. In contrast, learning the transition requires to estimate a probability distribution over $\mathcal{S}$ for each $(s,a)$.
> As a result, many works (e.g. Azar et al. (2017) and Jin et al. (2018)) assume known and deterministic reward for a cleaner presentation
> because extending their analysis to
> unknown stochastic rewards poses no real difficulty.
>
> $\mathbf{Q}$. Is there any reason why we would like to use Hoeffding bonuses instead of Bernstein bonuses in the rewarded case?\
> $\mathbf{A}$. Algorithmically, even though the Bernstein bonus gives stronger bounds, the Hoeffding bonus is cleaner and closer to the $1/\sqrt{t}$ style bonuses used in practice. Also, the Hoeffding bonus is easier to analyze (though giving a weaker bound) and allows for a cleaner proof, thus we presented it as an intermediate result.
>
> $\mathbf{Q}$. In the multi-player, rewarded case, it appears that using coarse correlated equilibrium instead of Nash equilibrium yields non-product policies, which is unfortunate. Is there any way that we could obtain product policies solving a relaxed notion of equilibria that would be computable in polynomial time? \
>  $\mathbf{A}$. So far, we are not aware of any such relaxation. We believe this is an interesting question even in game theory. We agree this is an interesting future direction worth exploring.
>
> $\mathbf{Q}$.  Similarly, is there a foreseeable way in which $\prod A_i$
>  could be transformed in a sum? Lower-bounds in this case are not discussed in this case, is there any?\
>  $\mathbf{A}$.
>  Bai et al. (2020) propose an algorithm that can find an $\epsilon$-approximate Nash equilibrium for two-player zero-sum Markov games using $\tilde{\mathcal{O}}(H^6S(A+B)/\epsilon^2)$ samples. However, it remains unclear how to adapt their approach and theoretical guarantees to the multi-player general-sum setting.
>  To our knowledge, there is no lower bounds known except for a trivial one  $\mathcal{O}(H^3S(\max_i A_i)/\epsilon^2)$. We agree these are interesting future directions worth exploring.
>
> $\mathbf{Q}$. Some parts of the text could be further explained: in particular the intuitions behind coarse correlated equilibria, which is introduced only mathematically.\
> $\mathbf{A}$. Intuitively, in a CCE the players
> choose their actions in a potentially correlated way such that no one can benefit from  unilateral unconditional deviation. We have added the explanation in the updated version.

---

### Official Review · AnonReviewer2 · 2020-11-01
**Review of Paper 374**

**Rating:** 5
**Confidence:** 4

**Review:**

This paper studies learning in stochastic games, which are extensions of Markov decision processes (MDPs) from the single-agent setup to the multi-agent one. Here the objective of each learner is to optimize her own reward function. Similarly to the case of MDPs, here one can devise learning algorithms with controlled sample complexity or regret (or both simultaneously) even when reward and transition functions are unknown.

The main contribution of the paper is a model-based algorithm called Nash-VI, enjoying simultaneously a regret bound of $\widetilde O(\sqrt{H^3SABT})$ after $T$ steps and a PAC-type sample complexity of $\widetilde {\mathcal O}(H^3SAB/\epsilon^2)$ for finding an $\epsilon$-approximate optimal policy.

The paper also presents an extension of Nash-VI to the case of reward-free exploration, which is called the VI-Zero algorithm. The sample complexity of VI-Zero depends logarithmically on the number of candidate reward functions, but has a worse dependence on $S$ than that of Nash-VI.

Main Comments:

The paper is very well-organized and well-written. There are a number of minor easy-to-fix typos that are reported below. The paper overall delivers a clear presentation of algorithms and results, and expect for a few unclear sentences (listed below), it is very a nice read.

On the technical side, one strong aspect of Nash-VI is that its sample complexity (almost) matches the lower bound except for dependencies on the action set cardinalities. Achieving a near-optimal regret bound is another strong aspect. Finally, the main algorithm outputs a single Markov policy, and not a history-dependent policy.

For the MDP setup, it is already well established that model-based algorithms are minimax-optimal in terms of both regret and sample complexity. The paper makes a good step towards understanding the benefits of model-based algorithms for stochastic games, thus indicating that their sample complexity could almost match the existing minimax lower bound.

It turns out that the presented algorithms mostly rely on existing tools already developed for the MDP setup, which are by now fairly standard. That said, the paper does not present any fundamentally different technique than existing ones for MDPs. However, I believe suitably combining all these tools is not a trivial task and deriving such sharp sample complexities requires care. This is in particular true for bonus $\gamma$ discussed in p. 5. Overall the paper conveys interesting messages advancing our understanding of model-based algorithms for stochastic games, and in my opinion is worth accepting.

I was unable to check the proofs in such a limited review period, but they appear correct to me.

Minor Comments:

- In p. 6 “during the training phase” please clarify.

- In the statement: “would not hurt the overall sample complexity up to a constant factor”: Did you mean “would hurt the overall sample  complexity only up to a constant factor”

- In p. 7, when referring to $\mathcal M(\mathbb P, r)$, it is not clear which reward function $r$ is. Does it one belonging to a set of reward functions? Please clarify.

Some typos:
p. 1: with the problem multi-agent -> … the problem of multi-agent
p. 1: Model-free algorithms has -> … have
p. 3: Markov games is -> … are
p. 3: distribution of actions -> … over actions
p. 6: the policies … is -> … are
p. 6: such policy -> such a policy
p. 6: a $\epsilon$-approximate -> an …
p. 8: since MDPs is a special case of -> since MDPs are special cases of
p. 8: rerunned -> rerun

Thanks for updating the paper in light of my earlier comments.

After a long discussion with other reviewers and ACs, we concluded that the paper would require another complete review process in view of newly added proofs, which were unfortunately missing in the first round. My lowered score signals this to the ACs.

I would like to highlight that I became very upset to find out that many important details were skipped and left as exercise to readers. While this would make sense for some repetitive details within a paper, it may not apply to non-trivial proof details, such as the generalization from 2 to arbitrary S or similar.

---

> ### Author Response · Authors · 2020-11-20
> **Response to Reviewer 2**
>
> Thank you very much for your positive feedback and valuable comments. We have carefully revised the draft accordingly. Please see our responses below.
>
> ${\mathbf{Q}}$. In p. 6 “during the training phase” please clarify. \
> ${\mathbf{A}}$. Here, by "during the training phase", we mean "when executing a CCE policy in line 17, Algorithm 1". We have clarified this point in the updated version.
>
> ${\mathbf{Q}}$. In the statement: “would not hurt the overall sample complexity up to a constant factor”: Did you mean “would hurt the overall sample complexity only up to a constant factor”\
> ${\mathbf{A}}$. Yes, we mean "would hurt the overall sample complexity only up to a constant factor". We have fixed it in the updated version.
>
> ${\mathbf{Q}}$. In p. 7, when referring to $\mathcal{M}(\mathbb{P},r)$, it is not clear which reward function  is. Does it one belonging to a set of reward functions? Please clarify.\
> ${\mathbf{A}}$. $r$ needs to be one of the $N$ arbitrary unknown reward functions that are chosen independent of the randomness of our algorithm. We have clarified it in the updated version.

---

### Comment · Area_Chair1 · 2020-11-23
**Flagging an issue in the author's response to AnonReviewer3**

I would like to point to a very serious issue with the author’s reply  to the review of AnonReviewer3.

First, the reviewer went all the way to check the appendix and the proof in detail and pointed out problematic sections where the proof would not go through or was incomplete and the paper could be accepted in this state.  What's more,  the reviewer went an extra mile and suggested fixes to the proofs and the analysis, in particular:

- correct the lemma 19,
- add max with 1 to deal with initialization
- introduce an event for the union bound...

This amounts to an excellent review that improves this work.  However, I see that

1° The authors did not acknowledge nor responded to the issues raised.
2° The authors uploaded a new version where they applied the corrections from this review.
3° The authors responded with “make our proofs more accessible to the general audience” and  “only omitted some standard arguments that have been repeatedly done in the literature” while the proofs the authors provided before the corrections coming from this reviewer were wrong. This response is somewhere between using euphemisms and plain rude.

I therefore flagging this behavior as violating https://iclr.cc/public/CodeOfConduct.

---

> ### Author Response · Authors · 2020-11-24
> **Response to AC**
>
> We thank the AC for raising this issue. We have no intention to ignore or discredit AnonReviewer3, and in fact we highly appreciate the deep and thoughtful reviews by AnonReviewer3, and have modified our paper according to many of his great suggestions. We sincerely apologize if our response reads dismissive, and will be more careful about our language in the future.
>
> However, in our defense, we respectfully disagree with the claim that “the proofs the authors provided before the corrections coming from this reviewer were wrong”. We do believe our proofs before the revision are technically correct, except for a few easily fixable issues.
>
> In particular, in response to the three major fixes highlighted by AC:
>
> 1) “correct the lemma 19”: Our Lemma 19 is correct before revision. The issue is that we did not explain well about the settings of Lemma 19. We thank AnonReviewer3 for pointing this out. In fact, we only added the clarifications to Lemma 19. We have not changed the proof of Lemma 19 at all, except for a few typos.
> 2)  “add max with 1 to deal with initialization”: This is a simple fix replacing all counts N with max{N,1} to deal with the initial case N=0. We thank AnonReviewer3 for pointing out this N=0 issue. We would also like to kindly point out that (a) This “max with 1” solution was not proposed by AnonReviewer3. (b) “max with 1 to deal with initialization” is a very standard technique that has been widely adopted in bandit, reinforcement learning literature [see e.g., 1, 2, 3, 4, 5].
> 3)  “introduce an event for the union bound”: This is only a matter of writing style, and our concentration inequalities before revisions are all correct. We have not changed any technical concentration inequality or usage of union bound. We introduce the event in our revision only to clarify our proofs in response to the suggestions of AnonReviewer3. These union bound arguments are indeed standard arguments that have been repeatedly used in the reinforcement learning literature [see e.g., 2, 3, 4, 5].
>
> In addition, in response to “The authors did not acknowledge nor responded to the issues raised.”: We have acknowledged and responded to many great comments pointed out by AnonReviewer3. We have not responded to the particular three issues mentioned above, because, as explained above, we personally considered the above issues as minor clarification problems or easily fixable problems, which are self-explained by revisions. We are sincerely sorry about the negative impression that our original responses brought.
>
> Finally, we would like to reiterate our great appreciation of all reviewers’ helpful feedback. We have made a deep pass cleaning up the typos, and added a significant amount of explanations in correspondence to reviewers’ comments. We hope our such attempt will not be considered as trying to discredit AnonReviewer3.
>
>
> Reference\
> [1] Tor Lattimore, Csaba Szepesvári. Bandit algorithms. Cambridge University Press, 2020.\
> [2] Thomas Jaksch, Ronald Ortner, and Peter Auer. Near-optimal regret bounds for reinforcement learning. Journal of Machine Learning Research, 2010.\
> [3] Mohammad Gheshlaghi Azar, Ian Osband, and Rémi Munos. Minimax regret bounds for reinforcement learning. ICML, 2017.\
> [4] Chi Jin, Zeyuan Allen-Zhu, Sebastien Bubeck, and Michael I Jordan. Is Q-learning provably efficient? NeurIPS, 2018.\
> [5] Andrea Zanette, Emma Brunskill. Tighter problem-dependent regret bounds in reinforcement learning without domain knowledge using value function bounds. ICML, 2019.

---

### Decision · Program_Chairs · 2021-01-07
**Final Decision**

**Decision:**

Reject

**Comment:**

The reviewers, AC, and PCs participated in a very thorough discussion. AC ultimately felt that the work was unfinished, and in particular that details in the proofs still needed work before publication.